# Stochastic Optimization with Heavy-Tailed Noise via Accelerated Gradient Clipping

**Eduard Gorbunov**[*]
MIPT and HSE, Russia

**Marina Danilova**[†]
ICS RAS and MIPT, Russia

**Alexander Gasnikov**[‡]
MIPT and HSE, Russia

## Abstract

In this paper, we propose a new accelerated stochastic first-order method called `clipped-SSTM` for smooth convex stochastic optimization with heavy-tailed distributed noise in stochastic gradients and derive the first high-probability complexity bounds for this method closing the gap in the theory of stochastic optimization with heavy-tailed noise. Our method is based on a special variant of accelerated Stochastic Gradient Descent (`SGD`) and clipping of stochastic gradients. We extend our method to the strongly convex case and prove new complexity bounds that outperform state-of-the-art results in this case. Finally, we extend our proof technique and derive the first non-trivial high-probability complexity bounds for `SGD` with clipping without light-tails assumption on the noise.

## 1 Introduction

In this paper we focus on the following problem

$$\min_{x \in \mathbb{R}^n} f(x), \quad f(x) = \mathbb{E}_\xi \left[ f(x, \xi) \right], \tag{1}$$

where $f(x)$ is a smooth convex function and the mathematical expectation in (1) is taken with respect to the random variable $\xi$ defined on the probability space $(\mathcal{X}, \mathcal{F}, \mathbb{P})$ with some $\sigma$-algebra $\mathcal{F}$ and probability measure $\mathbb{P}$. Such problems appear in various applications of machine learning [21, 61, 64] and mathematical statistics [66]. Perhaps, the most popular method to solve problems like (1) is Stochastic Gradient Descent (`SGD`) [26, 50, 51, 59, 63]. There is a lot of literature on the convergence in expectation of `SGD` for (strongly) convex [20, 24, 25, 46, 48, 49, 55] and non-convex [6, 20, 34] problems under different assumptions on stochastic gradient. When the problem is good enough, i.e. when the distributions of stochastic gradients are *light-tailed*, this theory correlates well with the real behavior of trajectories of `SGD` in practice. Moreover, the existing *high-probability* bounds for `SGD` [9, 11, 49] coincide with its counterpart from the theory of convergence in expectation up to logarithmic factors depending on the confidence level.

However, there are a lot of important applications where the noise distribution in the stochastic gradient is significantly *heavy-tailed* [65, 71]. For such problems `SGD` is often less robust and shows poor performance in practice. Furthermore, existing results for the convergence with high-probability for `SGD` are also much worse in the presence of heavy-tailed noise than its "light-tailed counterparts". In this case, rates of the convergence in expectation can be insufficient to describe the behavior of the method.

To illustrate this phenomenon we consider a simple example of stochastic optimization problem and apply `SGD` with constant stepsize to solve it. After that, we present a natural and simple way to resolve the issue of `SGD` based on the *clipping* of stochastic gradients. However, we need to introduce some important notations and definitions before we start to discuss this example.

---

[*]`eduard.gorbunov@phystech.edu`, `eduardgorbunov.github.io`

[†]`danilovamarina15@gmail.com`, `marinadanya.github.io`

[‡]`gasnikov@yandex.ru`

## 1.1 Preliminaries

In this section we introduce the main part of notations, assumption and definitions. The rest is classical for optimization literature and stated in the appendix (see Section A). Throughout the paper we assume that at each point $x \in \mathbb{R}^n$ function $f$ is accessible only via stochastic gradients $\nabla f(x, \xi)$ such that

$$\mathbb{E}_\xi[\nabla f(x, \xi)] = \nabla f(x), \quad \mathbb{E}_\xi \left[ \|\nabla f(x, \xi) - \nabla f(x)\|_2^2 \right] \leq \sigma^2, \tag{2}$$

i.e. we have an access to the unbiased estimator of $\nabla f(x)$ with uniformly bounded by $\sigma^2$ variance where $\sigma$ is some non-negative number. These assumptions on the stochastic gradient are standard in the stochastic optimization literature [18, 20, 31, 38, 49]. Below we introduce one of the most important definitions in this paper.

**Definition 1.1** (light-tailed random vector). We say that random vector $\eta$ has a light-tailed distribution, i.e. satisfies "light-tails" assumption, if there exist $\mathbb{E}[\eta]$ and $\mathbb{P}\left\{\|\eta - \mathbb{E}[\eta]\|_2 > b\right\} \leq 2 \exp\left(-\frac{b^2}{2\sigma^2}\right)$ for all $b > 0$

Such distributions are often called sub-Gaussian ones (see [30] and references therein). One can show (see Lemma 2 from [30]) that this definition is equivalent to

$$\mathbb{E}\left[\exp\left(\|\eta - \mathbb{E}[\eta]\|_2^2 / \sigma^2\right)\right] \leq \exp(1) \tag{3}$$

up to absolute constant difference in $\sigma$. Due to Jensen's inequality and convexity of $\exp(\cdot)$ one can easily show that inequality (3) implies $\mathbb{E}[\|\eta - \mathbb{E}[\eta]\|_2^2] \leq \sigma^2$. However, the reverse implication does not hold in general. Therefore, in the rest of the paper by stochastic gradient with heavy-tailed distribution, we mean such a stochastic gradient that satisfies (2) but not necessarily (3).

## 1.2 Simple Motivational Example: Convergence in Expectation and Clipping

In this section we consider SGD $x^{k+1} = x^k - \gamma \nabla f(x^k, \xi^k)$ applied to solve the problem (1) with $f(x, \xi) = \|x\|_2^2/2 + \langle \xi, x \rangle$, where $\xi$ is a random vector with zero mean and the variance by $\sigma^2$ (see the details in Section H.1). The state-of-the-art theory (e.g. [24, 25]) says that convergence properties in expectation of SGD in this case depend only on the stepsize $\gamma$, condition number of $f$, initial suboptimality $f(x^0) - f(x^*)$ and the variance $\sigma$, but does not depend on distribution of $\xi$. However, the trajectory of SGD significantly depends on the distribution of $\xi$. To illustrate this we consider 3 different distributions of $\xi$ with the same $\sigma$, i.e., Gaussian distribution, Weibull distribution [69] and Burr Type XII distribution [3, 42] with proper shifts and scales to get needed mean and variance for $\xi$ (see the details in Section H.1). For each distribution, we run SGD several times from the same starting point, the same stepsize $\gamma$, and the same batchsize, see typical runs in Figure 1. This simple

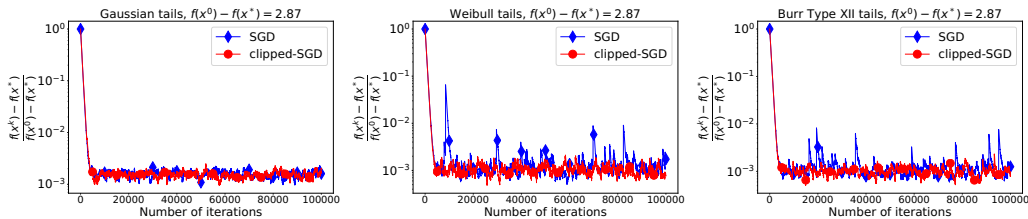

Figure 1: Typical trajectories of SGD and clipped-SGD applied to solve (130) with $\xi$ having Gaussian, Weibull, and Burr Type XII tails.

example shows that SGD in all 3 cases rapidly reaches a neighborhood of the solution and then starts to oscillate there. However, these oscillations are significantly larger for the second and the third cases where stochastic gradients are heavy-tailed. Unfortunately, guarantees for the convergence in expectation cannot express this phenomenon, since in expectation the convergence guarantees for all 3 cases are identical.

Moreover, in practice, e.g., in training big machine learning models, it is often used only a couple runs of SGD or another stochastic method. The training process can take hours or even days, so, it is extremely important to obtain good accuracy of the solution *with high probability*. However,

as our simple example shows, `SGD` fails to converge robustly if the noise in stochastic gradients is heavy-tailed which was also noticed for several real-world problems like training AlexNet [37] on CIFAR10 [36] (see [65]) and training an attention model [68] via BERT [8] (see [71]).

Clearly, since the distributions of stochastic gradients in the second and the third cases are heavy tailed the probability of sampling too large $\xi$ (in terms of the norm) and, as a consequence, too large $\nabla f(x, \xi)$ is high even if we are close to the solution. Once the current point $x^k$ is not too far from the solution and `SGD` gets a stochastic gradient with too large norm the method jumps far from the solution. Therefore, we see large oscillations. Since the reason of such oscillations is large norm of stochastic gradient it is natural to *clip* it, i.e., update $x^{k+1}$ according to $x^{k+1} = x^k - \gamma \min\{1, \lambda/\|\nabla f(x^k, \xi^k)\|_2\}\nabla f(x^k, \xi^k)$. The obtained method is known in literature as `clipped-SGD` (see [17, 21, 43, 44, 57, 70, 71] and references therein). Among the good properties of `clipped-SGD` we emphasize its robustness to the heavy-tailed noise in stochastic gradients (see also [71]). In our tests, trajectories of `clipped-SGD` oscillate not significantly even for heavy-tailed distributions, and clipping does not spoil the rate of convergence. These two factors make `clipped-SGD` preferable than `SGD` when we deal with heavy-tailed distributed stochastic gradients (see further discussion in Section B.2).

## 1.3    Related Work

### 1.3.1    Smooth Stochastic Optimization: Light-Tailed Noise

In the light-tailed case high-probability complexity bounds and complexity bounds in expectation for `SGD` and `AC-SA` differ only in logarithmical factors of $1/\beta$, see the details in Table 1. Such bounds were obtained in [9] for `SGD` in the convex case and then were extended to the $\mu$-strongly convex case in [11] for modification of `SGD` called Stochastic Intermediate Gradient Method (`SIGM`). Finally, optimal complexities were derived in [18, 19, 38] for the method called `AC-SA` in the convex case and for Multi-Staged `AC-SA` (`MS-AC-SA`) in the strongly convex case.

### 1.3.2    Smooth Stochastic Optimization: Heavy-Tailed Noise

Without light tails assumption the most straightforward results lead to $O(1/\beta^2)$ and $O(1/\beta)$ dependency on $\beta$ in the complexity bounds. Such bounds can be obtained from the complexity bounds for the convergence in expectation via Markov's inequality. However, for small $\beta$ these bounds become unacceptably poor. Classical results [13, 53, 62] reduce these dependence to $O(\ln(\beta^{-1}))$ but they have worse dependence on $\varepsilon$ than corresponding results relying on light tails assumption.

For a long time the following question was open: *is it possible to design stochastic methods having the same or comparable complexity bounds as in the light-tailed case but without light tails assumption on stochastic gradients?* In [47] and [7] the authors give a positive answer to this question *but only partially*. Let us discuss the results from these papers in detail.

In [47] Nazin et al. develop a new algorithm called Robust Stochastic Mirror Descent (`RSMD`) which is based on a special truncation of stochastic gradients and derive complexity guarantees similar to `SGD` in the convex case but without light assumption, see Table 1. This technique is very similar to gradient clipping. Moreover, in [47] authors consider also composite problems with non-smooth composite term. However, in [47] the optimization problem is defined on some *compact* convex set $X$ with diameter $\Theta = \max\{\|x - y\|_2 \mid x, y \in X\} < \infty$ and the analysis depends substantially on the boundedness of $X$. Using special restarts technique together with iterative squeezing of the set $X$ Nazin et al. extend their method to the $\mu$-strongly convex case, see Table 2. Finally, in the discussion section of [47] authors formulate the following question: *is it possible to develop such **accelerated** stochastic methods that have the same or comparable complexity bounds as in the light-tailed case but do not require stochastic gradients to be light-tailed?*

In the strongly convex case the positive answer to this question was given by Davis et al. [7] where authors propose a new method called `proxBoost` that is based on robust distance estimation [29, 51] and proximal point method [40, 41, 60], see Table 2. However, this approach requires solving an auxiliary optimization problem at each iteration that can lead to poor performance in practice.

In our paper we close the gap in theory, i.e., we provide a positive answer to the following question: *Is it possible to develop such an accelerated stochastic method that have the same or comparable*

*complexity bound as for* `AC-SA` *in the convex case but do not require stochastic gradients to be light-tailed?*

### 1.4 Our Contributions

- One of the main contributions of our paper is a new method called Clipped Stochastic Similar Triangles Method (`clipped-SSTM`). For the case when the objective function $f$ is convex and $L$-smooth we derive the following complexity bound *without light tails assumption on the stochastic gradients:* $O(\max\{\sqrt{LR_0^2/\varepsilon}, \sigma^2 R_0^2/\varepsilon^2\} \ln(LR_0^2/\varepsilon\beta))$. This bound outperforms all known bounds for this setting (see Table 1) and up to the difference in logarithmical factors recovers the complexity bound of `AC-SA` derived under light tails assumption. That is, in this paper we close the gap in theory theory of smooth convex stochastic optimization with heavy-tailed noise. Moreover, unlike in [47], we do not assume boundedness of the set where the optimization problem is defined, which makes our analysis more complicated. We also study different batchsize policies for `clipped-SSTM`.

- Using restarts technique we extend `clipped-SSTM` to the $\mu$-strongly convex objectives and obtain a new method called Restarted `clipped-SSTM` (`R-clipped-SSTM`). For this method we prove the following complexity bound (again, *without light tails assumption on the stochastic gradients*): $O(\max\{\sqrt{L/\mu}\ln(\mu R^2/\varepsilon), \sigma^2/\mu\varepsilon\} \ln(L/\mu\beta \ln(\mu R^2/\varepsilon)))$. Our bound outperforms the state-of-the-art result from [7] in terms of the dependence on $\ln\frac{L}{\mu}$, see Table 2 for the details.

- We prove the first high-probability complexity guarantees for `clipped-SGD` in convex and strongly convex cases *without light tails assumption on the stochastic gradients*, see Tables 1 and 2. The complexity we prove for `clipped-SGD` in the convex case is comparable with corresponding bound for `SGD` derived under light tails assumption. In the $\mu$-strongly convex case we derive a new complexity bound for the restarted version of `clipped-SGD` (`R-clipped-SGD`) which is comparable with its "light-tailed counterpart".

- We conduct several numerical experiments with the proposed methods in order to justify the theory we develop. In particular, we show that `clipped-SSTM` can outperform `SGD` and `clipped-SGD` in practice even without using large batchsizes. Moreover, in our experiments we illustrate how clipping makes the convergence of `SGD` and `SSTM` more robust and reduces their oscillations.

Table 1: Comparison of existing high-probability convergence results for stochastic optimization under assumptions (2) for convex and $L$-smooth objectives. The second column contains an overall number of stochastic first-order oracle calls needed to achieve $\varepsilon$-solution with probability at least $1 - \beta$. In the third column "light" means that $\nabla f(x, \xi)$ satisfies (3) and "heavy" means that the result holds even in the case when (3) does not hold. Column "Domain" describes the set where the optimization problem is defined. For `RSMD` $\Theta$ is a diameter of the set where the optimization problem is defined. We use red color to emphasize the restrictions we eliminate.

| Method | Complexity | Tails | Domain |
|--------|-----------|-------|--------|
| `SGD` [9] | $O\left(\max\left\{\frac{LR_0^2}{\varepsilon}, \frac{\sigma^2 R_0^2}{\varepsilon^2}\ln^2(\beta^{-1})\right\}\right)$ | light | bounded |
| `AC-SA` [18, 38] | $O\left(\max\left\{\sqrt{\frac{LR_0^2}{\varepsilon}}, \frac{\sigma^2 R_0^2}{\varepsilon^2}\ln(\beta^{-1})\right\}\right)$ | light | arbitrary |
| `RSMD` [47] | $O\left(\max\left\{\frac{L\Theta^2}{\varepsilon}, \frac{\sigma^2\Theta^2}{\varepsilon^2}\right\}\ln(\beta^{-1})\right)$ | heavy | bounded |
| `clipped-SGD` [This work] | $O\left(\max\left\{\frac{LR_0^2}{\varepsilon}, \frac{\sigma^2 R_0^2}{\varepsilon^2}\right\}\ln(\beta^{-1})\right)$ | heavy | $\mathbb{R}^n$ |
| `clipped-SSTM` [This work] | $O\left(\max\left\{\sqrt{\frac{LR_0^2}{\varepsilon}}, \frac{\sigma^2 R_0^2}{\varepsilon^2}\right\}\ln\frac{LR_0^2+\sigma R_0}{\varepsilon\beta}\right)$ | heavy | $\mathbb{R}^n$ |

### 1.4.1 Relation to [71]

While Zhang et al. [71] consider different setup, [71] is highly relevant to our paper, and, in some sense, it complements our findings. In particular, it contains the analysis of several versions of

Table 2: Comparison of existing high-probability convergence results for stochastic optimization under assumptions (2) for $\mu$-strongly convex and $L$-smooth objectives. The second column contains an overall number of stochastic first-order oracle calls needed to achieve $\varepsilon$-solution with probability at least $1 - \beta$. In the third column "light" means that $\nabla f(x, \xi)$ satisfies (3) and "heavy" means that the result holds even in the case when (3) does not hold. Column "Domain" describes the set where the optimization problem is defined. For RSMD $\Theta$ is a diameter of the set where the optimization problem is defined and $R = \sqrt{2(f(x^0) - f(x^*))/\mu}$, $r_0 = f(x^0) - f(x^*)$. We use red color to emphasize the restrictions we eliminate.

| Method | Complexity | Tails | Domain |
|---|---|---|---|
| SIGM [11] | $O\left(\max\left\{\frac{L}{\mu}\ln\frac{\mu R_0^2}{\varepsilon}, \frac{\sigma^2}{\mu\varepsilon}\ln\left(\beta^{-1}\ln\frac{\mu R_0^2}{\varepsilon}\right)\right\}\right)$ | light | arbitrary |
| MS-AC-SA [19] | $O\left(\max\left\{\sqrt{\frac{L}{\mu}}\ln\frac{LR_0^2}{\varepsilon}, \frac{\sigma^2}{\mu\varepsilon}\ln\left(\beta^{-1}\ln\frac{LR_0^2}{\varepsilon}\right)\right\}\right)$ | light | arbitrary |
| restarted-RSMD [47] | $O\left(\max\left\{\frac{L}{\mu}\ln\left(\frac{\mu\Theta^2}{\varepsilon}\right), \frac{\sigma^2}{\mu\varepsilon}\right\}\ln\left(\beta^{-1}\ln\frac{\mu\Theta^2}{\varepsilon}\right)\right)$ | heavy | bounded |
| proxBoost [7] | $O\left(\max\left\{\sqrt{\frac{L}{\mu}}\ln\left(\frac{LR_0^2\ln\frac{L}{\mu}}{\varepsilon}\right), \frac{\sigma^2\ln\frac{L}{\mu}}{\mu\varepsilon}\right\}\cdot C\right)$, where $C = \ln\left(\frac{L}{\mu}\right)\ln\left(\frac{\ln\frac{L}{\mu}}{\beta}\right)$ | heavy | arbitrary |
| clipped-SGD [This work] | $O\left(\max\left\{\frac{L}{\mu}, \frac{\sigma^2}{\mu\varepsilon}\cdot\frac{L}{\mu}\right\}\ln\left(\frac{r_0}{\varepsilon}\right)\ln\left(\frac{L}{\mu\beta}\ln\frac{r_0}{\varepsilon}\right)\right)$ | heavy | $\mathbb{R}^n$ |
| R-clipped-SGD [This work] | $O\left(\max\left\{\frac{L}{\mu}\ln\frac{\mu R^2}{\varepsilon}, \frac{\sigma^2}{\mu\varepsilon}\right\}\ln\left(\frac{L}{\mu\beta}\ln\frac{\mu R^2}{\varepsilon}\right)\right)$ | heavy | $\mathbb{R}^n$ |
| R-clipped-SSTM [This work] | $O\left(\max\left\{\sqrt{\frac{L}{\mu}}\ln\frac{\mu R^2}{\varepsilon}, \frac{\sigma^2}{\mu\varepsilon}\right\}\ln\left(\frac{L}{\mu\beta}\ln\frac{\mu R^2}{\varepsilon}\right)\right)$ | heavy | $\mathbb{R}^n$ |

clipped-SGD establishing the rates of convergence *in expectation* while we focus on the *high-probability* complexity guarantees. Secondly, we consider convex and strongly convex cases while [71] provides an analysis for non-convex and strongly convex problems. Finally, [71] relies on the following assumption: there exist such $G > 0$ and $\alpha \in (1, 2]$ that the stochastic gradient $g(x)$ satisfies $\mathbb{E}\|g(x)\|_2^\alpha \leq G^\alpha$. This assumption implies the boundedness of the gradient of the objective function $f(x)$ which is quite restrictive and does not hold on the whole space for strongly convex functions. In our paper, we assume only boundedness of the variance. Moreover, we consider *smooth* problems that allows us to accelerate clipped-SGD and obtain clipped-SSTM, while Zhang et al. [71] provide non-accelerated rates.

### 1.5 Paper Organization

The remaining part of the paper is organized as follows. In Section 2 we present clipped-SSTM together with the main complexity result in the convex case that we prove for this method. Then, we present the first high-probability complexity bounds for clipped-SGD for for the convex problems. In Section 4 we provide our numerical experiments justifying our theoretical results. Finally, in Section 5 we provide some concluding remarks and discuss the limitations and possible extensions of the results developed in the paper. Due to the space limitations, we put the exact formulations of all theorems, results for the strongly convex problems and the full proofs in the Appendix (see Sections F and G), together with auxiliary and technical results and additional experiments (see Section H). Moreover, in Section F.1.2 we present a sketch of the proof of the main convergence result for clipped-SSTM and explain the intuition behind it.

## 2 Accelerated SGD with Clipping

In this section we consider the situation when $f(x)$ is convex and $L$-smooth on $\mathbb{R}^n$. For this problem we present a new method called Clipped Stochastic Similar Triangles Method (clipped-SSTM, see Algorithm 1). In our method we use a clipped stochastic gradient that is defined in the following way:

$$\text{clip}(\nabla f(x, \boldsymbol{\xi}), \lambda) = \min\left\{1, \lambda/\|\nabla f(x, \boldsymbol{\xi})\|_2\right\}\nabla f(x, \boldsymbol{\xi}) \tag{4}$$

---

**Algorithm 1** Clipped Stochastic Similar Triangles Method (`clipped-SSTM`)

---

**Input:** starting point $x^0$, number of iterations $N$, batchsizes $\{m_k\}_{k=1}^N$, stepsize parameter $a$, clipping parameter $B$

1: Set $A_0 = \alpha_0 = 0$, $y^0 = z^0 = x^0$
2: **for** $k = 0, \ldots, N-1$ **do**
3:      Set $\alpha_{k+1} = \frac{k+2}{2aL}$, $A_{k+1} = A_k + \alpha_{k+1}$, $\lambda_{k+1} = \frac{B}{\alpha_{k+1}}$
4:      $x^{k+1} = (A_k y^k + \alpha_{k+1} z^k)/A_{k+1}$
5:      Draw fresh i.i.d. samples $\xi_1^k, \ldots, \xi_{m_k}^k$ and compute $\nabla f(x^{k+1}, \boldsymbol{\xi}^k) = \frac{1}{m_k} \sum_{i=1}^{m_k} \nabla f(x^{k+1}, \xi_i^k)$
6:      Compute $\widetilde{\nabla} f(x^{k+1}, \boldsymbol{\xi}^k) = \text{clip}(\nabla f(x^{k+1}, \boldsymbol{\xi}^k), \lambda_{k+1})$ using (4)
7:      $z^{k+1} = z^k - \alpha_{k+1} \widetilde{\nabla} f(x^{k+1}, \boldsymbol{\xi}^k)$
8:      $y^{k+1} = (A_k y^k + \alpha_{k+1} z^{k+1})/A_{k+1}$
9: **end for**
**Output:** $y^N$

---

where $\nabla f(x, \boldsymbol{\xi}) = \frac{1}{m} \sum_{i=1}^m \nabla f(x, \xi_i)$ is a mini-batched version of $\nabla f(x)$. That is, in order to compute $\text{clip}(\nabla f(x, \boldsymbol{\xi}), \lambda)$ one needs to get $m$ i.i.d. samples $\nabla f(x, \xi_1), \ldots, \nabla f(x, \xi_m)$, compute its average and then project the result $\nabla f(x, \boldsymbol{\xi})$ on the Euclidean ball with radius $\lambda$ and center at the origin. Next theorem summarizes the main convergence result for `clipped-SSTM`.

**Theorem 2.1.** Assume that function $f$ is convex and $L$-smooth. Then for all $\beta \in (0,1)$ and $N \geq 1$ such that $\ln(4N/\beta) \geq 2$ we have that after $N$ iterations of `clipped-SSTM` with $m_k = \Theta\left(\max\left\{1, \sigma^2 \alpha_{k+1}^2 N \ln(N/\beta)/R_0^2\right\}\right)$, $B = \Theta(R_0/\ln(N/\beta))$ and $a = \Theta(\ln^2(N/\beta))$ that $f(y^N) - f(x^*) = O(aLR_0^2/N^2)$ holds with probability at least $1 - \beta$ where $R_0 = \|x^0 - x^*\|_2$. In other words, if we choose $a$ to be equal to the maximum from (27), then the method achieves $f(y^N) - f(x^*) \leq \varepsilon$ with probability at least $1 - \beta$ after $O(\sqrt{LR_0^2/\varepsilon} \ln(LR_0^2/\varepsilon\beta))$ iterations and requires $O(\max\{\sqrt{LR_0^2/\varepsilon}, \sigma^2 R_0^2/\varepsilon^2\} \ln(LR_0^2/\varepsilon\beta))$ oracle calls.

The theorem says that for any $\beta \in (0,1)$ `clipped-SSTM` converges to $\varepsilon$-solution with probability at least $1 - \beta$ and requires exactly the same number of stochastic first-order oracle calls (up to the difference in constant and logarithmical factors) as optimal stochastic methods like `AC-SA` [18, 38] or Stochastic Similar Triangles Method [16, 22]. However, our method *achieves this rate under less restrictive assumption*. Indeed, Theorem 2.1 holds even in the case when the stochastic gradient $\nabla f(x, \xi)$ satisfies only (2) and can have *heavy-tailed* distribution. In contrast, all existing results that establish (30) and that are known in the literature hold only in the light-tails case, see Section 1.3.1.

Finally, when $\sigma^2$ is big then Theorem 2.1 says that at iteration $k$ `clipped-SGD` requires large batchsizes $m_k \sim k^2 N$ (see (26)) which is proportional to $\varepsilon^{-3/2}$ for last iterates. It can make the cost of one iteration extremely high, therefore, we also consider different stepsize policies that remove this drawback in Section F.1.1. In particular, the following result shows that `clipped-SSTM` achieves the same oracle complexity even with constant batchsizes $m_k$ when stepsize parameter $a$ is chosen properly.

**Corollary 2.2.** Let the assumptions of Theorem F.1 hold and $a = \Theta\left(\max\{1, \ln^2(N/\beta), \sqrt{\ln N/\beta} \sigma N^{3/2}/LR_0\}\right)$. Then $m_k = O(1)$ and `clipped-SSTM` achieves $f(y^N) - f(x^*) \leq \varepsilon$ with probability at least $1 - \beta$ after $O(\max\{\sqrt{LR_0^2/\varepsilon}, \sigma^2 R_0^2/\varepsilon^2\} \ln((LR_0^2 + \sigma R_0)/\varepsilon\beta))$ iterations/oracle calls.

## 3 `SGD` with Clipping

In this section we present our complexity results for `clipped-SGD` (see Algorithm 2) in the convex case. Next theorem summarizes the main convergence result for `clipped-SGD` in this case.

**Algorithm 2** Clipped Stochastic Gradient Descent (`clipped-SGD`)

---

**Input:** starting point $x^0$, number of iterations $N$, batchsizes $\{m_k\}_{k=0}^{N-1}$, stepsize $\gamma > 0$, clipping level $\lambda > 0$
1: **for** $k = 0, \ldots, N-1$ **do**
2:      Draw fresh i.i.d. samples $\xi_1^k, \ldots, \xi_{m_k}^k$ and compute $\nabla f(x^k, \boldsymbol{\xi}^k) = \frac{1}{m_k} \sum_{i=1}^{m_k} \nabla f(x^k, \xi_i^k)$
3:      Compute $\widetilde{\nabla} f(x^k, \boldsymbol{\xi}^k) = \mathrm{clip}(\nabla f(x^k, \boldsymbol{\xi}^k), \lambda)$ using (4)
4:      $x^{k+1} = x^k - \gamma \widetilde{\nabla} f(x^k, \boldsymbol{\xi}^k)$
5: **end for**
**Output:** $\bar{x}^N = \frac{1}{N} \sum_{k=0}^{N-1} x^k$

---

**Theorem 3.1.** Assume that function $f$ is convex and $L$-smooth. Then for all $\beta \in (0, 1)$ and $N \geq 1$ such that $\ln(4N/\beta) \geq 2$ we have that after $N$ iterations of `clipped-SGD` with $\lambda = \Theta(LR_0)$ and $m_k = m = \Theta(\max\{1, N\sigma^2/R_0^2 L^2 \ln(N/\beta)\})$ where $R_0 = \|x^0 - x^*\|_2$ and stepsize $\gamma = 1/80L\ln(4N/\beta)$ that $f(\bar{x}^N) - f(x^*) = O(LR_0^2 \ln(4N/\beta)/N)$ with probability at least $1 - \beta$ where $\bar{x}^N = \frac{1}{N} \sum_{k=0}^{N-1} x^k$. In other words, the method achieves $f(\bar{x}^N) - f(x^*) \leq \varepsilon$ with probability at least $1 - \beta$ after $O\left(LR_0^2/\varepsilon \ln(LR_0^2/\varepsilon\beta)\right)$ iterations and requires $O(\max\{LR_0^2/\varepsilon, \sigma^2 R_0^2/\varepsilon^2\} \ln(LR_0^2/\varepsilon\beta))$ oracle calls.

To the best of our knowledge, it is the first result for `clipped-SGD` establishing non-trivial complexity guarantees for the convergence with high probability. Up to the difference in logarithmical factors our bound recovers the complexity bound for `SGD` which was obtained under light tails assumption and the complexity bound for `RSMD`. However, unlike in [47], we do not assume that the optimization problem is defined on the bounded set. The proof technique is similar to one we use to prove Theorem F.1. One can find the full proof in Section G.3.1.

## 4 Numerical Experiments

We have tested[4] `clipped-SSTM` and `clipped-SGD` on the logistic regression problem, the datasets were taken from LIBSVM library [4]. To implement methods we use Python 3.7 and standard libraries. One can find additional experiments and details in Section H.2.

First of all, using standard solvers from `scipy` library we find good enough approximation of the solution of the problem for each dataset. For simplicity, we denote this approximation by $x^*$. Then, we numerically study the distribution of $\|\nabla f_i(x^*)\|_2$ and plot corresponding histograms for each dataset, see Figure 2. These histograms hint that near the solution for `heart` dataset tails of stochastic

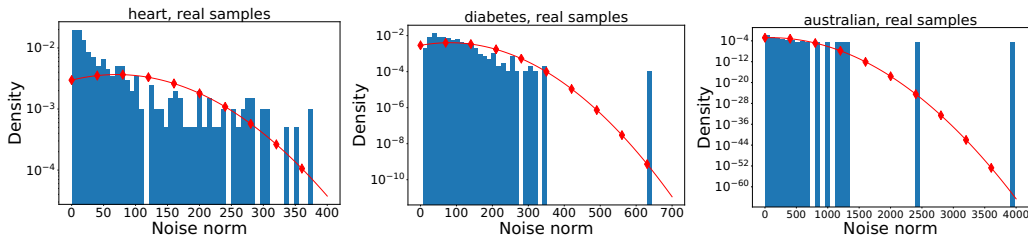

Figure 2: Histograms of $\|\nabla f_i(x^*)\|_2$ for different datasets. Red lines correspond to probability density functions of normal distributions with empirically estimated means and variances.

gradients are not heavy and the norm of the noise can be well-approximated by Gaussian distribution, whereas for `diabetes` and `australian` we see the presence of outliers that makes the distribution heavy-tailed.

Next, let us consider numerical results for `SGD` and `SSTM` with and without clipping applied to solve logistic regression problem on these 3 datasets, see Figures 3- 5. For all methods we used constant batchsizes $m$, stepsizes and clipping levels were tuned, see Section H.2 for the details. In our experiments we also consider `clipped-SGD` with periodically decreasing clipping level $\lambda$

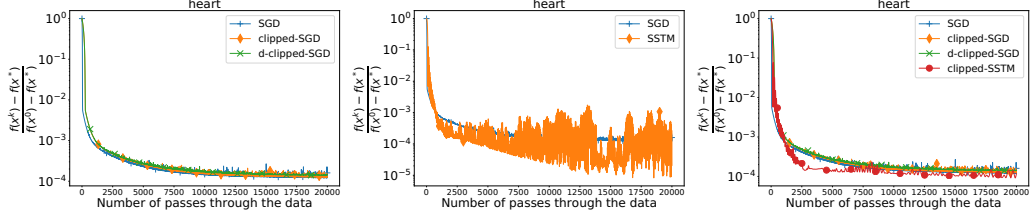

Figure 3: Trajectories of SGD, clipped-SGD, SSTM and clipped-SSTM applied to solve logistic regression problem on heart dataset.

(d-clipped-SGD in Figures), i.e. the method starts with some initial clipping level $\lambda_0$ and after every $l$ epochs or, equivalently, after every $\lceil rl/m \rceil$ iterations the clipping level is multiplied by some constant $\alpha \in (0, 1)$.

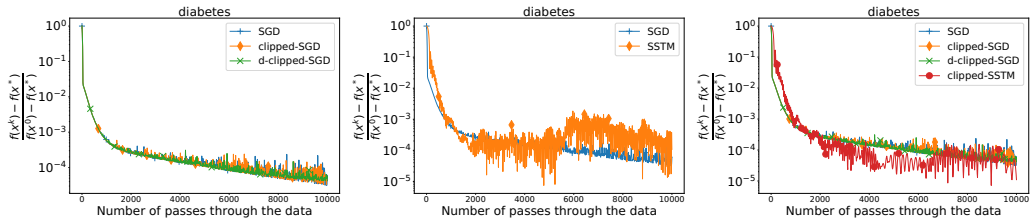

Figure 4: Trajectories of SGD, clipped-SGD, SSTM and clipped-SSTM applied to solve logistic regression problem on diabetes dataset.

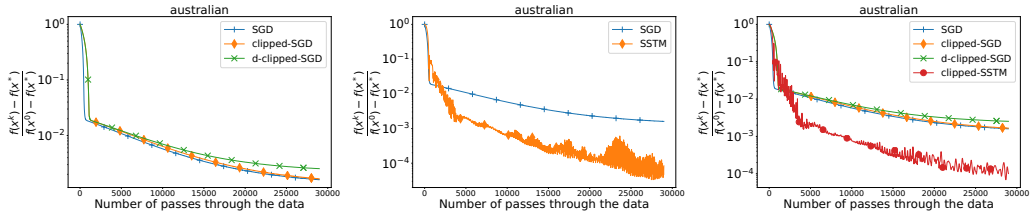

Figure 5: Trajectories of SGD, clipped-SGD, SSTM and clipped-SSTM applied to solve logistic regression problem on australian dataset.

Let us discuss the obtained numerical results. First of all, d-clipped-SGD stabilizes the oscillations of SGD even if the initial clipping level was high. In contrast, clipped-SGD with too large clipping level $\lambda$ behaves similarly to SGD. Secondly, we emphasize that due to the fact that we used small bathcsizes SSTM has very large oscillations in comparison to SGD. Actually, fast error/noise accumulation is a typical drawback of accelerated SGD with small batchsizes [35]. Moreover, deterministic accelerated and momentum-based methods often have non-monotone behavior (see [5] and references therein). However, to some extent clipped-SSTM suffers from the first drawback less than SSTM and has comparable convergence rate with SSTM. Finally, in our experiments on heart and australian datasets clipped-SSTM converges faster than SGD and clipped-SGD and oscillates little, while on diabetes dataset it also converges faster than SGD, but oscillates more if parameter $B$ is not fine-tuned.

We also want to mention that the behavior of SGD on heart and diabetes datasets correlates with the insights from Section 1.2 and our numerical study of the distribution of $\|\nabla f_i(x^*)\|_2$. Indeed, for heart dataset SGD has little oscillations since the distribution of $\|\nabla f_i(x^k) - \nabla f(x^k)\|_2$, where $x^k$ is the last iterate, is well concentrated near its mean and can be approximated by Gaussian distribution (see the details in Section H.2). In contrast, Figure 4 shows that SGD oscillates more than in the previous example. One can explain such behavior using Figure 2 showing that the distribution of $\|\nabla f(x^*)\|_2$ has heavier tails than for heart dataset.

However, we do not see any oscillations of `SGD` for `australian` dataset despite the fact that according to Figure 2 the distribution of $\|\nabla f_i(x^*)\|_2$ in this case has heavier tails than in previous examples. Actually, there is no contradiction and in this case it simply means that `SGD` does not get close to the solution in terms of functional value, despite the fact that we used $\gamma = 1/L$. In Section H.2 we present the results of different tests where we tried to use bigger stepsize $\gamma$ in order to reach oscillation region faster and show that in fact in that region `SGD` oscillates significantly more, but clipping fixes this issue without spoiling the convergence rate.

## 5 Discussion

In this paper we close the gap in the theory of high-probability complexity bounds for stochastic optimization with heavy-tailed noise. In particular, we propose a new accelerated stochastic method — `clipped-SSTM` — and prove the first accelerated high-probability complexity bounds for smooth convex stochastic optimization without light-tails assumption. Moreover, we extend our results to the strongly convex case and prove new complexity bounds outperforming the state-of-the-art results. Finally, we derive first high-probabiliy complexity bounds for the popular method called `clipped-SGD` in convex and strongly convex cases and conduct a numerical study of the considered methods.

However, our approach has several limitations. In particular, it significantly relies on the assumption that the optimization problem is defined on $\mathbb{R}^n$. Moreover, we do not consider regularized or composite problems like in [47] and [7]. However, in [47] it is significant in the analysis that the set where the problem is defined is bounded and in [7] the analysis works only for the strongly convex problems. It would also be interesting to generalize our approach to generally non-smooth problems using the trick from [52].

## Broader Impact

Our contribution is primarily theoretical. Therefore, a broader impact discussion is not applicable.

## Acknowledgments and Disclosure of Funding

The research of E. Gorbunov and A. Gasnikov was partially supported by the Ministry of Science and Higher Education of the Russian Federation (Goszadaniye) 075-00337-20-03, project No. 0714-2020-0005. The research of Marina Danilova was funded by RFBR, project number 20-31-90073.

## Footnotes

[4]One can find the code here: `https://github.com/eduardgorbunov/accelerated_clipping`.

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
