[Supplementary Material]

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

## Contents

## A    Notations and Definitions

We use $\langle x, y \rangle$ to define standard inner product between two vectors $x, y \in \mathbb{R}^n$, i.e. $\langle x, y \rangle \stackrel{\text{def}}{=} \sum_{i=1}^{n} x_i y_i$, where $x_i$ is $i$-th coordinate of vector $x$, $i = 1, \ldots, n$. Standard Euclidean norm of vector $x \in \mathbb{R}^n$ is defined as $\|x\|_2 \stackrel{\text{def}}{=} \sqrt{\langle x, x \rangle}$.

We use $\mathbb{P}\{\cdot\}$ to define probability measure which is always known from the context, $\mathbb{E}[\cdot]$ denotes mathematical expectation, $\mathbb{E}_\xi[\cdot]$ is used to define conditional mathematical expectation with respect to the randomness coming from $\xi$ only and $\mathbb{E}[\xi \mid \eta]$ denotes mathematical expectation of $\xi$ conditional on $\eta$. In our proofs, we also use $\mathbb{E}_k[\cdot]$ to denote conditional mathematical expectation with respect to all randomness coming from $k$-th iteration. For $\mathbb{P}$-measurable set $X$ we use $\mathbb{1}_X$ to denote indicator of event $X$, i.e.

$$\mathbb{1}_X = \begin{cases} 1, & \text{if event } X \text{ holds,} \\ 0, & \text{otherwise.} \end{cases} \tag{5}$$

Next, we introduce some standard definitions.

**Definition A.1** (*L*-smoothness). Function $f$ is called *L*-smooth on $\mathbb{R}^n$ with $L > 0$ when it is differentiable and its gradient is *L*-Lipschitz continuous on $\mathbb{R}^n$, i.e.

$$\|\nabla f(x) - \nabla f(y)\|_2 \le L\|x - y\|_2, \quad \forall x, y \in \mathbb{R}^n. \tag{6}$$

It is well-known that *L*-smoothness implies (see [54])

$$f(y) \quad \le \quad f(x) + \langle \nabla f(x), y - x \rangle + \frac{L}{2}\|y - x\|_2^2 \quad \forall x, y \in \mathbb{R}^n, \tag{7}$$

and if $f$ is additionally convex, then

$$\|\nabla f(x) - \nabla f(y)\|_2^2 \quad \le \quad 2L\left(f(x) - f(y) - \langle \nabla f(y), x - y \rangle\right) \quad \forall x, y \in \mathbb{R}^n. \tag{8}$$

Since in this paper we focus only on smooth optimization problems we introduce strong convexity in the following way.

**Definition A.2** ($\mu$-strong convexity). Differentiable function $f$ is called $\mu$-strongly convex on $\mathbb{R}^n$ with $\mu \geq 0$ if for all $x, y \in \mathbb{R}^n$

$$f(x) \geq f(y) + \langle \nabla f(y), x - y \rangle + \frac{\mu}{2}\|x - y\|_2^2. \tag{9}$$

In particular, $\mu$-strong convexity implies that for all $x \in \mathbb{R}^n$

$$f(x) - f(x^*) \geq \frac{\mu}{2}\|x - x^*\|_2^2. \tag{10}$$

Throughout the paper, we use $x^*$ to denote any solution of problem (1) assuming its existence. By the complexity of stochastic first-order method we always mean the total number of stochastic first-order oracle calls that the method needs in order to produce such a point $\hat{x}$ that $f(\hat{x}) - f(x^*) \leq \varepsilon$ with probability at least $1 - \beta$ for some $\varepsilon > 0$ and $\beta \in (0,1)$. Finally, in the complexity bounds we often use $R_0$ to denote $\|x^0 - x^*\|_2$ where $x^0$ is the starting point of the method.

# B  Related Work: Additional Details

## B.1  Related Work on Non-Smooth Stochastic Optimization

Here we present an overview of existing results in the convex non-smooth case, i.e. when $f$ is still convex but not necessarily $L$-smooth and the stochastic gradients have a bounded second moment: $\mathbb{E}_\xi[\|\nabla f(x,\xi)\|_2^2] \leq M^2$ for all $x \in \mathbb{R}^n$. Under additional assumption that the stochastic gradients have light-tailed distribution it was shown that SGD [49] has $O\left(M^2 R_0^2 \ln(\beta^{-1})/\varepsilon^2\right)$ complexity and if additionally $f$ is $\mu$-strongly convex it was shown in [31, 32] that the restarted version of SGD has $O\left(M^2 \ln\left(\beta^{-1} \ln(M^2 \mu^{-1} \varepsilon^{-1})\right)/\mu\varepsilon\right)$ complexity (see also [27, 33, 58]). Moreover, removing logarithmical factors from these bounds we get the complexity bounds of these methods for the convergence in expectation, i.e. needed number of oracle calls to find such $\hat{x}$ that $\mathbb{E}[f(\hat{x})] - f(x^*) \leq \varepsilon$. That is, under light tails assumption high-probability complexity bounds and complexity bounds in expectation for SGD and restarted-SGD differ only in logarithmical factors of $1/\beta$.

Unfortunately, for these methods the situation changes dramatically when the stochastic gradients are heavy-tailed. To the best of our knowledge, the best know bounds in the literature with the same dependency on $\varepsilon$ are $O\left(M^2 R_0^2/\beta^2\varepsilon^2\right)$ and $O\left(M^2/\mu\beta\varepsilon\right)$. One can obtain these bounds using complexity results for the convergence in expectation and Markov's inequality. However, it leads to significantly worse dependence on $\beta$: instead of $O(\ln(\beta^{-1}))$ we get $O(\beta^{-2})$ and $O(\beta^{-1})$ dependence on the confidence level $\beta$. Furthermore, based on the well-known results on the distribution of sum of i.i.d. random variables (see Section D.2) in [15] authors consider the case when the tails of the distribution of stochastic gradient satisfy $\mathbb{P}\{\|\nabla f(x,\xi) - \nabla f(x)\|_2 > s\} = O(s^{-\alpha})$ for $\alpha > 2$ and give the following complexity bounds without formal proofs that SGD for convex problems and restarted-SGD for $\mu$-strongly convex problems have following complexities:

$$O\left(M^2 R^2 \max\left\{\frac{\ln\left(\beta^{-1}\right)}{\varepsilon^2}, \left(\frac{1}{\beta\varepsilon^\alpha}\right)^{\frac{2}{3\alpha-2}}\right\}\right),$$

$$O\left(\max\left\{\frac{M^2 \ln\left(\beta^{-1} \ln \frac{M^2}{\mu\varepsilon}\right)}{\mu\varepsilon}, \left(\frac{M^2}{\mu\varepsilon}\right)^{\frac{\alpha}{3\alpha-2}} \left(\beta^{-1} \ln \frac{M^2}{\mu\varepsilon}\right)^{\frac{2}{3\alpha-2}}\right\}\right).$$

The first terms in maximums above correspond to the Central Limit Theorem regime, while the second terms correspond to the heavy-tailed regime, see Section D.2. These bounds show that heavy tailed distributions of the stochastic gradients significantly spoil complexity bounds of SGD and restarted-SGD when the confidence level $\beta$ is small enough.

## B.2  Related Work on Gradient Clipping

As we mentioned Section 1.2 clipped-SGD [21, 45, 56, 67] is known to be robust to the noise in stochastic gradients and performs better than SGD in the vicinity of extremely steep cliffs. Zhang et al. [71] analyse the convergence of clipped-SGD *in expectation* for strongly convex and non-convex objectives under assumption that $\mathbb{E}[\|\nabla f(x,\xi)\|_2^\alpha]$ is bounded for some $\alpha \in (1,2]$. For $\alpha < 2$ this assumption covers some heavy-tailed distributions of stochastic gradients appearing in practice. Moreover, in [71] authors conduct several numerical tests showing that in some real-world problems where the noise in stochastic gradients is heavy-tailed clipped-SGD converges faster than SGD. In [70] Zhang et al. found that clipped-GD is able to converge in non-convex case to the stationary point under the relaxed smoothness assumption with $O(\varepsilon^{-2})$ rate while Gradient Descent (GD) can fail to converge with the same rate in this setting. A very similar approach based on the normalization of GD is studied in [28, 39].

## C  Basic Facts

In this section we enumerate for convenience basic facts that we use many times in our proofs.

**Fenchel-Young inequality.** For all $a, b \in \mathbb{R}^n$ and $\lambda > 0$

$$|\langle a, b \rangle| \leq \frac{\|a\|_2^2}{2\lambda} + \frac{\lambda\|b\|_2^2}{2}. \tag{11}$$

**Squared norm of the sum.** For all $a, b \in \mathbb{R}^n$

$$\|a + b\|_2^2 \leq 2\|a\|_2^2 + 2\|b\|_2^2. \tag{12}$$

**Inner product representation.** For all $a, b \in \mathbb{R}^n$

$$\langle a, b \rangle = \frac{1}{2}\left(\|a + b\|_2^2 - \|a\|_2^2 - \|b\|_2^2\right) \tag{13}$$

**Variance decomposition.** If $\xi$ is a random vector in $\mathbb{R}^n$ with bounded second moment, then

$$\mathbb{E}\left[\|\xi + a\|_2^2\right] = \mathbb{E}\left[\|\xi - \mathbb{E}[\xi]\|_2^2\right] + \|\mathbb{E}[\xi] + a\|_2^2 \tag{14}$$

for any deterministic vector $a \in \mathbb{R}^n$. In particular, this implies

$$\mathbb{E}\left[\|\xi - \mathbb{E}[\xi]\|_2^2\right] \leq \mathbb{E}\left[\|\xi + a\|_2^2\right] \tag{15}$$

for any deterministic vector $a \in \mathbb{R}^n$.

## D  Auxiliary Results

### D.1  Bernstein Inequality

**Lemma D.1** (Bernstein inequality for martingale differences [1, 12, 14])**.** Let the sequence of random variables $\{X_i\}_{i \geq 1}$ form a martingale difference sequence, i.e. $\mathbb{E}[X_i \mid X_{i-1}, \ldots, X_1] = 0$ for all $i \geq 1$. Assume that conditional variances $\sigma_i^2 \stackrel{\text{def}}{=} \mathbb{E}[X_i^2 \mid X_{i-1}, \ldots, X_1]$ exist and are bounded and assume also that there exists deterministic constant $c > 0$ such that $\|X_i\|_2 \leq c$ almost surely for all $i \geq 1$. Then for all $b > 0$, $F > 0$ and $n \geq 1$

$$\mathbb{P}\left\{\left|\sum_{i=1}^n X_i\right| > b \text{ and } \sum_{i=1}^n \sigma_i^2 \leq F\right\} \leq 2\exp\left(-\frac{b^2}{2F + 2cb/3}\right). \tag{16}$$

### D.2  About the Sum of i.i.d. Random Variables with Heavy Tails

In this section we present some classical results about the distribution of sum of i.i.d. random variables $\sum_{k=1}^N \xi_k$ with heavy tails [2]. As one can see from our proofs of main results for `clipped-SSTM` and `clipped-SGD` such sums play a central role in the analysis of convergence with high probability. Assume that $\{\xi_k\}$ is i.i.d. with $\mathbb{E}[\xi_k] = 0$ and $\text{Var}[\xi_k] \stackrel{\text{def}}{=} \mathbb{E}[(\xi_k - \mathbb{E}[\xi_k])^2] = \sigma^2$. Assume also that $V(s) = \mathbb{P}\{\xi_k \geq s\} = \Theta(s^{-\alpha})$, where $\alpha > 2$. In this case

$$\mathbb{P}\left\{\sum_{k=1}^N \xi_k \geq s\right\} \simeq 1 - \Phi\left(\frac{s}{\sqrt{\sigma^2 N}}\right) + N \cdot V(s),$$

where $N \gg 1$ and $\Phi(x) = \frac{1}{2\pi}\int_{-\infty}^x \exp\left(-y^2/2\right) dy$. Since

$$0.2\exp\left(-\frac{2x^2}{\pi}\right) \leq 1 - \Phi(x) \leq \exp\left(-\frac{x^2}{2}\right),$$

we have[5]

$$\mathbb{P}\left\{\sum_{k=1}^N \xi_k \geq s\right\} \simeq 1 - \Phi\left(\frac{s}{\sqrt{\sigma^2 N}}\right), \quad s \leq \sqrt{(\alpha - 2)\sigma^2 N \ln N} \quad \text{(CLT regime)} \tag{17}$$

and

$$\mathbb{P}\left\{\sum_{k=1}^{N}\xi_k \geq s\right\} \simeq N \cdot V(s), \quad s > \sqrt{(\alpha-2)\sigma^2 N \ln N} \quad \text{(heavy-tailed regime).} \tag{18}$$

This simple observation can play a significant role in deriving complexity results for non-smooth convex optimization under the assumption that stochastic gradients are heavy-tailed, see [15] for the details.

## E    Technical Results

**Lemma E.1.** Consider two sequences of non-negative numbers $\{\alpha_k\}_{k\geq 0}$ and $\{A_k\}_{k\geq 0}$ such that

$$\alpha_0 = A_0 = 0, \quad A_{k+1} = A_k + \alpha_{k+1}, \quad \alpha_{k+1} = \frac{k+2}{2aL} \quad \forall k \geq 0, \tag{19}$$

where $a, L > 0$. Then for all $k \geq 0$

$$A_{k+1} = \frac{(k+1)(k+4)}{4aL}, \tag{20}$$

$$A_{k+1} \geq aL\alpha_{k+1}^2. \tag{21}$$

*Proof.* By definition of $A_{k+1}$ we have that

$$A_{k+1} = \sum_{l=1}^{k+1}\alpha_l = \frac{1}{2aL}\sum_{l=1}^{k+1}(l+1) = \frac{(k+1)(k+4)}{4aL}.$$

Using $(k+1)(k+4) \geq (k+2)^2$ together with the inequality above we derive (21). $\square$

# F Accelerated SGD with Clipping: Exact Formulations and Missing Proofs

In this section we provide exact formulations of all the results that we have for `clipped-SSTM` and `R-clipped-SSTM` together with the full proofs.

## F.1 Convex Case

Recall that in order to compute $\text{clip}(\nabla f(x, \boldsymbol{\xi}), \lambda)$ one needs to get $m$ i.i.d. samples $\nabla f(x, \xi_1), \ldots, \nabla f(x, \xi_m)$, compute its average

$$\nabla f(x, \boldsymbol{\xi}) = \frac{1}{m} \sum_{i=1}^{m} \nabla f(x, \xi_i), \tag{22}$$

and then project the result $\nabla f(x, \boldsymbol{\xi})$ on the Euclidean ball with radius $\lambda$ and center at the origin. We also notice that

$$\mathbb{E}_\xi[\nabla f(x, \boldsymbol{\xi})] = \nabla f(x), \tag{23}$$

$$\mathbb{E}_\xi\left[\|\nabla f(x, \boldsymbol{\xi}) - \nabla f(x)\|_2^2\right] \leq \frac{\sigma^2}{m}. \tag{24}$$

### F.1.1 Convergence Guarantees for `clipped-SSTM`

Next theorem summarizes the main convergence result for `clipped-SSTM`.

---

**Theorem F.1.** Assume that function $f$ is convex and $L$-smooth. Then for all $\beta \in (0, 1)$ and $N \geq 1$ such that

$$\ln \frac{4N}{\beta} \geq 2 \tag{25}$$

we have that after $N$ iterations of `clipped-SSTM` with

$$m_k = \max\left\{1, \frac{6000\sigma^2\alpha_{k+1}^2 N \ln \frac{4N}{\beta}}{C^2 R_0^2}, \frac{10368\sigma^2\alpha_{k+1}^2 N}{C^2 R_0^2}\right\}, \tag{26}$$

$$B = \frac{CR_0}{8\ln\frac{4N}{\beta}}, \quad a \geq \max\left\{1, \frac{16\ln\frac{4N}{\beta}}{C}, 36\left(2\ln\frac{4N}{\beta} + \sqrt{4\ln^2\frac{4N}{\beta} + 2\ln\frac{4N}{\beta}}\right)^2\right\}, \tag{27}$$

that with probability at least $1 - \beta$

$$f(y^N) - f(x^*) \leq \frac{2aLC^2R_0^2}{N(N+3)}, \tag{28}$$

where $R_0 = \|x^0 - x^*\|_2$ and

$$C = \sqrt{5}. \tag{29}$$

In other words, if we choose $a$ to be equal to the maximum from (27), then the method achieves $f(y^N) - f(x^*) \leq \varepsilon$ with probability at least $1 - \beta$ after $O\left(\sqrt{\frac{LR_0^2}{\varepsilon}} \ln \frac{LR_0^2}{\varepsilon\beta}\right)$ iterations and requires

$$O\left(\max\left\{\sqrt{\frac{LR_0^2}{\varepsilon}}, \frac{\sigma^2 R_0^2}{\varepsilon^2}\right\} \ln \frac{LR_0^2}{\varepsilon\beta}\right) \text{ oracle calls.} \tag{30}$$

---

One can easily notice that multiplicative constant factors in formulas for $m_k$ and $a$ are too big and seem to be impractical, but in practice one can tune these constants to get good enough performance. That is, big constants in (26) and (27) are needed only in our analysis in order to get bound (30).

Finally, when $\sigma^2$ is big then Theorem F.1 says that at iteration $k$ `clipped-SGD` requires large batchsizes $m_k \sim k^2 N$ (see (26)) which is proportional to $\varepsilon^{-3/2}$ for last iterates. It can make the cost of one iteration extremely high, therefore, we consider different stepsize policies that remove this drawback.

**Corollary F.2.** Let the assumptions of Theorem F.1 hold.

1. **(Medium batchsize).** If $N$ and $\beta$ are such that $N \ln \frac{4N}{\beta}$ is bigger than the maximum from (27), then for $a = N \ln \frac{4N}{\beta}$ we have

$$m_k = \max\left\{1, \frac{6000\sigma^2(k+2)^2}{4L^2NC^2R_0^2 \ln \frac{4N}{\beta}}, \frac{10368\sigma^2(k+2)^2}{4L^2C^2R_0^2N \ln^2 \frac{4N}{\beta}}\right\} \qquad (31)$$

and the method achieves $f(y^N) - f(x^*) \leq \varepsilon$ with probability at least $1 - \beta$ after $O\left(\frac{LR_0^2}{\varepsilon} \ln \frac{LR_0^2}{\varepsilon\beta}\right)$ iterations and requires

$$O\left(\max\left\{\frac{LR_0^2}{\varepsilon}, \frac{\sigma^2R_0^2}{\varepsilon^2}\right\} \ln \frac{LR_0^2}{\varepsilon\beta}\right) \text{ oracle calls.} \qquad (32)$$

2. **(Constant batchsize).** If $N$ and $\beta$ are such that $a_0 N^{3/2}\sqrt{\ln \frac{4N}{\beta}}$ is bigger than the maximum from (27) for some positive constant $a_0$, then for $a = a_0 N^{3/2}\sqrt{\ln \frac{4N}{\beta}}$ we have

$$m_k = \max\left\{1, \frac{6000\sigma^2(k+2)^2}{4a_0^2L^2N^2C^2R_0^2}, \frac{10368\sigma^2(k+2)^2}{4a_0^2L^2C^2R_0^2N^2 \ln \frac{4N}{\beta}}\right\} \qquad (33)$$

and the method achieves $f(y^N) - f(x^*) \leq \varepsilon$ with probability at least $1 - \beta$ after $O\left(\frac{a_0^2L^2R_0^4}{\varepsilon^2} \ln \frac{a_0LR_0^2}{\varepsilon\beta}\right)$ iterations and requires

$$O\left(\max\left\{\frac{a_0^2L^2R_0^4}{\varepsilon^2}, \frac{\sigma^2R_0^2}{\varepsilon^2}\right\} \ln \frac{a_0LR_0^2}{\varepsilon\beta}\right) \text{ oracle calls.} \qquad (34)$$

Finally, if $a_0 = \frac{\sigma}{LR_0}$, then $m_k = O(1)$ for $k = 0, 1, \ldots, N$ and `clipped-SSTM` finds $\varepsilon$-solution with probability at least $1 - \beta$ after $O\left(\frac{\sigma^2R_0^2}{\varepsilon^2} \ln \frac{\sigma R_0}{\varepsilon\beta}\right)$ iterations and requires $O(1)$ oracle calls per iteration.

In the first case batchsizes increase from $O(1)$ for $k = 1$ to $O(\varepsilon^{-1})$ for $k = N$ and the overall complexity recovers the complexity of Robust Stochastic Mirror Descent (RSMD) from [47]. However, analysis from [47] works only for the optimization problems on *compact* convex sets, whereas our analysis handles an unconstrained optimization on $\mathbb{R}^n$. Despite the similarities of our approach and [47], it seems that the technique from [47] cannot be generalized to obtain the complexity like in (30) due to the fast bias accumulation that appears because of the special truncation of stochastic gradients that is used in RSMD.

In the second case the corollary establishes $\varepsilon^{-2}\ln(\varepsilon^{-1}\beta^{-1})$ rate for `clipped-SSTM` with constant batchsizes, i.e. $m_k = O(1)$ for all $k$. The ability of `clipped-SSTM` to converge with constant batchsizes makes it more practical and applicable for wider class of problems where it can be very expensive to compute large batchsizes, e.g. training deep neural networks. Moreover, when $\sigma$ is not too small, i.e. $\sigma^2 \geq L\varepsilon$, this rate is optimal (up to logarithmical factors) and also recovers the rate of RSMD.

Finally, setting

$$a' = \max\left\{1, \frac{16\ln \frac{4N}{\beta}}{C}, 36\left(2\ln \frac{4N}{\beta} + \sqrt{4\ln^2 \frac{4N}{\beta} + 2\ln \frac{4N}{\beta}}\right)^2\right\},$$

$$a = \max\left\{a', \frac{\sigma N^{3/2}}{LR_0}\sqrt{\ln \frac{4N}{\beta}}\right\} \qquad (35)$$

and $m_k$ as in (26), we get $m_k = O(1)$ for $k = 0, 1, \ldots, N$ and derive the following result.

**Corollary F.3.** Let the assumptions of Theorem F.1 hold, $a$ is chosen as in (35) and $m_k$ is computed via (26). Then `clipped-SSTM` achieves $f(y^N) - f(x^*) \le \varepsilon$ with probability at least $1 - \beta$ after

$$O\left(\max\left\{\sqrt{\frac{LR_0^2}{\varepsilon}}, \frac{\sigma^2 R_0^2}{\varepsilon^2}\right\} \ln \frac{LR_0^2 + \sigma R_0}{\varepsilon\beta}\right) \quad \text{iterations/oracle calls.}$$

### F.1.2 Sketch of the Proof of Theorem F.1

We start with the following lemma that is pretty standard in the analysis of Stochastic Similar Triangles Method, e.g. see the proof of Theorem 1 from [10].

**Lemma F.4.** Let $f$ be a convex $L$-smooth function and let stepsize parameter $a$ satisfy $a \ge 1$. Then after $N \ge 0$ iterations of `clipped-SSTM` for all $z \in \mathbb{R}^n$ we have

$$A_N\left(f(y^N) - f(z)\right) \le \frac{1}{2}\|z^0 - z\|_2^2 - \frac{1}{2}\|z^N - z\|_2^2 + \sum_{k=0}^{N-1} \alpha_{k+1} \left\langle \theta_{k+1}, z - z^k \right\rangle$$

$$+ \sum_{k=0}^{N-1} \alpha_{k+1}^2 \|\theta_{k+1}\|_2^2 + \sum_{k=0}^{N-1} \alpha_{k+1}^2 \left\langle \theta_{k+1}, \nabla f(x^{k+1}) \right\rangle, \quad (36)$$

$$\theta_{k+1} \overset{\text{def}}{=} \widetilde{\nabla} f(x^{k+1}, \boldsymbol{\xi}^k) - \nabla f(x^{k+1}). \quad (37)$$

That is, if $z = x^*$, then the result above gives a preliminary upper bound for $A_N(f(y^N) - f(x^*))$. The first and the second terms in the r.h.s. of (36) come from the analysis of Similar Triangles Method [16] and three last terms have a stochastic nature. In particular, they explicitly depend on differences $\theta_{k+1} = \widetilde{\nabla} f(x^{k+1}, \boldsymbol{\xi}^k) - \nabla f(x^{k+1})$ between clipped mini-batched stochastic gradients and full gradients at $x^{k+1}$, so, if $\widetilde{\nabla} f(x^{k+1}, \boldsymbol{\xi}^k) = \nabla f(x^{k+1})$ with probability 1, then we easily get needed convergence rate. However, we are interested in the more general case and, as a consequence, to continue the proof, we need to find a good enough upper bound for the last three terms from (36). In other words, we need to show that choosing parameters $a$, $m_k$ and $\lambda_{k+1}$ properly we can upper bound these terms by something that coincides with $\|z^0 - x^*\|_2^2$ up to numerical multiplicative constant. The proof of convergence result for `RSMD` from [47] where authors provide upper bound for similar sums hints that Bernstein's inequality (see Lemma D.1) applied to estimate these terms can help us to reach our goal. In order to apply Bernstein's inequality one should derive tight bounds for such characteristics of $\widetilde{\nabla} f(x^{k+1}, \boldsymbol{\xi}^k)$ as upper bounds for the magnitude, bias, variance and distortion and the next lemma provides us with this.

**Lemma F.5.** For all $k \ge 0$ the following inequality holds:

$$\left\|\widetilde{\nabla} f(x^{k+1}, \boldsymbol{\xi}^k) - \mathbb{E}_{\boldsymbol{\xi}^k}\left[\widetilde{\nabla} f(x^{k+1}, \boldsymbol{\xi}^k)\right]\right\|_2 \le 2\lambda_{k+1}. \quad (38)$$

Moreover, if $\|\nabla f(x^{k+1})\|_2 \le \frac{\lambda_{k+1}}{2}$ for some $k \ge 0$, then for this $k$ we have:

$$\left\|\mathbb{E}_{\boldsymbol{\xi}^k}\left[\widetilde{\nabla} f(x^{k+1}, \boldsymbol{\xi}^k)\right] - \nabla f(x^{k+1})\right\|_2 \le \frac{4\sigma^2}{m_k \lambda_{k+1}}, \quad (39)$$

$$\mathbb{E}_{\boldsymbol{\xi}^k}\left[\left\|\widetilde{\nabla} f(x^{k+1}, \boldsymbol{\xi}^k) - \nabla f(x^{k+1})\right\|_2^2\right] \le \frac{18\sigma^2}{m_k}, \quad (40)$$

$$\mathbb{E}_{\boldsymbol{\xi}^k}\left[\left\|\widetilde{\nabla} f(x^{k+1}, \boldsymbol{\xi}^k) - \mathbb{E}_{\boldsymbol{\xi}^k}\left[\widetilde{\nabla} f(x^{k+1}, \boldsymbol{\xi}^k)\right]\right\|_2^2\right] \le \frac{18\sigma^2}{m_k}. \quad (41)$$

Clearly, clipping introduces a bias in $\widetilde{\nabla} f(x^{k+1}, \boldsymbol{\xi}^k)$ which influences the convergence of the method. Hence, the clipping level $\lambda_{k+1}$ should be chosen in a very accurate way. Below we informally describe what does it mean and present the sketch of the remaining part of the proof.

Imagine the ideal situation: $\nabla f(x^{k+1}, \boldsymbol{\xi}^k) = \nabla f(x^{k+1})$ with probability 1 for all $k$, i.e. we have an access to the full gradients at points $x^{k+1}$. Then it is natural to choose $\lambda_{k+1}$ in such a way that $\text{clip}(\nabla f(x^{k+1}), \lambda^{k+1}) = \nabla f(x^{k+1})$ in order to recover Similar Triangles Method (STM) that

converges with optimal rate in the deterministic case. In other words, one can pick $\lambda_{k+1}$ such that $\|\nabla f(x^{k+1})\|_2 \le \lambda_{k+1}$ and get an optimal method. Since we know that in this case the method should converge with $O(1/k^2)$ rate in terms of $f(x^k) - f(x^*)$ one can expect that the gradient's norm decays with $O(1/k)$ rate, so, one can choose $\lambda_{k+1}$ to be proportional to $1/k$. It is exactly what we do when we define $\lambda_{k+1}$ as $B/\alpha_{k+1}$.

The ideal case described above gives a good insight on how to choose $\lambda_{k+1}$ in the general case and can be described as follows: if we want to prevent our gradient estimator $\widetilde{\nabla} f(x^{k+1}, \boldsymbol{\xi}^k)$ from large deviations from $\nabla f(x^{k+1})$ with high probability, then it is needed to choose $\lambda_{k+1}$ such that $\|\nabla f(x^k)\|_2 \le c\lambda_{k+1}$ with high probability where $c < 1$ is some positive number. This choice guarantees that with high probability clipped mini-batched gradient $\widetilde{\nabla} f(x^{k+1}, \boldsymbol{\xi}^k)$ cannot deviates from $\nabla f(x^{k+1})$ significantly and, as a consequence, the convergence rate of clipped-SSTM in terms of the number of iterations needed to achieve the desired accuracy of the solution with high probability becomes similar to the convergence rate of STM up to some logarithmical factors depending on the confidence level.

In particular, we choose $\lambda_{k+1}$ such that $\|\nabla f(x^{k+1})\|_2 \le \lambda_{k+1}/2$ with high probability. Moreover, we derive this relation by induction via refined estimation of the three last terms from the r.h.s. of (36) that is based on the new variant of advanced recurrences technique from [22, 23]. The main trick there is in showing by induction that sequence $\|z^k - x^*\|_2$ is bounded by some constant multiplied by $\|x^0 - x^*\|_2$ and in deriving $\|\nabla f(x^{k+1})\|_2 \le \lambda_{k+1}/2$ simultaneously for all $k = 0, 1, \ldots, N$. With such bounds and Lemma F.5 in hand, it is possible to apply Bernstein's inequality to three sums from the r.h.s. of (36) since all summands are bounded *with high probability*. After applying Bernstein's inequality we adjust parameters $\alpha_{k+1}$ and $m_k$ in such a way that after rearranging the terms in the obtained upper bounds we get that r.h.s. in (36) (with $z = x^*$) is smaller than $\|x^0 - x^*\|_2^2$ up to some multiplicative numerical constant. This finishes the proof.

To conclude, the key tools in our analysis are Bernstein's inequality (see Lemma D.1) and advanced recurrences technique [22, 23] that helps us to show boundedness of $\|z^N - x^*\|_2$ and $\|\nabla f(x^{k+1})\|_2 \le \lambda_{k+1}/2$ with high probability. We provide detailed proofs of presented result in the Appendix (see Section F.3).

### F.2 Strongly Convex Case

In this section we assume additionally that $f(x)$ is $\mu$-strongly convex. For this case we modify Algorithm 1 and propose a new method called Restarted Clipped Similar Triangles Method (R-clipped-SSTM), see Algorithm 3. At each iteration R-clipped-SSTM runs clipped-SSTM for

---

**Algorithm 3** Restarted Clipped Stochastic Similar Triangles Method (R-clipped-SSTM)

---

**Input:** starting point $x^0$, number of iterations $N_0$ of clipped-SSTM, number of clipped-SSTM runs, batchsizes $\{m_k^0\}_{k=0}^{N_0-1}, \{m_k^1\}_{k=0}^{N_0-1}, \ldots, \{m_k^\tau\}_{k=0}^{N_0-1}$, stepsize parameter $a$, clipping parameters $\{B_t\}_{t=0}$
  1: Set $\hat{x}^0 = x^0$
  2: **for** $t = 0, 1, \ldots, \tau - 1$ **do**
  3:     Run clipped-SSTM (Algorithm 1) for $N_0$ iterations with batchsizes $\{m_k^t\}_{k=1}^{N_0}$, stepsize parameter $a$, clipping parameter $B_t$ and starting point $\hat{x}^t$. Define the output of clipped-SSTM by $\hat{x}^{t+1}$.
  4: **end for**
**Output:** $\hat{x}^\tau$

---

$N_0$ iterations from the current point $\hat{x}^k$ and use its output as next iterate $\hat{x}^{k+1}$. In literature this approach is known as the restarts technique [11, 31, 32, 51]. Choosing $N_0$ and parameters $m_k$, $a$ and $B$ in a proper way one can get an accelerated method for strongly convex objectives. Theorem below states the main convergence result for R-clipped-SSTM.

**Theorem F.6.** Assume that $f$ is $\mu$-strongly convex and $L$-smooth. If we choose $\beta \in (0,1)$, $\tau$ and $N_0 \geq 1$ such that

$$\ln \frac{4N_0\tau}{\beta} \geq 2, \quad N_0 \geq C\sqrt{\frac{8aL}{\mu}}, \tag{42}$$

and

$$m_k^t = \max\left\{1, \frac{6000 \cdot 2^t \sigma^2 \alpha_{k+1}^2 N_0 \ln \frac{4N_0\tau}{\beta}}{C^2 R^2}, \frac{10368 \cdot 2^t \sigma^2 \alpha_{k+1}^2 N_0}{C^2 R^2}\right\}, \tag{43}$$

$$B_t = \frac{CR}{8 \cdot 2^t \ln \frac{4N_0\tau}{\beta}}, \tag{44}$$

$$a \geq \max\left\{1, \frac{16\ln \frac{4N_0\tau}{\beta}}{C}, 36\left(2\ln \frac{4N_0\tau}{\beta} + \sqrt{4\ln^2 \frac{4N_0\tau}{\beta} + 2\ln \frac{4N_0\tau}{\beta}}\right)^2\right\}, \tag{45}$$

where $R = \sqrt{\frac{2(f(x^0)-f(x^*))}{\mu}}$ and $C = \sqrt{5}$, then we have that after $\tau$ runs of `clipped-SSTM` in `R-clipped-SSTM` the inequality

$$f(\hat{x}^\tau) - f(x^*) \leq 2^{-\tau}\left(f(x^0) - f(x^*)\right) \tag{46}$$

holds with probability at least $1 - \beta$. That is, if we choose $a$ to be equal to the maximum from (45) and $N_0 \leq C_1\sqrt{\frac{8aL}{\mu}}$ with some numerical constant $C_1 \geq C$, then the method achieves $f(\hat{x}^\tau) - f(x^*) \leq \varepsilon$ with probability at least $1 - \beta$ after

$$O\left(\sqrt{\frac{L}{\mu}} \ln\left(\frac{\mu R^2}{\varepsilon}\right) \ln\left(\frac{L}{\mu\beta} \ln \frac{\mu R^2}{\varepsilon}\right)\right) \quad \text{iterations (in total)} \tag{47}$$

of `clipped-SSTM` and requires

$$O\left(\max\left\{\sqrt{\frac{L}{\mu}} \ln \frac{\mu R^2}{\varepsilon}, \frac{\sigma^2}{\mu\varepsilon}\right\} \ln\left(\frac{L}{\mu\beta} \ln \frac{\mu R^2}{\varepsilon}\right)\right) \quad \text{oracle calls.} \tag{48}$$

In other words, `R-clipped-SSTM` has the same convergence rate as optimal stochastic methods for strongly convex problems like Multi-Staged `AC-SA` (`MS-AC-SA`) [19] or Stochastic Similar Triangles Method for strongly convex problems (`SSTM_sc`) [16, 22]. Moreover, in Theorem F.6 we *do not assume* that stochastic gradients are sampled from sub-Gaussian distribution while corresponding results for `MS-AC-SA` and `SSTM_sc` are substantially based on the light tails assumption. Our bound outperforms the state-of-the-art result from [7] in terms of the dependence on $\ln \frac{L}{\mu}$. It is worth to mention here that using special restarts technique Nazin et al. [47] generalize their method (`RSMD`) for the strongly convex case, but since `RSMD` is not accelerated their approach gives only non-accelerated convergence rate.

We also emphasize that big numerical factors in formulas for $m_k^t$ and $a$ are needed only in our analysis and in practice they can be tuned. However, when $\sigma^2$ is big batchsizes $m_k^t$ become of the order $k^2\varepsilon^{-1}$. It can make the cost of one iteration extremely high, therefore, as for `clipped-SSTM` we consider a different stepsize policy removing this drawback.

**Corollary F.7.** Let the assumptions of Theorem F.6 hold. Assume that conditions (42), (43), (44) and (45) are satisfied for

$$a = \Theta\left(\frac{\sigma^4 \ln^2 \frac{N_0\tau}{\beta}}{L\mu\varepsilon^2}\right), \quad N_0 = \Theta\left(\sqrt{\frac{aL}{\mu}}\right). \tag{49}$$

Then after $\tau = \lceil \ln(\mu R^2/2\varepsilon) \rceil$ runs of `clipped-SSTM` in `R-clipped-SSTM` the method achieves $f(\hat{x}^\tau) - f(x^*) \leq \varepsilon$ with probability at least $1 - \beta$. Moreover, the total number of iterations of

When $\sigma^2$ is big the obtained bound is comparable with bounds for `restarted-RSMD` and `proxBoost`, see Table 2.

### F.3   Proofs

#### F.3.1   Proof of Lemma F.4

Using $z^{k+1} = z^k - \alpha_{k+1}\widetilde{\nabla} f(x^{k+1}, \boldsymbol{\xi}^k)$ we get that for all $z \in \mathbb{R}^n$

$$
\begin{aligned}
\alpha_{k+1}\left\langle\widetilde{\nabla} f(x^{k+1}, \boldsymbol{\xi}^k), z^k - z\right\rangle &= \alpha_{k+1}\left\langle\widetilde{\nabla} f(x^{k+1}, \boldsymbol{\xi}^k), z^k - z^{k+1}\right\rangle \\
&\quad + \alpha_{k+1}\left\langle\widetilde{\nabla} f(x^{k+1}, \boldsymbol{\xi}^k), z^{k+1} - z\right\rangle \\
&= \alpha_{k+1}\left\langle\widetilde{\nabla} f(x^{k+1}, \boldsymbol{\xi}^k), z^k - z^{k+1}\right\rangle + \left\langle z^{k+1} - z^k, z - z^{k+1}\right\rangle \\
&\overset{(13)}{\leq} \alpha_{k+1}\left\langle\widetilde{\nabla} f(x^{k+1}, \boldsymbol{\xi}^k), z^k - z^{k+1}\right\rangle - \frac{1}{2}\|z^k - z^{k+1}\|_2^2 \\
&\quad + \frac{1}{2}\|z^k - z\|_2^2 - \frac{1}{2}\|z^{k+1} - z\|_2^2. \tag{51}
\end{aligned}
$$

Next, we notice that

$$y^{k+1} = \frac{A_k y^k + \alpha_{k+1} z^{k+1}}{A_{k+1}} = \frac{A_k y^k + \alpha_{k+1} z^k}{A_{k+1}} + \frac{\alpha_{k+1}}{A_{k+1}}(z^{k+1} - z^k) = x^{k+1} + \frac{\alpha_{k+1}}{A_{k+1}}(z^{k+1} - z^k) \tag{52}$$

which implies:

$$
\begin{aligned}
\alpha_{k+1}\left\langle\widetilde{\nabla} f(x^{k+1}, \boldsymbol{\xi}^k), z^k - z\right\rangle &\overset{(37),(51)}{\leq} \alpha_{k+1}\left\langle\nabla f(x^{k+1}), z^k - z^{k+1}\right\rangle - \frac{1}{2}\|z^k - z^{k+1}\|_2^2 \\
&\quad + \alpha_{k+1}\left\langle\theta_{k+1}, z^k - z^{k+1}\right\rangle + \frac{1}{2}\|z^k - z\|_2^2 - \frac{1}{2}\|z^{k+1} - z\|_2^2 \\
&\overset{(52)}{=} A_{k+1}\left\langle\nabla f(x^{k+1}), x^{k+1} - y^{k+1}\right\rangle - \frac{1}{2}\|z^k - z^{k+1}\|_2^2 \\
&\quad + \alpha_{k+1}\left\langle\theta_{k+1}, z^k - z^{k+1}\right\rangle + \frac{1}{2}\|z^k - z\|_2^2 - \frac{1}{2}\|z^{k+1} - z\|_2^2 \\
&\overset{(7)}{\leq} A_{k+1}\left(f(x^{k+1}) - f(y^{k+1})\right) + \frac{A_{k+1}L}{2}\|x^{k+1} - y^{k+1}\|_2^2 \\
&\quad - \frac{1}{2}\|z^k - z^{k+1}\|_2^2 + \alpha_{k+1}\left\langle\theta_{k+1}, z^k - z^{k+1}\right\rangle \\
&\quad + \frac{1}{2}\|z^k - z\|_2^2 - \frac{1}{2}\|z^{k+1} - z\|_2^2 \\
&\overset{(52)}{=} A_{k+1}\left(f(x^{k+1}) - f(y^{k+1})\right) + \frac{1}{2}\left(\frac{\alpha_{k+1}^2 L}{A_{k+1}} - 1\right)\|z^k - z^{k+1}\|_2^2 \\
&\quad + \alpha_{k+1}\left\langle\theta_{k+1}, z^k - z^{k+1}\right\rangle + \frac{1}{2}\|z^k - z\|_2^2 - \frac{1}{2}\|z^{k+1} - z\|_2^2.
\end{aligned}
$$

Since $A_{k+1} \geq aL\alpha_{k+1}^2$ (see Lemma E.1) and $a \geq 1$ we can continue our derivations:

$$
\begin{aligned}
\alpha_{k+1}\left\langle\widetilde{\nabla} f(x^{k+1}, \boldsymbol{\xi}^k), z^k - z\right\rangle &\leq A_{k+1}\left(f(x^{k+1}) - f(y^{k+1})\right) + \alpha_{k+1}\left\langle\theta_{k+1}, z^k - z^{k+1}\right\rangle \\
&\quad + \frac{1}{2}\|z^k - z\|_2^2 - \frac{1}{2}\|z^{k+1} - z\|_2^2. \tag{53}
\end{aligned}
$$

Next, due to convexity of $f$ we have

$$
\begin{aligned}
\left\langle \widetilde{\nabla} f(x^{k+1}, \boldsymbol{\xi}^k), y^k - x^{k+1} \right\rangle \overset{(37)}{=} & \left\langle \nabla f(x^{k+1}), y^k - x^{k+1} \right\rangle + \left\langle \theta_{k+1}, y^k - x^{k+1} \right\rangle \\
\leq & \ f(y^k) - f(x^{k+1}) + \left\langle \theta_{k+1}, y^k - x^{k+1} \right\rangle.
\end{aligned} \tag{54}
$$

By definition of $x^{k+1}$ we have $x^{k+1} = \frac{A_k y^k + \alpha_{k+1} z^k}{A_{k+1}}$ which implies

$$
\alpha_{k+1} \left( x^{k+1} - z^k \right) = A_k \left( y^k - x^{k+1} \right) \tag{55}
$$

since $A_{k+1} = A_k + \alpha_{k+1}$. Putting all together we derive that

$$
\begin{aligned}
\alpha_{k+1} \left\langle \widetilde{\nabla} f(x^{k+1}, \boldsymbol{\xi}^k), x^{k+1} - z \right\rangle \quad = \quad & \alpha_{k+1} \left\langle \widetilde{\nabla} f(x^{k+1}, \boldsymbol{\xi}^k), x^{k+1} - z^k \right\rangle \\
& + \alpha_{k+1} \left\langle \widetilde{\nabla} f(x^{k+1}, \boldsymbol{\xi}^k), z^k - z \right\rangle \\
\overset{(55)}{=} \quad & A_k \left\langle \widetilde{\nabla} f(x^{k+1}, \boldsymbol{\xi}^k), y^k - x^{k+1} \right\rangle \\
& + \alpha_{k+1} \left\langle \widetilde{\nabla} f(x^{k+1}, \boldsymbol{\xi}^k), z^k - z \right\rangle \\
\overset{(54),(53)}{\leq} \quad & A_k \left( f(y^k) - f(x^{k+1}) \right) + A_k \left\langle \theta_{k+1}, y^k - x^{k+1} \right\rangle \\
& + A_{k+1} \left( f(x^{k+1}) - f(y^{k+1}) \right) + \alpha_{k+1} \left\langle \theta_{k+1}, z^k - z^{k+1} \right\rangle \\
& + \frac{1}{2} \|z^k - z\|_2^2 - \frac{1}{2} \|z^{k+1} - z\|_2^2 \\
\overset{(55)}{=} \quad & A_k f(y^k) - A_{k+1} f(y^{k+1}) + \alpha_{k+1} \left\langle \theta_{k+1}, x^{k+1} - z^k \right\rangle \\
& + \alpha_{k+1} f(x^{k+1}) + \alpha_{k+1} \left\langle \theta_{k+1}, z^k - z^{k+1} \right\rangle \\
& + \frac{1}{2} \|z^k - z\|_2^2 - \frac{1}{2} \|z^{k+1} - z\|_2^2 \\
\leq \quad & A_k f(y^k) - A_{k+1} f(y^{k+1}) + \alpha_{k+1} f(x^{k+1}) \\
& + \alpha_{k+1} \left\langle \theta_{k+1}, x^{k+1} - z^{k+1} \right\rangle \\
& + \frac{1}{2} \|z^k - z\|_2^2 - \frac{1}{2} \|z^{k+1} - z\|_2^2.
\end{aligned}
$$

Rearranging the terms we get

$$
\begin{aligned}
A_{k+1} f(y^{k+1}) - A_k f(y^k) \quad \leq \quad & \alpha_{k+1} \left( f(x^{k+1}) + \left\langle \widetilde{\nabla} f(x^{k+1}, \boldsymbol{\xi}^k), z - x^{k+1} \right\rangle \right) + \frac{1}{2} \|z^k - z\|_2^2 \\
& - \frac{1}{2} \|z^{k+1} - z\|_2^2 + \alpha_{k+1} \left\langle \theta_{k+1}, x^{k+1} - z^{k+1} \right\rangle \\
\overset{(37)}{=} \quad & \alpha_{k+1} \left( f(x^{k+1}) + \left\langle \nabla f(x^{k+1}), z - x^{k+1} \right\rangle \right) \\
& + \alpha_{k+1} \left\langle \theta_{k+1}, z - x^{k+1} \right\rangle + \frac{1}{2} \|z^k - z\|_2^2 - \frac{1}{2} \|z^{k+1} - z\|_2^2 \\
& + \alpha_{k+1} \left\langle \theta_{k+1}, x^{k+1} - z^{k+1} \right\rangle \\
\leq \quad & \alpha_{k+1} f(z) + \frac{1}{2} \|z^k - z\|_2^2 - \frac{1}{2} \|z^{k+1} - z\|_2^2 + \alpha_{k+1} \left\langle \theta_{k+1}, z - z^{k+1} \right\rangle
\end{aligned}
$$

where in the last inequality we use the convexity of $f$. Taking into account $A_0 = \alpha_0 = 0$ and $A_N = \sum_{k=0}^{N-1} \alpha_{k+1}$ we sum up these inequalities for $k = 0, \ldots, N-1$ and get

$$
\begin{aligned}
A_N f(y^N) \;\leq\; & A_N f(z) + \frac{1}{2}\|z^0 - z\|_2^2 - \frac{1}{2}\|z^N - z\|_2^2 + \sum_{k=0}^{N-1} \alpha_{k+1} \left\langle \theta_{k+1}, z - z^{k+1} \right\rangle \\
=\; & A_N f(z) + \frac{1}{2}\|z^0 - z\|_2^2 - \frac{1}{2}\|z^N - z\|_2^2 + \sum_{k=0}^{N-1} \alpha_{k+1} \left\langle \theta_{k+1}, z - z^k \right\rangle \\
& + \sum_{k=0}^{N-1} \alpha_{k+1}^2 \left\langle \theta_{k+1}, \widetilde{\nabla} f(x^{k+1}, \boldsymbol{\xi}^k) \right\rangle \\
\stackrel{(37)}{=}\; & A_N f(z) + \frac{1}{2}\|z^0 - z\|_2^2 - \frac{1}{2}\|z^N - z\|_2^2 + \sum_{k=0}^{N-1} \alpha_{k+1} \left\langle \theta_{k+1}, z - z^k \right\rangle \\
& + \sum_{k=0}^{N-1} \alpha_{k+1}^2 \|\theta_{k+1}\|_2^2 + \sum_{k=0}^{N-1} \alpha_{k+1}^2 \left\langle \theta_{k+1}, \nabla f(x^{k+1}) \right\rangle
\end{aligned}
$$

which concludes the proof.

### F.3.2 Proof of Lemma F.5

**Proof of** (38). By definition of $\widetilde{\nabla} f(x^{k+1}, \boldsymbol{\xi}^k)$ we have that $\|\widetilde{\nabla} f(x^{k+1}, \boldsymbol{\xi}^k)\|_2 \leq \lambda_{k+1}$ and, as a consequence, $\left\|\mathbb{E}_{\boldsymbol{\xi}^k}[\widetilde{\nabla} f(x^{k+1}, \boldsymbol{\xi}^k)]\right\|_2 \leq \lambda_{k+1}$. Using this we get

$$
\left\|\widetilde{\nabla} f(x^{k+1}, \boldsymbol{\xi}^k) - \mathbb{E}_{\boldsymbol{\xi}^k}\left[\widetilde{\nabla} f(x^{k+1}, \boldsymbol{\xi}^k)\right]\right\|_2 \leq \left\|\widetilde{\nabla} f(x^{k+1}, \boldsymbol{\xi}^k)\right\|_2 + \left\|\mathbb{E}_{\boldsymbol{\xi}^k}\left[\widetilde{\nabla} f(x^{k+1}, \boldsymbol{\xi}^k)\right]\right\|_2 \leq 2\lambda_{k+1}.
$$

**Proof of** (39). In order to prove this bound we introduce following indicator random variables:

$$
\chi_k \overset{\text{def}}{=} \mathbb{1}_{\|\nabla f(x^{k+1}, \boldsymbol{\xi}^k)\|_2 > \lambda_{k+1}}, \qquad \eta_k \overset{\text{def}}{=} \mathbb{1}_{\|\nabla f(x^{k+1}, \boldsymbol{\xi}^k) - \nabla f(x^{k+1})\|_2 > \frac{1}{2}\lambda_{k+1}}. \tag{56}
$$

From the assumptions of the lemma, we have that $\|\nabla f(x^{k+1})\|_2 \leq \frac{\lambda_{k+1}}{2}$ which implies

$$
\begin{aligned}
\left\|\nabla f(x^{k+1}, \boldsymbol{\xi}^k)\right\|_2 \;\leq\; & \left\|\nabla f(x^{k+1}, \boldsymbol{\xi}^k) - \nabla f(x^{k+1})\right\|_2 + \left\|\nabla f(x^{k+1})\right\|_2 \\
\leq\; & \left\|\nabla f(x^{k+1}, \boldsymbol{\xi}^k) - \nabla f(x^{k+1})\right\|_2 + \frac{\lambda_{k+1}}{2},
\end{aligned}
$$

hence

$$
\chi_k \leq \eta_k. \tag{57}
$$

The introduced notation helps us to rewrite $\widetilde{\nabla} f(x^{k+1}, \boldsymbol{\xi}^k)$ in the following way:

$$
\begin{aligned}
\widetilde{\nabla} f(x^{k+1}, \boldsymbol{\xi}^k) \;=\; & \nabla f(x^{k+1}, \boldsymbol{\xi}^k)(1 - \chi_k) + \frac{\lambda_{k+1}}{\|\nabla f(x^{k+1}, \boldsymbol{\xi})\|_2} \nabla f(x^{k+1}, \boldsymbol{\xi}^k)\chi_k \tag{58} \\
=\; & \nabla f(x^{k+1}, \boldsymbol{\xi}^k) + \left( \frac{\lambda_{k+1}}{\left\|\nabla f(x^{k+1}, \boldsymbol{\xi}^k)\right\|_2} - 1 \right) \nabla f(x^{k+1}, \boldsymbol{\xi}^k)\chi_k. \tag{59}
\end{aligned}
$$

We use this representation to obtain the following inequality:

$$
\left\| \mathbb{E}_{\boldsymbol{\xi}^k} \left[ \widetilde{\nabla} f(x^{k+1}, \boldsymbol{\xi}^k) \right] - \nabla f(x^{k+1}) \right\|_2 \overset{(23),(59)}{=} \left\| \mathbb{E}_{\boldsymbol{\xi}^k} \left[ \left( \frac{\lambda_{k+1}}{\left\| \nabla f(x^{k+1}, \boldsymbol{\xi}^k) \right\|_2} - 1 \right) \nabla f(x^{k+1}, \boldsymbol{\xi}^k) \chi_k \right] \right\|_2
$$

$$
\leq \quad \mathbb{E}_{\boldsymbol{\xi}^k} \left[ \left\| \nabla f(x^{k+1}, \boldsymbol{\xi}^k) \right\|_2 \cdot \left| \frac{\lambda_{k+1}}{\left\| \nabla f(x^{k+1}, \boldsymbol{\xi}^k) \right\|_2} - 1 \right| \chi_k \right]
$$

$$
\overset{(56)}{=} \quad \mathbb{E}_{\boldsymbol{\xi}^k} \left[ \left\| \nabla f(x^{k+1}, \boldsymbol{\xi}^k) \right\|_2 \cdot \left( 1 - \frac{\lambda_{k+1}}{\left\| \nabla f(x^{k+1}, \boldsymbol{\xi}^k) \right\|_2} \right) \chi_k \right]
$$

$$
\overset{(56)}{\leq} \quad \mathbb{E}_{\boldsymbol{\xi}^k} \left[ \left\| \nabla f(x^{k+1}, \boldsymbol{\xi}^k) \right\|_2 \chi_k \right]
$$

$$
\overset{(57)}{\leq} \quad \mathbb{E}_{\boldsymbol{\xi}^k} \left[ \left\| \nabla f(x^{k+1}, \boldsymbol{\xi}^k) \right\|_2 \eta_k \right]
$$

$$
\leq \quad \mathbb{E}_{\boldsymbol{\xi}^k} \left[ \left\| \nabla f(x^{k+1}, \boldsymbol{\xi}^k) - \nabla f(x^{k+1}) \right\|_2 \eta_k \right]
$$
$$
+ \left\| \nabla f(x^{k+1}) \right\|_2 \mathbb{E}_{\boldsymbol{\xi}^k} [\eta_k]
$$

$$
\leq \quad \sqrt{ \mathbb{E}_{\boldsymbol{\xi}^k} \left[ \left\| \nabla f(x^{k+1}, \boldsymbol{\xi}^k) - \nabla f(x^{k+1}) \right\|_2^2 \right] \mathbb{E}_{\boldsymbol{\xi}^k} [\eta_k^2] }
$$
$$
+ \left\| \nabla f(x^{k+1}) \right\|_2 \mathbb{E}_{\boldsymbol{\xi}^k} [\eta_k]
$$

$$
\overset{(24)}{\leq} \quad \frac{\sigma}{\sqrt{m_k}} \sqrt{\mathbb{E}_{\boldsymbol{\xi}^k} [\eta_k^2]} + \frac{\lambda_{k+1}}{2} \mathbb{E}_{\boldsymbol{\xi}^k} [\eta_k]. \tag{60}
$$

Next, we derive an upper bound for the expectation of $\eta_k$ using Markov's inequality:

$$
\mathbb{E}_{\boldsymbol{\xi}^k} [\eta_k] = \mathbb{E}_{\boldsymbol{\xi}^k} [\eta_k^2] = \mathbb{P}_{\boldsymbol{\xi}^k} \{ \eta_k = 1 \}
$$
$$
\overset{(56)}{=} \mathbb{P}_{\boldsymbol{\xi}^k} \left\{ \left\| \nabla f(x^{k+1}, \boldsymbol{\xi}^k) - \nabla f(x^{k+1}) \right\|_2 > \frac{\lambda_{k+1}}{2} \right\}
$$
$$
\leq \frac{4 \mathbb{E}_{\boldsymbol{\xi}^k} \left[ \left\| \nabla f(x^{k+1}, \boldsymbol{\xi}^k) - \nabla f(x^{k+1}) \right\|_2^2 \right]}{\lambda_{k+1}^2} \overset{(24)}{\leq} \frac{4\sigma^2}{m_k \lambda_{k+1}^2}. \tag{61}
$$

Putting all together we derive (39):

$$
\left\| \mathbb{E}_{\boldsymbol{\xi}^k} \left[ \widetilde{\nabla} f(x^{k+1}, \boldsymbol{\xi}^k) \right] - \nabla f(x^{k+1}) \right\|_2 \overset{(60),(61)}{\leq} \frac{2\sigma^2}{m_k \lambda_{k+1}} + \frac{\lambda_{k+1}}{2} \cdot \frac{4\sigma^2}{m_k \lambda_{k+1}^2} = \frac{4\sigma^2}{m_k \lambda_{k+1}}.
$$

**Proof of** (40). Recall that in the space of random variables with finite second moment, i.e. in $L_2$, one can introduce a norm as $\sqrt{\mathbb{E}|X|^2}$ for an arbitrary random variable $X$ from this space. Using triangle

inequality for this norm we get

$$\sqrt{\mathbb{E}_{\boldsymbol{\xi}^k}\left[\left\|\nabla f(x^{k+1},\boldsymbol{\xi}^k)-\nabla f(x^{k+1})\right\|_2^2\right]} \overset{(58)}{\leq} \sqrt{\mathbb{E}_{\boldsymbol{\xi}^k}\left[\left\|\frac{\lambda_{k+1}\nabla f(x^{k+1},\boldsymbol{\xi}^k)}{\left\|\nabla f(x^{k+1},\boldsymbol{\xi}^k)\right\|_2}-\nabla f(x^{k+1})\right\|_2^2 \chi_k^2\right]}$$

$$+\sqrt{\mathbb{E}_{\boldsymbol{\xi}^k}\left[\left\|\nabla f(x^{k+1},\boldsymbol{\xi}^k)-\nabla f(x^{k+1})\right\|_2^2 (1-\chi_k)^2\right]}$$

$$\overset{(12)}{\leq} \sqrt{\mathbb{E}_{\boldsymbol{\xi}^k}\left[\left(2\left\|\frac{\lambda_{k+1}\nabla f(x^{k+1},\boldsymbol{\xi}^k)}{\left\|\nabla f(x^{k+1},\boldsymbol{\xi}^k)\right\|_2}\right\|_2^2+2\left\|\nabla f(x^{k+1})\right\|_2^2\right)\chi_k^2\right]}$$

$$+\sqrt{\mathbb{E}_{\boldsymbol{\xi}^k}\left[\left\|\nabla f(x^{k+1},\boldsymbol{\xi}^k)-\nabla f(x^{k+1})\right\|_2^2\right]}$$

$$\overset{(24)}{\leq} \sqrt{\frac{5}{2}}\lambda_{k+1}\sqrt{\mathbb{E}_{\boldsymbol{\xi}^k}\left[\chi_k^2\right]}+\frac{\sigma}{\sqrt{m_k}}$$

$$\overset{(57),(61)}{\leq} \sqrt{\frac{5}{2}}\lambda_{k+1}\cdot\frac{2\sigma}{\sqrt{m_k}\lambda_{k+1}}+\frac{\sigma}{\sqrt{m_k}}=\left(\sqrt{10}+1\right)\frac{\sigma}{\sqrt{m_k}}$$

$$\leq \frac{\sqrt{18}\sigma}{\sqrt{m_k}}.$$

**Proof of** (41). To derive (41) we use (40):

$$\mathbb{E}_{\boldsymbol{\xi}^k}\left[\left\|\widetilde{\nabla} f(x^{k+1},\boldsymbol{\xi}^k)-\mathbb{E}_{\boldsymbol{\xi}^k}\left[\widetilde{\nabla} f(x^{k+1},\boldsymbol{\xi}^k)\right]\right\|_2^2\right] \overset{(15)}{\leq} \mathbb{E}_{\boldsymbol{\xi}^k}\left[\left\|\widetilde{\nabla} f(x^{k+1},\boldsymbol{\xi}^k)-\nabla f(x^{k+1})\right\|_2^2\right]$$

$$\overset{(40)}{\leq} \frac{18\sigma^2}{m_k}.$$

### F.3.3 Proof of Theorem F.1

Lemma F.4 implies that the inequality

$$A_N\left(f(y^N)-f(x^*)\right) \leq \frac{1}{2}\|z^0-x^*\|_2^2-\frac{1}{2}\|z^N-x^*\|_2^2+\sum_{k=0}^{N-1}\alpha_{k+1}\left\langle\theta_{k+1},x^*-z^k\right\rangle$$

$$+\sum_{k=0}^{N-1}\alpha_{k+1}^2\|\theta_{k+1}\|_2^2+\sum_{k=0}^{N-1}\alpha_{k+1}^2\left\langle\theta_{k+1},\nabla f(x^{k+1})\right\rangle, \quad (62)$$

$$\theta_{k+1}\overset{\text{def}}{=}\widetilde{\nabla} f(x^{k+1},\boldsymbol{\xi}^k)-\nabla f(x^{k+1}) \quad (63)$$

holds for all $N\geq 0$. Taking into account that $f(y^N)-f(x^*)\geq 0$ for all $y^N$ and using new notation $R_k\overset{\text{def}}{=}\|z^k-x^*\|_2$, $\widetilde{R}_0=R_0$, $\widetilde{R}_{k+1}=\max\{\widetilde{R}_k,R_{k+1}\}$ we derive that for all $k\geq 0$

$$R_k^2\leq R_0^2+2\sum_{l=0}^{k-1}\alpha_{l+1}\left\langle\theta_{l+1},x^*-z^l\right\rangle+2\sum_{l=0}^{k-1}\alpha_{l+1}^2\left\langle\theta_{l+1},\nabla f(x^{l+1})\right\rangle+2\sum_{l=0}^{k-1}\alpha_{l+1}^2\|\theta_{l+1}\|_2^2. \quad (64)$$

First of all, we notice that for each $k\geq 0$ iterates $x^{k+1},z^k,y^k$ lie in the ball $B_{\widetilde{R}_k}(x^*)$. We prove it using induction. Since $y^0=z^0=x^0$, $\widetilde{R}_0=R_0=\|z^0-x^*\|_2$ and $x^1=\frac{A_0 y^0+\alpha_1 z^0}{A_1}=z^0$ we have that $x^1,z^0,y^0\in B_{\widetilde{R}_0}(x^*)$. Next, assume that $x^l,z^{l-1},y^{l-1}\in B_{\widetilde{R}_{l-1}}(x^*)$ for some $l\geq 1$. By definitions of $R_l$ and $\widetilde{R}_l$ we have that $z^l\in B_{R_l}(x^*)\subseteq B_{\widetilde{R}_l}(x^*)$. Since $y^l$ is a convex combination of $y^{l-1}\in B_{\widetilde{R}_{l-1}}(x^*)\subseteq B_{\widetilde{R}_l}(x^*)$, $z^l\in B_{\widetilde{R}_l}(x^*)$ and $B_{\widetilde{R}_l}(x^*)$ is a convex set we conclude that

$y^l \in B_{\widetilde{R}_l}(x^*)$. Finally, since $x^{l+1}$ is a convex combination of $y^l$ and $z^l$ we have that $x^{l+1}$ lies in $B_{\widetilde{R}_l}(x^*)$ as well.

The rest of the proof is based on the refined analysis of inequality (64). In particular, via induction we prove that for all $k = 0, 1, \ldots, N$ with probability at least $1 - \frac{k\beta}{N}$ the following statement holds: inequalities

$$
\begin{aligned}
R_t^2 &\overset{(64)}{\leq} R_0^2 + 2\sum_{l=0}^{t-1} \alpha_{l+1} \langle \theta_{l+1}, x^* - z^l \rangle + 2\sum_{l=0}^{t-1} \alpha_{l+1}^2 \langle \theta_{l+1}, \nabla f(x^{l+1}) \rangle + 2\sum_{l=0}^{t-1} \alpha_{k+1}^2 \|\theta_{l+1}\|_2^2 \\
&\leq C^2 R_0^2 \hspace{9cm} (65)
\end{aligned}
$$

hold for $t = 0, 1, \ldots, k$ simultaneously where $C$ is defined in (29). Let us define the probability event when this statement holds as $E_k$. Then, our goal is to show that $\mathbb{P}\{E_k\} \geq 1 - \frac{k\beta}{N}$ for all $k = 0, 1, \ldots, N$. For $t = 0$ inequality (65) holds with probability 1 since $C \geq 1$, hence $\mathbb{P}\{E_0\} = 1$. Next, assume that for some $k = T - 1 \leq N - 1$ we have $\mathbb{P}\{E_k\} = \mathbb{P}\{E_{T-1}\} \geq 1 - \frac{(T-1)\beta}{N}$. Let us prove that $\mathbb{P}\{E_T\} \geq 1 - \frac{T\beta}{N}$. First of all, probability event $E_{T-1}$ implies that

$$
\begin{aligned}
f(y^t) - f(x^*) &\overset{(62)}{\leq} \frac{1}{A_t} \left( \frac{1}{2} R_0^2 + \sum_{l=0}^{t-1} \alpha_{l+1} \langle \theta_{l+1}, x^* - z^l + \alpha_{l+1} \nabla f(x^{l+1}) \rangle + \sum_{l=0}^{t-1} \alpha_{k+1}^2 \|\theta_{l+1}\|_2^2 \right) \\
&\overset{(65)}{\leq} \frac{C^2 R_0^2}{2 A_t} \hspace{8cm} (66)
\end{aligned}
$$

hold for $t = 0, 1, \ldots, T - 1$. Then, inequalities

$$
\begin{aligned}
\|\nabla f(x^1)\|_2 &= \|\nabla f(z^0)\|_2 \overset{(6)}{\leq} L\|z^0 - x^*\|_2 = \frac{1}{a} \cdot \frac{R_0}{\alpha_1}, \\
\|\nabla f(x^{t+1})\|_2 &\leq \|\nabla f(x^{t+1}) - \nabla f(y^t)\|_2 + \|\nabla f(y^t)\|_2 \\
&\overset{(6),(8)}{\leq} L\|x^{t+1} - y^t\|_2 + \sqrt{2L(f(y^t) - f(x^*))} \\
&\overset{(55),(66)}{\leq} \frac{\alpha_{t+1} L}{A_t} \|x^{t+1} - z^k\|_2 + \sqrt{\frac{LC^2 R_0^2}{A_t}} \\
&\overset{(20)}{\leq} \frac{2L(t+2)}{t(t+3)} \left( \|x^{k+1} - x^*\|_2 + \|x^* - z^k\|_2 \right) + \frac{2LCR_0\sqrt{a}}{\sqrt{t(t+3)}} \\
&\leq \frac{4L(t+2)\widetilde{R}_k}{t(t+3)} + \frac{2LCR_0\sqrt{a}}{\sqrt{t(t+3)}} \\
&\overset{(65)}{\leq} \frac{2aLCR_0}{t+2} \left( \frac{2(t+2)^2}{at(t+3)} + \frac{t+2}{\sqrt{at(t+3)}} \right) \\
&\leq \frac{CR_0}{\alpha_{t+1}} \left( \frac{9}{2a} + \frac{3}{2\sqrt{a}} \right)
\end{aligned}
$$

hold for $t = 1, \ldots, T - 1$ where the last inequality follows from $\frac{(t+2)^2}{t(t+3)} \leq \frac{(1+2)^2}{1(1+3)} = \frac{9}{4}$. Taking $a$ such that

$$
a \geq \frac{2R_0}{B} \quad \text{and} \quad \frac{9}{2a} + \frac{3}{2\sqrt{a}} \leq \frac{B}{2CR_0}
$$

we obtain that probability event $E_{T-1}$ implies

$$
\|\nabla f(x^{t+1})\|_2 \leq \frac{B}{2\alpha_{t+1}} = \frac{\lambda_{t+1}}{2} \hspace{5cm} (67)
$$

for $t = 0, \ldots, T - 1$. Since $B = \frac{CR_0}{8 \ln \frac{4N}{\beta}}$ we have to choose such $a$ that

$$
a \geq \frac{16 \ln \frac{4N}{\beta}}{C} \quad \text{and} \quad \frac{9}{a} + \frac{3}{\sqrt{a}} \leq \frac{1}{8 \ln \frac{4N}{\beta}}.
$$

Solving quadratic inequality

$$a - 24\sqrt{a}\ln\frac{4N}{\beta} - 72\ln\frac{4N}{\beta} \geq 0$$

w.r.t. $\sqrt{a}$ we get that $a$ should satisfy

$$a \geq \max\left\{\frac{16\ln\frac{4N}{\beta}}{C}, 36\left(2\ln\frac{4N}{\beta} + \sqrt{4\ln^2\frac{4N}{\beta} + 2\ln\frac{4N}{\beta}}\right)^2\right\}.$$

Having inequalities (67) in hand we show in the rest of the proof that (65) holds for $t = T$ with big enough probability. First of all, we introduce new random variables:

$$\eta_l = \begin{cases} x^* - z^l, & \text{if } \|x^* - z^l\|_2 \leq CR_0, \\ 0, & \text{otherwise}, \end{cases} \quad \text{and} \quad \zeta_l = \begin{cases} \nabla f(x^{l+1}), & \text{if } \|\nabla f(x^{l+1})\|_2 \leq \frac{B}{2\alpha_{l+1}}, \\ 0, & \text{otherwise}, \end{cases} \tag{68}$$

for $l = 0, 1, \ldots T - 1$. Note that these random variables are bounded with probability 1, i.e. with probability 1 we have

$$\|\eta_l\|_2 \leq CR_0 \quad \text{and} \quad \|\zeta_l\|_2 \leq \frac{B}{2\alpha_{l+1}}. \tag{69}$$

Secondly, we use the introduced notation and get that $E_{T-1}$ implies

$$R_T^2 \overset{(64),(65),(67),(68)}{\leq} R_0^2 + 2\sum_{l=0}^{T-1}\alpha_{l+1}\langle\theta_{l+1}, \eta_l\rangle + 2\sum_{l=0}^{T-1}\alpha_{l+1}^2\|\theta_{l+1}\|_2^2 + 2\sum_{l=0}^{T-1}\alpha_{l+1}^2\langle\theta_{l+1}, \zeta_l\rangle$$

$$= R_0^2 + \sum_{l=0}^{T-1}\alpha_{l+1}\langle\theta_{l+1}, 2\eta_l + 2\alpha_{l+1}\zeta_l\rangle + 2\sum_{l=0}^{T-1}\alpha_{l+1}^2\|\theta_{l+1}\|_2^2.$$

Finally, we do some preliminaries in order to apply Bernstein's inequality (see Lemma D.1) and obtain that $E_{T-1}$ implies

$$R_T^2 \overset{(12)}{\leq} R_0^2 + \underbrace{\sum_{l=0}^{T-1}\alpha_{l+1}\langle\theta_{l+1}^u, 2\eta_l + 2\alpha_{l+1}\zeta_l\rangle}_{①} + \underbrace{\sum_{l=0}^{T-1}\alpha_{l+1}\langle\theta_{l+1}^b, 2\eta_l + 2\alpha_{l+1}\zeta_l\rangle}_{②}$$

$$+ \underbrace{\sum_{l=0}^{T-1}4\alpha_{l+1}^2\left(\|\theta_{l+1}^u\|_2^2 - \mathbb{E}_{\boldsymbol{\xi}^l}\left[\|\theta_{l+1}^u\|_2^2\right]\right)}_{③} + \underbrace{\sum_{l=0}^{T-1}4\alpha_{l+1}^2\mathbb{E}_{\boldsymbol{\xi}^l}\left[\|\theta_{l+1}^u\|_2^2\right]}_{④}$$

$$+ \underbrace{\sum_{l=0}^{T-1}4\alpha_{l+1}^2\|\theta_{l+1}^b\|_2^2}_{⑤} \tag{70}$$

where we introduce new notations:

$$\theta_{l+1}^u \overset{\text{def}}{=} \widetilde{\nabla}f(x^{l+1}, \boldsymbol{\xi}^l) - \mathbb{E}_{\boldsymbol{\xi}^l}\left[\widetilde{\nabla}f(x^{l+1}, \boldsymbol{\xi}^l)\right], \quad \theta_{l+1}^b \overset{\text{def}}{=} \mathbb{E}_{\boldsymbol{\xi}^l}\left[\widetilde{\nabla}f(x^{l+1}, \boldsymbol{\xi}^l)\right] - \nabla f(x^{l+1}), \tag{71}$$

$$\theta_{l+1} \overset{(37)}{=} \theta_{l+1}^u + \theta_{l+1}^b.$$

It remains to provide tight upper bounds for ①, ②, ③, ④ and ⑤, i.e. in the remaining part of the proof we show that $① + ② + ③ + ④ + ⑤ \leq \delta C^2 R_0^2$ for some $\delta < 1$.

**Upper bound for ①.** First of all, since $\mathbb{E}_{\boldsymbol{\xi}^l}[\theta_{l+1}^u] = 0$ summands in ① are conditionally unbiased:

$$\mathbb{E}_{\boldsymbol{\xi}^l}\left[\alpha_{l+1}\langle\theta_{l+1}^u, 2\eta_l + 2\alpha_{l+1}\zeta_l\rangle\right] = 0.$$

Secondly, these summands are bounded with probability 1:

$$
\begin{aligned}
\left|\alpha_{l+1}\left\langle\theta_{l+1}^u, 2\eta_l + 2\alpha_{l+1}\zeta_l\right\rangle\right| &\leq \alpha_{l+1}\|\theta_{l+1}^u\|_2\|2\eta_l + 2\alpha_{l+1}\zeta_l\|_2 \\
&\overset{(38),(69)}{\leq} 2\alpha_{l+1}\lambda_{l+1}\left(2CR_0 + B\right) = 2B(2CR_0 + B) \\
&= \frac{C^2 R_0^2}{2\ln\frac{4N}{\beta}} + \frac{C^2 R_0^2}{32\ln^2\frac{4N}{\beta}} \\
&\overset{(25)}{\leq} \frac{C^2 R_0^2}{2\ln\frac{4N}{\beta}} + \frac{C^2 R_0^2}{64\ln\frac{4N}{\beta}} \leq \frac{33 C^2 R_0^2}{64\ln\frac{4N}{\beta}}.
\end{aligned}
$$

Finally, one can bound conditional variances $\sigma_l^2 \overset{\text{def}}{=} \mathbb{E}_{\boldsymbol{\xi}^l}\left[\alpha_{l+1}^2\left\langle\theta_{l+1}^u, 2\eta_l + 2\alpha_{l+1}\zeta_l\right\rangle^2\right]$ in the following way:

$$
\begin{aligned}
\sigma_l^2 &\leq \mathbb{E}_{\boldsymbol{\xi}^l}\left[\alpha_{l+1}^2\left\|\theta_{l+1}^u\right\|_2^2\left\|2\eta_l + 2\alpha_{l+1}\zeta_l\right\|_2^2\right] \\
&\overset{(69)}{\leq} \alpha_{l+1}^2\mathbb{E}_{\boldsymbol{\xi}^l}\left[\left\|\theta_{l+1}^u\right\|_2^2\right](2CR_0 + B)^2. \quad (72)
\end{aligned}
$$

In other words, sequence $\left\{\alpha_{l+1}\left\langle\theta_{l+1}^u, 2\eta_l + 2\alpha_{l+1}\zeta_l\right\rangle\right\}_{l\geq 0}$ is bounded martingale difference sequence with bounded conditional variances $\{\sigma_l^2\}_{l\geq 0}$. Therefore, we can apply Bernstein's inequality, i.e. we apply Lemma D.1 with $X_l = \alpha_{l+1}\left\langle\theta_{l+1}^u, 2\eta_l + 2\alpha_{l+1}\zeta_l\right\rangle, c = \frac{33 C^2 R_0^2}{64\ln\frac{4N}{\beta}}$ and $F = \frac{c^2\ln\frac{4N}{\beta}}{18}$ and get that for all $b > 0$

$$
\mathbb{P}\left\{\left|\sum_{l=0}^{T-1} X_l\right| > b \text{ and } \sum_{l=0}^{T-1}\sigma_l^2 \leq F\right\} \leq 2\exp\left(-\frac{b^2}{2F + 2cb/3}\right)
$$

or, equivalently, with probability at least $1 - 2\exp\left(-\frac{b^2}{2F + 2cb/3}\right)$

$$
\text{either } \sum_{l=0}^{T-1}\sigma_l^2 > F \quad \text{or} \quad \underbrace{\left|\sum_{l=0}^{T-1} X_l\right|}_{|\textcircled{1}|} \leq b.
$$

The choice of $F$ will be clarified further, let us now choose $b$ in such a way that $2\exp\left(-\frac{b^2}{2F + 2cb/3}\right) = \frac{\beta}{2N}$. This implies that $b$ is the positive root of the quadratic equation

$$
b^2 - \frac{2c\ln\frac{4N}{\beta}}{3}b - 2F\ln\frac{4N}{\beta} = 0,
$$

hence

$$
\begin{aligned}
b &= \frac{c\ln\frac{4N}{\beta}}{3} + \sqrt{\frac{c^2\ln^2\frac{4N}{\beta}}{9} + 2F\ln\frac{4N}{\beta}} \leq \frac{c\ln\frac{4N}{\beta}}{3} + \sqrt{\frac{2c^2\ln^2\frac{4N}{\beta}}{9}} \\
&= \frac{1 + \sqrt{2}}{3}c\ln\frac{4N}{\beta} \leq \frac{33 C^2 R_0^2}{64}.
\end{aligned}
$$

That is, with probability at least $1 - \frac{\beta}{2N}$

$$
\underbrace{\text{either } \sum_{l=0}^{T-1}\sigma_l^2 > F \quad \text{or} \quad |\textcircled{1}| \leq \frac{33 C^2 R_0^2}{64}}_{\text{probability event } E_\textcircled{1}}.
$$

Next, we notice that probability event $E_{T-1}$ implies that

$$\sum_{l=0}^{T-1} \sigma_l^2 \overset{(72)}{\leq} (2CR_0+B)^2 \sum_{l=0}^{T-1} \alpha_{l+1}^2 \mathbb{E}_{\boldsymbol{\xi}^l}\left[\left\|\theta_{l+1}^u\right\|_2^2\right]$$

$$\overset{(41),(67)}{\leq} 18\sigma^2 C^2 R_0^2 \left(2 + \frac{1}{8\ln\frac{4N}{\beta}}\right)^2 \sum_{l=0}^{T-1} \frac{\alpha_{l+1}^2}{m_l}$$

$$\overset{(25),(26)}{\leq} 18\sigma^2 C^2 R_0^2 \left(2 + \frac{1}{16}\right)^2 \sum_{l=0}^{T-1} \frac{\alpha_{l+1}^2 C^2 R_0^2}{6000\sigma^2 \alpha_{l+1}^2 N \ln\frac{4N}{\beta}}$$

$$\overset{T\leq N}{\leq} \frac{18\left(2+\frac{1}{16}\right)^2}{6000\ln\frac{4N}{\beta}} C^4 R_0^4 \sum_{l=0}^{N-1} \frac{1}{N} \leq \frac{c^2\ln\frac{4N}{\beta}}{18} = F,$$

where the last inequality follows from $c = \frac{33C^2R_0^2}{64\ln\frac{4N}{\beta}}$ and simple arithmetic.

**Upper bound for ②.** First of all, we notice that probability event $E_{T-1}$ implies

$$\alpha_{l+1}\left\langle \theta_{l+1}^b, 2\eta_l + 2\alpha_{l+1}\zeta_l\right\rangle \leq \alpha_{l+1}\left\|\theta_{l+1}^b\right\|_2 \left\|2\eta_l + 2\alpha_{l+1}\zeta_l\right\|_2$$

$$\overset{(39),(69)}{\leq} \alpha_{l+1}\cdot\frac{4\sigma^2}{m_l\lambda_{l+1}}(2CR_0+B)$$

$$= \frac{32\alpha_{l+1}^2\sigma^2\ln\frac{4N}{\beta}}{m_l CR_0}\left(2CR_0 + \frac{CR_0}{8\ln\frac{4N}{\beta}}\right)$$

$$\overset{(25),(26)}{\leq} \frac{32\alpha_{l+1}^2\sigma^2 C^2 R_0^2\ln\frac{4N}{\beta}}{6000\alpha_{l+1}^2 N\sigma^2\ln\frac{4N}{\beta}}\left(2+\frac{1}{16}\right)$$

$$= \frac{11C^2R_0^2}{1000N}.$$

This implies that

$$② = \sum_{l=0}^{T-1} \alpha_{l+1}\left\langle\theta_{l+1}^b, 2\eta_l + 2\alpha_{l+1}\zeta_l\right\rangle \overset{T\leq N}{\leq} \frac{11C^2R_0^2}{1000}.$$

**Upper bound for ③.** We derive the upper bound for ③ using the same technique as for ①. First of all, we notice that the summands in ③ are conditionally independent:

$$\mathbb{E}_{\boldsymbol{\xi}^l}\left[4\alpha_{l+1}^2\left(\|\theta_{l+1}^u\|_2^2 - \mathbb{E}_{\boldsymbol{\xi}^l}\left[\|\theta_{l+1}^u\|_2^2\right]\right)\right] = 0.$$

Secondly, the summands are bounded with probability 1:

$$\left|4\alpha_{l+1}^2\left(\|\theta_{l+1}^u\|_2^2 - \mathbb{E}_{\boldsymbol{\xi}^l}\left[\|\theta_{l+1}^u\|_2^2\right]\right)\right| \leq 4\alpha_{l+1}^2\left(\|\theta_{l+1}^u\|_2^2 + \mathbb{E}_{\boldsymbol{\xi}^l}\left[\|\theta_{l+1}^u\|_2^2\right]\right)$$

$$\overset{(38)}{\leq} 4\alpha_{l+1}^2\left(4\lambda_{l+1}^2 + 4\lambda_{l+1}^2\right)$$

$$= 32B^2 = \frac{C^2R_0^2}{2\ln^2\frac{4N}{\beta}} \overset{(25)}{\leq} \frac{C^2R_0^2}{4\ln\frac{4N}{\beta}} \overset{\text{def}}{=} c_1. \qquad (73)$$

Finally, one can bound conditional variances $\hat{\sigma}_l^2 \overset{\text{def}}{=} \mathbb{E}_{\boldsymbol{\xi}^l}\left[\left|4\alpha_{l+1}^2\left(\|\theta_{l+1}^u\|_2^2 - \mathbb{E}_{\boldsymbol{\xi}^l}\left[\|\theta_{l+1}^u\|_2^2\right]\right)\right|^2\right]$ in the following way:

$$\hat{\sigma}_l^2 \overset{(73)}{\leq} c_1\mathbb{E}_{\boldsymbol{\xi}^l}\left[\left|4\alpha_{l+1}^2\left(\|\theta_{l+1}^u\|_2^2 - \mathbb{E}_{\boldsymbol{\xi}^l}\left[\|\theta_{l+1}^u\|_2^2\right]\right)\right|\right]$$

$$\leq 4c_1\alpha_{l+1}^2\mathbb{E}_{\boldsymbol{\xi}^l}\left[\|\theta_{l+1}^u\|_2^2 + \mathbb{E}_{\boldsymbol{\xi}^l}\left[\|\theta_{l+1}^u\|_2^2\right]\right] = 8c_1\alpha_{l+1}^2\mathbb{E}_{\boldsymbol{\xi}^l}\left[\|\theta_{l+1}^u\|_2^2\right]. \qquad (74)$$

In other words, sequence $\left\{4\alpha_{l+1}^2\left(\|\theta_{l+1}^u\|_2^2 - \mathbb{E}_{\boldsymbol{\xi}^l}\left[\|\theta_{l+1}^u\|_2^2\right]\right)\right\}_{l\geq 0}$ is bounded martingale difference sequence with bounded conditional variances $\{\hat{\sigma}_l^2\}_{l\geq 0}$. Therefore, we can apply Bernstein's inequality, i.e. we apply Lemma D.1 with $X_l = \hat{X}_l = 4\alpha_{l+1}^2\left(\|\theta_{l+1}^u\|_2^2 - \mathbb{E}_{\boldsymbol{\xi}^l}\left[\|\theta_{l+1}^u\|_2^2\right]\right)$, $c = c_1 = \frac{C^2R_0^2}{4\ln\frac{4N}{\beta}}$

and $F = F_1 = \frac{c_1^2 \ln \frac{4N}{\beta}}{18}$ and get that for all $b > 0$

$$\mathbb{P}\left\{\left|\sum_{l=0}^{T-1} \hat{X}_l\right| > b \text{ and } \sum_{l=0}^{T-1} \hat{\sigma}_l^2 \leq F_1\right\} \leq 2\exp\left(-\frac{b^2}{2F_1 + 2c_1 b/3}\right)$$

or, equivalently, with probability at least $1 - 2\exp\left(-\frac{b^2}{2F_1 + 2c_1 b/3}\right)$

$$\text{either} \quad \sum_{l=0}^{T-1} \hat{\sigma}_l^2 > F_1 \quad \text{or} \quad \underbrace{\left|\sum_{l=0}^{T-1} \hat{X}_l\right|}_{|③|} \leq b.$$

As in our derivations of the upper bound for ① we choose such $b$ that $2\exp\left(-\frac{b^2}{2F_1 + 2c_1 b/3}\right) = \frac{\beta}{2N}$, i.e.

$$b = \frac{c_1 \ln \frac{4N}{\beta}}{3} + \sqrt{\frac{c_1^2 \ln^2 \frac{4N}{\beta}}{9} + 2F_1 \ln \frac{4N}{\beta}} \leq \frac{1 + \sqrt{2}}{3} c_1 \ln \frac{4N}{\beta} \leq \frac{C^2 R_0^2}{4}.$$

That is, with probability at least $1 - \frac{\beta}{2N}$

$$\underbrace{\text{either} \quad \sum_{l=0}^{T-1} \hat{\sigma}_l^2 > F_1 \quad \text{or} \quad |③| \leq \frac{C^2 R_0^2}{4}}_{\text{probability event } E_③}.$$

Next, we notice that probability event $E_{T-1}$ implies that

$$\sum_{l=0}^{T-1} \hat{\sigma}_l^2 \overset{(74)}{\leq} 8c_1 \sum_{l=0}^{T-1} \alpha_{l+1}^2 \mathbb{E}_{\boldsymbol{\xi}^l}\left[\left\|\theta_{l+1}^u\right\|_2^2\right]$$

$$\overset{(41),(67)}{\leq} c_1 \sum_{l=0}^{T-1} \frac{144\sigma^2 \alpha_{l+1}^2}{m_l}$$

$$\overset{(26)}{\leq} c_1 \sum_{l=0}^{T-1} \frac{144\sigma^2 \alpha_{l+1}^2 C^2 R_0^2}{10368\sigma^2 \alpha_{l+1}^2 N}$$

$$\overset{T \leq N}{\leq} \underbrace{c_1 \cdot \frac{C^2 R_0^2}{4 \ln \frac{4N}{\beta}}}_{c_1} \cdot \frac{\ln \frac{4N}{\beta}}{18} = F_1.$$

**Upper bound for ④.** The probability event $E_{T-1}$ implies

$$④ = \sum_{l=0}^{T-1} 4\alpha_{l+1}^2 \mathbb{E}_{\boldsymbol{\xi}^l}\left[\left\|\theta_{l+1}^u\right\|_2^2\right] \overset{(41),(67)}{\leq} \sum_{l=0}^{T-1} \frac{72\alpha_{l+1}^2 \sigma^2}{m_l} \overset{(26)}{\leq} \sum_{l=0}^{T-1} \frac{72\alpha_{l+1}^2 \sigma^2 C^2 R_0^2}{10368\alpha_{l+1}^2 \sigma^2 N}$$

$$\overset{T \leq N}{\leq} \frac{C^2 R_0^2}{144}.$$

**Upper bound for ⑤.** Again, we use corollaries of probability event $E_{T-1}$:

$$⑤ = \sum_{l=0}^{T-1} 4\alpha_{l+1}^2 \|\theta_{l+1}^b\|_2^2 \overset{(39),(67)}{\leq} \sum_{l=0}^{T-1} \frac{64\alpha_{l+1}^2 \sigma^4}{m_l^2 \lambda_{l+1}^2} = \frac{1}{B^2} \sum_{l=0}^{T-1} \frac{64\alpha_{l+1}^4 \sigma^4}{m_l^2}$$

$$\overset{(26)}{\leq} \frac{64 \ln^2 \frac{4N}{\beta}}{C^2 R_0^2} \sum_{l=0}^{T-1} \frac{64\alpha_{l+1}^4 \sigma^4 C^4 R_0^4}{6000^2 \sigma^4 \alpha_{l+1}^4 N^2 \ln^2 \frac{4N}{\beta}}$$

$$\overset{T \leq N}{\leq} \frac{16C^2 R_0^2}{140625}.$$

Now we summarize all bound that we have: probability event $E_{T-1}$ implies

$$R_T^2 \overset{(64)}{\leq} R_0^2 + 2\sum_{l=0}^{T-1} \alpha_{l+1} \langle \theta_{l+1}, x^* - z^l \rangle + 2\sum_{l=0}^{k-1} \alpha_{l+1}^2 \langle \theta_{l+1}, \nabla f(x^{l+1}) \rangle + 2\sum_{l=0}^{T-1} \alpha_{l+1}^2 \|\theta_{l+1}\|_2^2$$

$$\overset{(70)}{\leq} R_0^2 + ① + ② + ③ + ④ + ⑤,$$

$$② \leq \frac{11C^2 R_0^2}{1000}, \quad ④ \leq \frac{C^2 R_0^2}{144}, \quad ⑤ \leq \frac{16C^2 R_0^2}{140625},$$

$$\sum_{l=0}^{T-1} \sigma_l^2 \leq F, \quad \sum_{l=0}^{T-1} \hat{\sigma}_l^2 \leq F_1$$

and

$$\mathbb{P}\{E_{T-1}\} \geq 1 - \frac{(T-1)\beta}{N}, \quad \mathbb{P}\{E_①\} \geq 1 - \frac{\beta}{2N}, \quad \mathbb{P}\{E_③\} \geq 1 - \frac{\beta}{2N},$$

where

$$E_① = \left\{ \text{either} \ \sum_{l=0}^{T-1} \sigma_l^2 > F \quad \text{or} \quad |①| \leq \frac{33C^2 R_0^2}{64} \right\},$$

$$E_③ = \left\{ \text{either} \ \sum_{l=0}^{T-1} \hat{\sigma}_l^2 > F_1 \quad \text{or} \quad |③| \leq \frac{C^2 R_0^2}{4} \right\}.$$

Taking into account these inequalities we get that probability event $E_{T-1} \cap E_① \cap E_③$ implies

$$R_T^2 \overset{(64)}{\leq} R_0^2 + 2\sum_{l=0}^{T-1} \alpha_{l+1} \langle \theta_{l+1}, x^* - z^l \rangle + 2\sum_{l=0}^{k-1} \alpha_{l+1}^2 \langle \theta_{l+1}, \nabla f(x^{l+1}) \rangle + 2\sum_{l=0}^{T-1} \alpha_{l+1}^2 \|\theta_{l+1}\|_2^2$$

$$\leq R_0^2 + \left( \frac{33}{64} + \frac{11}{1000} + \frac{1}{4} + \frac{1}{144} + \frac{16}{140625} \right) C^2 R_0^2$$

$$\leq \left( 1 + \frac{4}{5}C^2 \right) R_0^2 \overset{(29)}{\leq} C^2 R_0^2. \tag{75}$$

Moreover, using union bound we derive

$$\mathbb{P}\left\{ E_{T-1} \cap E_① \cap E_③ \right\} = 1 - \mathbb{P}\left\{ \overline{E}_{T-1} \cup \overline{E}_① \cup \overline{E}_③ \right\} \geq 1 - \frac{T\beta}{N}. \tag{76}$$

That is, by definition of $E_T$ and $E_{T-1}$ we have proved that

$$\mathbb{P}\{E_T\} \overset{(75)}{\geq} \mathbb{P}\left\{ E_{T-1} \cap E_① \cap E_③ \right\} \overset{(76)}{\geq} 1 - \frac{T\beta}{N},$$

which implies that for all $k = 0, 1, \ldots, N$ we have $\mathbb{P}\{E_k\} \geq 1 - \frac{k\beta}{N}$. Then, for $k = N$ we have that with probability at least $1 - \beta$

$$A_N \left( f(y^N) - f(x^*) \right) \overset{(62)}{\leq} \frac{1}{2}\|z^0 - z\|_2^2 - \frac{1}{2}\|z^N - z\|_2^2 + \sum_{k=0}^{N-1} \alpha_{k+1} \langle \theta_{k+1}, z - z^k \rangle$$

$$+ \sum_{k=0}^{N-1} \alpha_{k+1}^2 \|\theta_{k+1}\|_2^2 + \sum_{k=0}^{N-1} \alpha_{k+1}^2 \langle \theta_{k+1}, \nabla f(x^{k+1}) \rangle$$

$$\overset{(65)}{\leq} \frac{C^2 R_0^2}{2}.$$

Since $A_N = \frac{N(N+3)}{4aL}$ (see Lemma E.1) we get that with probability at least $1 - \beta$

$$f(y^N) - f(x^*) \leq \frac{2aLC^2 R_0^2}{N(N+3)}.$$

In other words, `clipped-SSTM` with $a = \max\left\{1, \frac{16\ln\frac{4N}{\beta}}{C}, 36\left(2\ln\frac{4N}{\beta} + \sqrt{4\ln^2\frac{4N}{\beta} + 2\ln\frac{4N}{\beta}}\right)^2\right\} = 36\left(2\ln\frac{4N}{\beta} + \sqrt{4\ln^2\frac{4N}{\beta} + 2\ln\frac{4N}{\beta}}\right)^2$ achieves $f(y^N) - f(x^*) \le \varepsilon$ with probability at least $1 - \beta$ after $O\left(\sqrt{\frac{LR_0^2}{\varepsilon}}\ln\frac{LR_0^2}{\varepsilon\beta}\right)$ iterations and requires

$$
\sum_{k=0}^{N-1} m_k \overset{(26)}{=} \sum_{k=0}^{N-1} O\left(\max\left\{1, \frac{\sigma^2\alpha_{k+1}^2 N\ln\frac{N}{\beta}}{R_0^2}\right\}\right)
$$

$$
= O\left(\max\left\{N, \sum_{k=0}^{N-1}\frac{\sigma^2(k+2)^2 N\ln\frac{N}{\beta}}{a^2 L^2 R_0^2}\right\}\right)
$$

$$
\overset{(27)}{=} O\left(\max\left\{N, \frac{\sigma^2 N^4}{\ln^3\frac{N}{\beta}L^2 R_0^2}\right\}\right)
$$

$$
= O\left(\max\left\{\sqrt{\frac{LR_0^2}{\varepsilon}}, \frac{\sigma^2 R_0^2}{\varepsilon^2}\right\}\ln\frac{LR_0^2}{\varepsilon\beta}\right).
$$

oracle calls.

### F.3.4 Proof of Corollary F.2

Theorem F.1 implies that with probability at least $1 - \beta$

$$
f(y^N) - f(x^*) \overset{(28)}{\le} \frac{2aLC^2 R_0^2}{N(N+3)}, \tag{77}
$$

where $a$ satisfies

$$
a \overset{(27)}{\ge} \max\left\{1, \frac{16\ln\frac{4N}{\beta}}{C}, 36\left(2\ln\frac{4N}{\beta} + \sqrt{4\ln^2\frac{4N}{\beta} + 2\ln\frac{4N}{\beta}}\right)^2\right\} \overset{\text{def}}{=} \hat{a}, \tag{78}
$$

$\alpha_{k+1} = \frac{k+2}{2aL}$ and batchsizes $m_k$ are chosen according to (26):

$$
m_k \overset{(26)}{=} \max\left\{1, \frac{1185\sigma^2\alpha_{k+1}^2 N\ln\frac{4N}{\beta}}{C^2 R_0^2}, \frac{10368\sigma^2\alpha_{k+1}^2 N}{C^2 R_0^2}\right\}
$$

$$
= \max\left\{1, \frac{1185\sigma^2(k+2)^2 N\ln\frac{4N}{\beta}}{4a^2 L^2 C^2 R_0^2}, \frac{10368\sigma^2(k+2)^2 N}{4a^2 L^2 C^2 R_0^2}\right\}. \tag{79}
$$

We consider two different options for $a$.

1. If $N\ln\frac{4N}{\beta}$ is bigger than $\hat{a}$, then we take $a = N\ln\frac{4N}{\beta}$ which implies that

$$
m_k = \max\left\{1, \frac{1185\sigma^2(k+2)^2}{4L^2 N C^2 R_0^2\ln\frac{4N}{\beta}}, \frac{10368\sigma^2(k+2)^2}{4L^2 C^2 R_0^2 N\ln^2\frac{4N}{\beta}}\right\} = O\left(\max\left\{1, \frac{\sigma^2(k+2)^2}{L^2 R_0^2 N\ln\frac{4N}{\beta}}\right\}\right)
$$

and with probability at least $1 - \beta$

$$
f(y^N) - f(x^*) \le \frac{2LC^2 R_0^2\ln\frac{4N}{\beta}}{N+3}. \tag{80}
$$

That is, if $\varepsilon$ is small enough to satisfy $\frac{LR_0^2}{\varepsilon}\ln\frac{LR_0^2}{\varepsilon\beta} \ge C_1\ln^2\frac{LR_0^2}{\varepsilon\beta}$ for some constant $C_1$, then due to (80) we have that after

$$
N = O\left(\frac{LR_0^2}{\varepsilon}\ln\frac{LR_0^2}{\varepsilon\beta}\right)\text{ iterations}
$$

of `clipped-SSTM` we obtain such point $y^N$ that with probability at least $1 - \beta$ inequality $f(y^N) - f(x^*) \le \varepsilon$ holds and the method requires

$$\sum_{k=0}^{N-1} m_k = \sum_{k=0}^{N-1} O\left(\max\left\{1, \frac{\sigma^2(k+2)^2}{L^2 R_0^2 N \ln \frac{4N}{\beta}}\right\}\right)$$

$$= O\left(\max\left\{N, \frac{\sigma^2 N^2}{L^2 R_0^2 \ln \frac{4N}{\beta}}\right\}\right) = O\left(\max\left\{\frac{LR_0^2}{\varepsilon}, \frac{\sigma^2 R_0^2}{\varepsilon^2}\right\} \ln \frac{LR_0^2}{\varepsilon\beta}\right)$$

stochastic first-order oracle calls.

2. If $a_0 N^{3/2}\sqrt{\ln \frac{4N}{\beta}}$ is bigger than $\hat{a}$ for some $a_0 > 0$, then we take $a = a_0 N^{3/2}\sqrt{\ln \frac{4N}{\beta}}$ which implies that

$$m_k = \max\left\{1, \frac{1185\sigma^2(k+2)^2}{4a_0^2 L^2 N^2 C^2 R_0^2}, \frac{10368\sigma^2(k+2)^2}{4a_0^2 L^2 C^2 R_0^2 N^2 \sqrt{\ln \frac{4N}{\beta}}}\right\} = O\left(\max\left\{1, \frac{\sigma^2(k+2)^2}{a_0^2 L^2 R_0^2 N^2}\right\}\right)$$

and with probability at least $1 - \beta$

$$f(y^N) - f(x^*) \le \frac{2a_0 L C^2 R_0^2 \sqrt{N \ln \frac{4N}{\beta}}}{N+3}. \tag{81}$$

That is, if $\varepsilon$ is small enough to satisfy $\frac{a_0^3 L^3 R_0^6}{\varepsilon^3}\left(\ln \frac{LR_0^2}{\varepsilon\beta}\right)^{3/2} \ge C_2 \ln^2 \frac{LR_0^2}{\varepsilon\beta}$ for some constant $C_2$, then due to (81) we have that after

$$N = O\left(\frac{a_0^2 L^2 R_0^4}{\varepsilon^2} \ln \frac{a_0^2 L^2 R_0^4}{\varepsilon^2 \beta}\right) = O\left(\frac{a_0^2 L^2 R_0^4}{\varepsilon^2} \ln \frac{a_0 L R_0^2}{\varepsilon\beta}\right) \text{ iterations}$$

of `clipped-SSTM` we obtain such point $y^N$ that with probability at least $1 - \beta$ inequality $f(y^N) - f(x^*) \le \varepsilon$ holds and the method requires

$$\sum_{k=0}^{N-1} m_k = \sum_{k=0}^{N-1} O\left(\max\left\{1, \frac{\sigma^2(k+2)^2}{a_0^2 L^2 R_0^2 N^2}\right\}\right)$$

$$= O\left(\max\left\{N, \frac{\sigma^2 N}{a_0^2 L^2 R_0^2}\right\}\right) = O\left(\max\left\{\frac{a_0^2 L^2 R_0^4}{\varepsilon^2}, \frac{\sigma^2 R_0^2}{\varepsilon^2}\right\} \ln \frac{a_0 L R_0^2}{\varepsilon\beta}\right)$$

stochastic first-order oracle calls. Finally, if all assumptions on $N$, $\beta$ and $\varepsilon$ hold for $a_0 = \frac{\sigma}{LR_0}$, then for all $k = 0, 1, \ldots, N-1$

$$m_k = O\left(\max\left\{1, \frac{\sigma^2(k+2)^2}{a_0^2 L^2 R_0^2 N^2}\right\}\right) = O\left(\max\left\{1, \frac{(k+2)^2}{N^2}\right\}\right) = O(1),$$

i.e. one iteration of `clipped-SSTM` requires $O(1)$ oracle calls, and $f(y^N) - f(x^*) \le \varepsilon$ with probability at least $1 - \beta$ after

$$N = O\left(\frac{\sigma^2 R_0^2}{\varepsilon^2} \ln \frac{\sigma R_0}{\varepsilon\beta}\right) \text{ iterations.}$$

### F.3.5   Proof of Corollary F.3

Recall that

$$a' = \max\left\{1, \frac{16 \ln \frac{4N}{\beta}}{C}, 36\left(2\ln \frac{4N}{\beta} + \sqrt{4\ln^2 \frac{4N}{\beta} + 2\ln \frac{4N}{\beta}}\right)^2\right\},$$

$$a = \max\left\{a', \frac{\sigma N^{3/2}}{LR_0}\sqrt{\ln \frac{4N}{\beta}}\right\}, \qquad \alpha_{k+1} = \frac{k+2}{2aL},$$

$$m_k = \max\left\{1, \frac{6000\sigma^2 \alpha_{k+1}^2 N \ln \frac{4N}{\beta}}{C^2 R_0^2}, \frac{10368\sigma^2 \alpha_{k+1}^2 N}{C^2 R_0^2}\right\}.$$

Since $a \ge \frac{\sigma N^{3/2}}{LR_0}$ we have that $m_k = O(1)$. Next, there are two possible situations.

1. If $a = a'$, then we are in the settings of Theorem F.1. This means that `clipped-SSTM` achieves $f(y^N) - f(x^*) \le \varepsilon$ with probability at least $1 - \beta$ after

$$O\left(\max\left\{\sqrt{\frac{LR_0^2}{\varepsilon}}, \frac{\sigma^2 R_0^2}{\varepsilon^2}\right\} \ln \frac{LR_0^2}{\varepsilon\beta}\right) \text{ oracle calls.}$$

2. If $a = \frac{\sigma N^{3/2}}{LR_0}\sqrt{\ln \frac{4N}{\beta}}$, then we are in the settings of Corollary F.2 which implies that `clipped-SSTM` achieves $f(y^N) - f(x^*) \le \varepsilon$ with probability at least $1 - \beta$ after

$$O\left(\frac{\sigma^2 R_0^2}{\varepsilon^2} \ln \frac{\sigma R_0}{\varepsilon\beta}\right) \text{ oracle calls.}$$

Finally, we combine these two cases and obtain that with $a = \max\left\{a', \frac{\sigma N^{3/2}}{LR_0}\sqrt{\ln \frac{4N}{\beta}}\right\}$ `clipped-SSTM` guarantees $f(y^N) - f(x^*) \le \varepsilon$ with probability at least $1 - \beta$ after

$$O\left(\max\left\{\max\left\{\sqrt{\frac{LR_0^2}{\varepsilon}}, \frac{\sigma^2 R_0^2}{\varepsilon^2}\right\} \ln \frac{LR_0^2}{\varepsilon\beta}, \frac{\sigma^2 R_0^2}{\varepsilon^2} \ln \frac{\sigma R_0}{\varepsilon\beta}\right\}\right)$$

$$= O\left(\max\left\{\sqrt{\frac{LR_0^2}{\varepsilon}}, \frac{\sigma^2 R_0^2}{\varepsilon^2}\right\} \ln \frac{LR_0^2 + \sigma R_0}{\varepsilon\beta}\right)$$

iterations/oracle calls.

### F.3.6 Proof of Theorem F.6

First of all, consider behavior of `clipped-SSTM` during the first run in `R-clipped-SSTM`. We notice that the proof of Theorem F.1 will be valid if we substitute $R_0$ everywhere by its upper bound $R$. From $\mu$-strong convexity of $f$ we have

$$R_0^2 = \|x^0 - x^*\|_2^2 \overset{(10)}{\le} \frac{2}{\mu}\left(f(x^0) - f(x^*)\right),$$

therefore, one can choose $R = \sqrt{\frac{2}{\mu}\left(f(x^0) - f(x^*)\right)}$. It implies that after $N_0$ iterations of `clipped-SSTM` we have

$$f(y^{N_0}) - f(x^*) \le \frac{2aC^2 LR^2}{N_0(N_0 + 3)} = \frac{4aC^2 L}{N_0^2 \mu}(f(x^0) - f(x^*)).$$

with probability at least $1 - \frac{\beta}{\tau}$, hence with the same probability $f(y^{N_0}) - f(x^*) \le \frac{1}{2}(f(x^0) - f(x^*))$ since $N_0 \ge C\sqrt{\frac{8aL}{\mu}}$. In other words, with probability at least $1 - \frac{\beta}{\tau}$

$$f(\hat{x}^1) - f(x^*) \le \frac{1}{2}\left(f(x^0) - f(x^*)\right) = \frac{1}{4}\mu R^2.$$

Then, by induction one can show that for arbitrary $k \in \{0, 1, \ldots, \tau - 1\}$ the inequality

$$f(\hat{x}^{k+1}) - f(x^*) \le \frac{1}{2}\left(f(\hat{x}^k) - f(x^*)\right)$$

holds with probability at least $1 - \frac{\beta}{\tau}$. Therefore, these inequalities hold simultaneously with probability at least $1 - \beta$. Using this we derive that inequality

$$f(\hat{x}^\tau) - f(x^*) \le \frac{1}{2}\left(f(\hat{x}^{\tau-1}) - f(x^*)\right) \le \frac{1}{2^2}\left(f(\hat{x}^{\tau-2}) - f(x^*)\right) \le \ldots \le \frac{1}{2^\tau}\left(f(x^0) - f(x^*)\right) = \frac{\mu R^2}{2^{\tau+1}}$$

holds with probability $\ge 1 - \beta$. That is, after $\tau = \left\lceil \log_2 \frac{\mu R^2}{2\varepsilon}\right\rceil$ restarts `R-clipped-SSTM` generates such a point $\hat{x}^\tau$ that $f(\hat{x}^\tau) - f(x^*) \le \varepsilon$ with probability at least $1 - \beta$. Moreover, if $a$ equals

the maximum from (45) and $N_0 \leq C_1 \sqrt{\frac{8aL}{\mu}}$ with some numerical constant $C_1 \geq C$, then $a \sim \left( \ln \frac{N_0 \tau}{\beta} \right)^2$, the total number of iterations of `clipped-SSTM` equals

$$N_0 \tau = O\left( \sqrt{\frac{L}{\mu}} \ln\left( \frac{\mu R^2}{\varepsilon} \right) \ln\left( \frac{L}{\mu\beta} \ln \frac{\mu R^2}{\varepsilon} \right) \right)$$

and the overall number of stochastic first-order oracle calls is

$$
\begin{aligned}
\sum_{t=0}^{\tau-1} \sum_{k=0}^{N_0-1} m_k^t &= \sum_{t=0}^{\tau-1} \sum_{k=0}^{N_0-1} O\left( \max\left\{ 1, \frac{2^t \sigma^2 \alpha_{k+1}^2 N_0 \ln \frac{4N_0\tau}{\beta}}{R^2} \right\} \right) \\
&= \sum_{t=0}^{\tau-1} \sum_{k=0}^{N_0-1} O\left( \max\left\{ 1, \frac{2^t \sigma^2 (k+2)^2 N_0}{\ln^3 \frac{4N_0\tau}{\beta} L^2 R^2} \right\} \right) \\
&= O\left( \max\left\{ N_0 \tau, \frac{\sigma^2 2^\tau N_0^4}{\ln^3 \frac{4N_0\tau}{\beta} L^2 R^2} \right\} \right) \\
&= O\left( \max\left\{ \sqrt{\frac{L}{\mu}} \ln\left( \frac{\mu R^2}{\varepsilon} \right), \frac{\sigma^2}{\mu\varepsilon} \right\} \ln\left( \frac{L}{\mu\beta} \ln \frac{\mu R^2}{\varepsilon} \right) \right).
\end{aligned}
$$

### F.3.7 Proof of Corollary F.7

Similarly to the proof of Theorem F.6 (see the previous subsection) we derive that under assumptions of the corollary after $\tau = \left\lceil \log_2 \frac{\mu R^2}{2\varepsilon} \right\rceil$ restarts `R-clipped-SSTM` generates such a point $\hat{x}^\tau$ that $f(\hat{x}^\tau) - f(x^*) \leq \varepsilon$ with probability at least $1 - \beta$. Moreover, $a$ and $N_0$ satisfy the following system of inequalities

$$a = \Theta\left( \frac{\sigma^4 \ln^2 \frac{N_0 \tau}{\beta}}{L\mu\varepsilon^2} \right), \quad N_0 = \Theta\left( \sqrt{\frac{aL}{\mu}} \right) \tag{82}$$

which is consistent and implies that

$$a = \Theta\left( \frac{\sigma^4}{L\mu\varepsilon} \ln^2\left( \frac{\sigma^2}{\mu\varepsilon\beta} \ln \frac{\mu R^2}{\varepsilon} \right) \right), \quad N_0 = \Theta\left( \frac{\sigma^2}{\mu\varepsilon} \ln\left( \frac{\sigma^2}{\mu\varepsilon\beta} \ln \frac{\mu R^2}{\varepsilon} \right) \right). \tag{83}$$

Then, for all $k = 0, 1, \ldots, N_0 - 1$ and $t = 0, 1, \ldots, \tau - 1$ batchsizes satisfy

$$
\begin{aligned}
m_k^t \leq m_{N_0-1}^{\tau-1} &= O\left( \max\left\{ 1, \frac{2^\tau \sigma^2 \alpha_{N_0}^2 N_0 \ln \frac{N_0\tau}{\beta}}{R^2} \right\} \right) \\
&= O\left( \max\left\{ 1, \frac{\mu R^2 \sigma^2 N_0^3 \ln \frac{N_0\tau}{\beta}}{a^2 L^2 \varepsilon R^2} \right\} \right) \overset{(82),(83)}{=} O(1),
\end{aligned}
$$

i.e. the algorithm requires $O(1)$ oracle calls per iteration. Finally, the total number of iterations is

$$N_0 \tau = O\left( \frac{\sigma^2}{\mu\varepsilon} \ln\left( \frac{\mu R^2}{\varepsilon} \right) \ln\left( \frac{\sigma^2}{\mu\varepsilon\beta} \ln \frac{\mu R^2}{\varepsilon} \right) \right).$$

# G  SGD with Clipping: Exact Formulations and Missing Proofs

In this section we provide exact formulations of all the results that we have for `clipped-SGD` and `R-clipped-SGD` together with the full proofs.

## G.1  Convex Case

We start with the case when $f(x)$ is convex and $L$-smooth and, as before, we assume that at each point $x \in \mathbb{R}^n$ function $f$ is accessible only via stochastic gradients $\nabla f(x, \xi)$ such that (2) holds. Next theorem summarizes the main convergence result for `clipped-SGD` in this case.

**Theorem G.1.** Assume that function $f$ is convex and $L$-smooth. Then for all $\beta \in (0, 1)$ and $N \geq 1$ such that

$$\ln \frac{4N}{\beta} \geq 2 \tag{84}$$

we have that after $N$ iterations of `clipped-SGD` with

$$\lambda = 2LCR_0, \quad m_k = m = \max \left\{ 1, \frac{27N\sigma^2}{2(CR_0)^2 L^2 \ln \frac{4N}{\beta}} \right\}, \tag{85}$$

where $R_0 = \|x^0 - x^*\|_2$ and stepsize

$$\gamma = \frac{1}{80L \ln \frac{4N}{\beta}}, \tag{86}$$

that with probability at least $1 - \beta$

$$f(\bar{x}^N) - f(x^*) \leq \frac{80LC^2 R_0^2 \ln \frac{4N}{\beta}}{N}, \tag{87}$$

where $\bar{x}^N = \frac{1}{N} \sum_{k=0}^{N-1} x^k$ and

$$C = \sqrt{2}. \tag{88}$$

In other words, the method achieves $f(\bar{x}^N) - f(x^*) \leq \varepsilon$ with probability at least $1 - \beta$ after $O\left( \frac{LR_0^2}{\varepsilon} \ln \frac{LR_0^2}{\varepsilon \beta} \right)$ iterations and requires

$$O\left( \max \left\{ \frac{LR_0^2}{\varepsilon}, \frac{\sigma^2 R_0^2}{\varepsilon^2} \right\} \ln \frac{LR_0^2}{\varepsilon \beta} \right) \text{ oracle calls.} \tag{89}$$

To the best of our knowledge, it is the first result for `clipped-SGD` establishing non-trivial complexity guarantees for the convergence with high probability. One can find the full proof in Section G.3.1.

## G.2  Strongly Convex Case

Next, we consider the situation when $f$ is additionally $\mu$-strongly convex and propose a restarted version of `clipped-SGD` (`R-clipped-SGD`), see Algorithm 4. For this method we prove the following

---

**Algorithm 4** Restarted Clipped Stochastic Gradient Descent (`R-clipped-SGD`)

---

**Input:** starting point $x^0$, number of iterations $N_0$ of `clipped-SGD`, number $\tau$ of `clipped-SGD` runs, batchsizes $m^0, m^1, \ldots, m^\tau$
1: Set $\hat{x}^0 = x^0$, stepsize $\gamma > 0$
2: **for** $t = 0, 1, \ldots, \tau - 1$ **do**
3:     Run `clipped-SGD` (Algorithm 2) for $N_0$ iterations with constant batchsizes $m^t$, stepsize $\gamma$ and starting point $\hat{x}^t$. Define the output of `clipped-SGD` by $\hat{x}^{t+1}$.
4: **end for**
**Output:** $\hat{x}^\tau$

---

result.

**Theorem G.2.** Assume that $f$ is $\mu$-strongly convex and $L$-smooth. If we choose $\beta \in (0, 1)$, $\tau$ and $N_0 \geq 1$ such that

$$\ln \frac{4N_0\tau}{\beta} \geq 2, \qquad \frac{N_0}{\ln \frac{4N_0\tau}{\beta}} \geq \frac{320C^2L}{\mu}, \tag{90}$$

and

$$m^t = \max\left\{1, \frac{27 \cdot 2^t N_0 \sigma^2}{2(CR)^2 L^2 \ln \frac{4N_0\tau}{\beta}}\right\}, \tag{91}$$

where $R = \sqrt{\frac{2(f(x^0) - f(x^*))}{\mu}}$ and $C = \sqrt{2}$, then we have that after $\tau$ runs of `clipped-SGD` in `R-clipped-SGD` the inequality

$$f(\hat{x}^\tau) - f(x^*) \leq 2^{-\tau}\left(f(x^0) - f(x^*)\right) \tag{92}$$

holds with probability at least $1 - \beta$. That is, if we choose $\frac{N_0}{\ln \frac{4N_0\tau}{\beta}} \leq \frac{C_1 L}{\mu}$ with some numerical constant $C_1 \geq 320C^2$, then the method achieves $f(\hat{x}^\tau) - f(x^*) \leq \varepsilon$ with probability at least $1 - \beta$ after

$$O\left(\frac{L}{\mu}\ln\left(\frac{\mu R^2}{\varepsilon}\right)\ln\left(\frac{L}{\mu\beta}\ln\frac{\mu R^2}{\varepsilon}\right)\right) \text{ iterations (in total)} \tag{93}$$

of `clipped-SGD` and requires

$$O\left(\max\left\{\frac{L}{\mu}\ln\frac{\mu R^2}{\varepsilon}, \frac{\sigma^2}{\mu\varepsilon}\right\}\ln\left(\frac{L}{\mu\beta}\ln\frac{\mu R^2}{\varepsilon}\right)\right) \text{ oracle calls.} \tag{94}$$

This theorem implies that `R-clipped-SGD` has the same complexity as the restarted version of `RSMD` from [47] up to the difference in logarithmical factors. We notice that the main difference between our result and one from [47] is that we do not need to assume that the optimization problem is considered on the bounded set.

However, in order to get (94) `R-clipped-SGD` requires to know strong convexity parameter $\mu$. In order to remove this drawback we analyse `clipped-SGD` for the strongly convex case and get the following result.

**Theorem G.3.** Assume that function $f$ is $\mu$-strongly convex and $L$-smooth. Then for all $\beta \in (0, 1)$ and $N \geq 1$ such that

$$\ln \frac{4N}{\beta} \geq 2 \tag{95}$$

we have that after $N$ iterations of `clipped-SGD` with

$$\lambda_l = 4\sqrt{L(1 - \gamma\mu)^l r_0}, \quad m_k = \max\left\{1, \frac{27N\sigma^2}{16Lr_0(1 - \gamma\mu)^k \ln \frac{4N}{\beta}}\right\}, \tag{96}$$

where $r_0 = f(x^0) - f(x^*)$ and stepsize

$$\gamma = \frac{1}{81L\ln\frac{4N}{\beta}}, \tag{97}$$

that with probability at least $1 - \beta$

$$f(x^N) - f(x^*) \leq 2(1 - \gamma\mu)^N(f(x^0) - f(x^*)). \tag{98}$$

In other words, the method achieves $f(x^N) - f(x^*) \leq \varepsilon$ with probability at least $1 - \beta$ after $O\left(\frac{L}{\mu}\ln\left(\frac{r_0}{\varepsilon}\right)\ln\left(\frac{L}{\mu\beta}\ln\frac{r_0}{\varepsilon}\right)\right)$ iterations and requires

$$O\left(\max\left\{\frac{L}{\mu}, \frac{\sigma^2}{\mu\varepsilon}\cdot\frac{L}{\mu}\right\}\ln\left(\frac{r_0}{\varepsilon}\right)\ln\left(\frac{L}{\mu\beta}\ln\frac{r_0}{\varepsilon}\right)\right) \text{ oracle calls.} \tag{99}$$

Unfortunately, our approach leads to worse complexity bound than we have for `R-clipped-SGD`: in the second term of the maximum in (99) we get an extra factor $L/\mu$ that can be large. Nevertheless, to

the best of our knowledge it is the first non-trivial complexity result for `clipped-SGD` that guarantees convergence with high probability. One can find the full proof of Theorem G.3 in Section G.3.3.

## G.3 Proofs

### G.3.1 Proof of Theorem G.1

Since $f(x)$ is convex and $L$-smooth, we get the following inequality:

$$
\begin{aligned}
\|x^{k+1} - x^*\|_2^2 &= \|x^k - \gamma \widetilde{\nabla} f(x^k, \boldsymbol{\xi}^k) - x^*\|_2^2 = \|x^k - x^*\|_2^2 + \gamma^2 \|\widetilde{\nabla} f(x^k, \boldsymbol{\xi}^k)\|_2^2 - 2\gamma \left\langle x^k - x^*, g^k \right\rangle \\
&= \|x^k - x^*\|_2^2 + \gamma^2 \|\nabla f(x^k) + \theta_k\|_2^2 - 2\gamma \left\langle x^k - x^*, \nabla f(x^k) + \theta_k \right\rangle \\
&\overset{(12)}{\leq} \|x^k - x^*\|_2^2 + 2\gamma^2 \|\nabla f(x^k)\|_2^2 + 2\gamma^2 \|\theta_k\|_2^2 - 2\gamma \left\langle x^k - x^*, \nabla f(x^k) + \theta_k \right\rangle \\
&\overset{(8)}{\leq} \|x^k - x^*\|_2^2 + 4\gamma^2 L \left( f(x^k) - f(x^*) \right) + 2\gamma^2 \|\theta_k\|_2^2 - 2\gamma \left\langle x^k - x^*, \nabla f(x^k) + \theta_k \right\rangle \\
&\leq \|x^k - x^*\|_2^2 + (4\gamma^2 L - 2\gamma) \left( f(x^k) - f(x^*) \right) + 2\gamma^2 \|\theta_k\|_2^2 - 2\gamma \left\langle x^k - x^*, \theta_k \right\rangle,
\end{aligned}
$$

where $\theta_k = \widetilde{\nabla} f(x^k, \boldsymbol{\xi}^k) - \nabla f(x^k)$ and the last inequality follows from the convexity of $f$. Using notation $R_k \overset{\text{def}}{=} \|x^k - x^*\|_2$ we derive that for all $k \geq 0$

$$
R_{k+1}^2 \leq R_k^2 + (4\gamma^2 L - 2\gamma) \left( f(x^k) - f(x^*) \right) + 2\gamma^2 \|\theta_k\|_2^2 - 2\gamma \left\langle x^k - x^*, \theta_k \right\rangle.
$$

Let us define $A = \left( 2\gamma - 4\gamma^2 L \right)$, then

$$
A \left( f(x^k) - f(x^*) \right) \leq R_k^2 - R_{k+1}^2 + 2\gamma^2 \|\theta_k\|_2^2 - 2\gamma \left\langle x^k - x^*, \theta_k \right\rangle.
$$

Summing up these inequalities for $k = 0, \ldots, N-1$ we obtain

$$
\begin{aligned}
\frac{A}{N} \sum_{k=0}^{N-1} \left[ f(x^k) - f(x^*) \right] &\leq \frac{1}{N} \sum_{k=0}^{N-1} \left( R_k^2 - R_{k+1}^2 \right) + \frac{2\gamma^2}{N} \sum_{k=0}^{N-1} \|\theta_k\|_2^2 - \frac{2\gamma^2}{N} \sum_{k=0}^{N-1} \left\langle x^k - x^*, \theta_k \right\rangle \\
&= \frac{1}{N} \left( R_0^2 - R_N^2 \right) + \frac{2\gamma^2}{N} \sum_{k=0}^{N-1} \|\theta_k\|_2^2 - \frac{2\gamma^2}{N} \sum_{k=0}^{N-1} \left\langle x^k - x^*, \theta_k \right\rangle.
\end{aligned}
$$

Noticing that for $\bar{x}^N = \frac{1}{N} \sum_{k=0}^{N-1} x^k$ Jensen's inequality gives $f(\bar{x}^N) = f \left( \frac{1}{N} \sum_{k=0}^{N-1} x^k \right) \leq \frac{1}{N} \sum_{k=0}^{N-1} f(x^k)$ we have

$$
AN \left( f(\bar{x}^N) - f(x^*) \right) \leq R_0^2 - R_N^2 + 2\gamma^2 \sum_{k=0}^{N-1} \|\theta_k\|_2^2 - 2\gamma \sum_{k=0}^{N-1} \left\langle x^k - x^*, \theta_k \right\rangle. \tag{100}
$$

Taking into account that $f(\bar{x}^N) - f(x^*) \geq 0$ and changing the indices we get that for all $k \geq 0$

$$
R_k^2 \leq R_0^2 + 2\gamma^2 \sum_{l=0}^{k-1} \|\theta_l\|_2^2 - 2\gamma \sum_{l=0}^{k-1} \left\langle x^l - x^*, \theta_k \right\rangle. \tag{101}
$$

The remaining part of the proof is based on the analysis of inequality (101). In particular, via induction we prove that for all $k = 0, 1, \ldots, N$ with probability at least $1 - \frac{k\beta}{N}$ the following statement holds: inequalities

$$
R_t^2 \overset{(101)}{\leq} R_0^2 + 2\gamma^2 \sum_{l=0}^{t-1} \|\theta_k\|_2^2 - 2\gamma \sum_{l=0}^{t-1} \left\langle x^k - x^*, \theta_k \right\rangle \leq C^2 R_0^2 \tag{102}
$$

hold for $t = 0, 1, \ldots, k$ simultaneously where $C$ is defined in (88). Let us define the probability event when this statement holds as $E_k$. Then, our goal is to show that $\mathbb{P}\{E_k\} \geq 1 - \frac{k\beta}{N}$ for all $k = 0, 1, \ldots, N$. For $t = 0$ inequality (102) holds with probability 1 since $C \geq 1$. Next, assume

that for some $k = T - 1 \leq N - 1$ we have $\mathbb{P}\{E_k\} = \mathbb{P}\{E_{T-1}\} \geq 1 - \frac{(T-1)\beta}{N}$. Let us prove that $\mathbb{P}\{E_T\} \geq 1 - \frac{T\beta}{N}$. First of all, probability event $E_{T-1}$ implies that

$$f(\bar{x}^N) - f(x^*) \overset{(100)}{\leq} \frac{1}{AN} \left( R_0^2 + 2\gamma^2 \sum_{k=0}^{N-1} \|\theta_k\|_2^2 - 2\gamma \sum_{k=0}^{N-1} \langle x^k - x^*, \theta_k \rangle \right) \overset{(102)}{\leq} \frac{C^2 R_0^2}{AN}$$

hold for $t = 0, 1, \ldots, T - 1$. Since $f$ is $L$-smooth, we have that probability event $E_{T-1}$ implies

$$\left\|\nabla f(x^t)\right\|_2 \leq L\|x^t - x^*\|_2 \leq LCR_0 = \frac{\lambda}{2} \tag{103}$$

for $t = 0, \ldots, T - 1$, where the clipping level is defined as

$$\lambda = 2LCR_0. \tag{104}$$

Having inequalities (103) in hand we show in the rest of the proof that (102) holds for $t = T$ with big enough probability. First of all, we introduce new random variables:

$$\eta_l = \begin{cases} x^* - z^l, & \text{if } \|x^* - z^l\|_2 \leq CR_0, \\ 0, & \text{otherwise}, \end{cases} \tag{105}$$

for $l = 0, 1, \ldots T - 1$. Note that these random variables are bounded with probability 1, i.e. with probability 1 we have

$$\|\eta_l\|_2 \leq CR_0. \tag{106}$$

Secondly, we use the introduced notation and get that $E_{T-1}$ implies

$$R_T^2 \overset{(101),(102),(103),(105)}{\leq} R_0^2 + 2\gamma \sum_{l=0}^{T-1} \langle \theta_l, \eta_l \rangle + 2\gamma^2 \sum_{l=0}^{T-1} \|\theta_{l+1}\|_2^2.$$

Finally, we do some preliminaries in order to apply Bernstein's inequality (see Lemma D.1) and obtain that $E_{T-1}$ implies

$$R_T^2 \overset{(12)}{\leq} R_0^2 + \underbrace{2\gamma \sum_{l=0}^{T-1} \langle \theta_l^u, \eta_l \rangle}_{\text{①}} + \underbrace{2\gamma \sum_{l=0}^{T-1} \langle \theta_l^b, \eta_l \rangle}_{\text{②}} + \underbrace{4\gamma^2 \sum_{l=0}^{T-1} \left( \|\theta_l^u\|_2^2 - \mathbb{E}_{\boldsymbol{\xi}^l}\left[\|\theta_l^u\|_2^2\right] \right)}_{\text{③}}$$

$$+ \underbrace{4\gamma^2 \sum_{l=0}^{T-1} \mathbb{E}_{\boldsymbol{\xi}^l}\left[\|\theta_l^u\|_2^2\right]}_{\text{④}} + \underbrace{4\gamma^2 \sum_{l=0}^{T-1} \|\theta_l^b\|_2^2}_{\text{⑤}} \tag{107}$$

where we introduce new notations:

$$\theta_l^u \overset{\text{def}}{=} \widetilde{\nabla} f(x^l, \boldsymbol{\xi}^l) - \mathbb{E}_{\boldsymbol{\xi}^l}\left[\widetilde{\nabla} f(x^l, \boldsymbol{\xi}^l)\right], \quad \theta_l^b \overset{\text{def}}{=} \mathbb{E}_{\boldsymbol{\xi}^l}\left[\widetilde{\nabla} f(x^l, \boldsymbol{\xi}^l)\right] - \nabla f(x^l), \tag{108}$$

$$\theta_l = \theta_l^u + \theta_l^b.$$

It remains to provide tight upper bounds for ①, ②, ③, ④ and ⑤, i.e. in the remaining part of the proof we show that $① + ② + ③ + ④ + ⑤ \leq \delta C^2 R_0^2$ for some $\delta < 1$.

**Upper bound for ①.** First of all, since $\mathbb{E}_{\boldsymbol{\xi}^l}[\theta_l^u] = 0$ summands in ① are conditionally unbiased:

$$\mathbb{E}_{\boldsymbol{\xi}^l}\left[2\gamma \langle \theta_l^u, \eta_l \rangle\right] = 0.$$

Secondly, these summands are bounded with probability 1:

$$|2\gamma \langle \theta_l^u, \eta_l \rangle| \leq 2\gamma\|\theta_l^u\|_2 \|\eta_l\|_2 \overset{(38),(106)}{\leq} 4\gamma\lambda CR_0 \overset{(104)}{=} 8\gamma(CR_0)^2 L.$$

Finally, one can bound conditional variances $\sigma_l^2 \overset{\text{def}}{=} \mathbb{E}_{\boldsymbol{\xi}^l}\left[4\gamma^2 \langle \theta_l^u, \eta_l \rangle^2\right]$ in the following way:

$$\sigma_l^2 \leq \mathbb{E}_{\boldsymbol{\xi}^l}\left[4\gamma^2 \|\theta_l^u\|_2^2 \|\eta_l\|_2^2\right] \overset{(106)}{\leq} 4\gamma^2(CR_0)^2 \mathbb{E}_{\boldsymbol{\xi}^l}\left[\|\theta_l^u\|_2^2\right].$$

In other words, sequence $\{2\gamma \langle \theta_l^u, \eta_l \rangle\}_{l \geq 0}$ is a bounded martingale difference sequence with bounded conditional variances $\{\sigma_l^2\}_{l \geq 0}$. Therefore, we can apply Bernstein's inequality, i.e. we apply Lemma D.1 with $X_l = 2\gamma \langle \theta_l^u, \eta_l \rangle$, $c = 8\gamma (CR_0)^2 L$ and $F = \frac{c^2 \ln \frac{4N}{\beta}}{6}$ and get that for all $b > 0$

$$\mathbb{P}\left\{ \left| \sum_{l=0}^{T-1} X_l \right| > b \text{ and } \sum_{l=0}^{T-1} \sigma_l^2 \leq F \right\} \leq 2\exp\left( -\frac{b^2}{2F + 2cb/3} \right)$$

or, equivalently, with probability at least $1 - 2\exp\left( -\frac{b^2}{2F + 2cb/3} \right)$

$$\text{either } \sum_{l=0}^{T-1} \sigma_l^2 > F \quad \text{or} \quad \underbrace{\left| \sum_{l=0}^{T-1} X_l \right| \leq b}_{|\text{①}|}.$$

The choice of $F$ will be clarified further, let us now choose $b$ in such a way that $2\exp\left( -\frac{b^2}{2F + 2cb/3} \right) = \frac{\beta}{2N}$. This implies that $b$ is the positive root of the quadratic equation

$$b^2 - \frac{2c \ln \frac{4N}{\beta}}{3} b - 2F \ln \frac{4N}{\beta} = 0,$$

hence

$$\begin{aligned}
b &= \frac{c \ln \frac{4N}{\beta}}{3} + \sqrt{\frac{c^2 \ln^2 \frac{4N}{\beta}}{9} + 2F \ln \frac{4N}{\beta}} = \frac{c \ln \frac{4N}{\beta}}{3} + \sqrt{\frac{4c^2 \ln^2 \frac{4N}{\beta}}{9}} \\
&= c \ln \frac{4N}{\beta} = 8\gamma (CR_0)^2 L \ln \frac{4N}{\beta}.
\end{aligned}$$

That is, with probability at least $1 - \frac{\beta}{2N}$

$$\underbrace{\text{either } \sum_{l=0}^{T-1} \sigma_l^2 > F \quad \text{or} \quad |\text{①}| \leq 8\gamma (CR_0)^2 L \ln \frac{4N}{\beta}}_{\text{probability event } E_{\text{①}}}.$$

Next, we notice that probability event $E_{T-1}$ implies that

$$\begin{aligned}
\sum_{l=0}^{T-1} \sigma_l^2 &\leq 4\gamma^2 (CR_0)^2 \sum_{l=0}^{T-1} \mathbb{E}_{\xi^l} \left[ \|\theta_l^u\|_2^2 \right] \overset{(41)}{\leq} 72\gamma^2 (CR_0)^2 \sigma^2 \frac{T}{m} \\
&\overset{(85)}{\leq} 72\gamma^2 (CR_0)^2 \sigma^2 \frac{2T(CR_0)^2 L^2 \ln \frac{4N}{\beta}}{27N\sigma^2} \\
&\overset{T \leq N}{\leq} \frac{16}{3} \gamma^2 (CR_0)^4 L^2 \ln \frac{4N}{\beta} \leq \frac{c^2 \ln \frac{4N}{\beta}}{6} = F,
\end{aligned}$$

where the last inequality follows from $c = 8\gamma (CR_0)^2 L$ and simple arithmetic.

**Upper bound for ②.** First of all, we notice that probability event $E_{T-1}$ implies

$$2\gamma \langle \theta_l^b, \eta_l \rangle \leq 2\gamma \|\theta_l^b\|_2 \|\eta_l\|_2 \overset{(39),(106)}{\leq} 2\gamma \frac{4\sigma^2}{m\lambda} CR_0 \overset{(104)}{=} \frac{4\gamma\sigma^2}{Lm}.$$

This implies that

$$\text{②} = 2\gamma \sum_{l=0}^{T-1} \langle \theta_l^b, \eta_l \rangle \overset{T \leq N}{\leq} \frac{4\gamma N \sigma^2}{mL}.$$

**Upper bound for ③.** We derive the upper bound for ③ using the same technique as for ①. First of all, we notice that the summands in ③ are conditionally independent:

$$\mathbb{E}_{\xi^l} \left[ 4\gamma^2 \left( \|\theta_l^u\|_2^2 - \mathbb{E}_{\xi^l} \left[ \|\theta_l^u\|_2^2 \right] \right) \right] = 0.$$

Secondly, the summands are bounded with probability 1:

$$\left|4\gamma^2\left(\|\theta_l^u\|_2^2 - \mathbb{E}_{\boldsymbol{\xi}^l}\left[\|\theta_l^u\|_2^2\right]\right)\right| \leq 4\gamma^2\left(\|\theta_l^u\|_2^2 + \mathbb{E}_{\boldsymbol{\xi}^l}\left[\|\theta_l^u\|_2^2\right]\right) \overset{(38)}{\leq} 4\gamma^2\left(4\lambda^2 + 4\lambda^2\right)$$
$$\overset{(104)}{=} 128\gamma^2(CR_0)^2L^2 \overset{\text{def}}{=} c_1. \tag{109}$$

Finally, one can bound conditional variances $\hat{\sigma}_l^2 \overset{\text{def}}{=} \mathbb{E}_{\boldsymbol{\xi}^l}\left[\left|4\gamma^2\left(\|\theta_l^u\|_2^2 - \mathbb{E}_{\boldsymbol{\xi}^l}\left[\|\theta_l^u\|_2^2\right]\right)\right|^2\right]$ in the following way:

$$\hat{\sigma}_l^2 \overset{(109)}{\leq} c_1\mathbb{E}_{\boldsymbol{\xi}^l}\left[\left|4\gamma^2\left(\|\theta_l^u\|_2^2 - \mathbb{E}_{\boldsymbol{\xi}^l}\left[\|\theta_l^u\|_2^2\right]\right)\right|\right]$$
$$\leq 4\gamma^2 c_1\mathbb{E}_{\boldsymbol{\xi}^l}\left[\|\theta_l^u\|_2^2 + \mathbb{E}_{\boldsymbol{\xi}^l}\left[\|\theta_l^u\|_2^2\right]\right] = 8\gamma^2 c_1\mathbb{E}_{\boldsymbol{\xi}^l}\left[\|\theta_l^u\|_2^2\right]. \tag{110}$$

In other words, sequence $\left\{4\gamma^2\left(\|\theta_l^u\|_2^2 - \mathbb{E}_{\boldsymbol{\xi}^l}\left[\|\theta_l^u\|_2^2\right]\right)\right\}_{l\geq 0}$ is a bounded martingale difference sequence with bounded conditional variances $\{\hat{\sigma}_l^2\}_{l\geq 0}$. Therefore, we can apply Bernstein's inequality, i.e. we apply Lemma D.1 with $X_l = \hat{X}_l = 4\gamma^2\left(\|\theta_l^u\|_2^2 - \mathbb{E}_{\boldsymbol{\xi}^l}\left[\|\theta_l^u\|_2^2\right]\right)$, $c = c_1 = 128\gamma^2(CR_0)^2L^2$ and $F = F_1 = \frac{c_1^2\ln\frac{4N}{\beta}}{6}$ and get that for all $b > 0$

$$\mathbb{P}\left\{\left|\sum_{l=0}^{T-1}\hat{X}_l\right| > b \text{ and } \sum_{l=0}^{T-1}\hat{\sigma}_l^2 \leq F_1\right\} \leq 2\exp\left(-\frac{b^2}{2F_1 + 2c_1b/3}\right)$$

or, equivalently, with probability at least $1 - 2\exp\left(-\frac{b^2}{2F_1 + 2c_1b/3}\right)$

$$\text{either } \sum_{l=0}^{T-1}\hat{\sigma}_l^2 > F_1 \quad \text{or} \quad \underbrace{\left|\sum_{l=0}^{T-1}\hat{X}_l\right|}_{|\circled{3}|} \leq b.$$

As in our derivations of the upper bound for $\circled{1}$ we choose such $b$ that $2\exp\left(-\frac{b^2}{2F_1 + 2c_1b/3}\right) = \frac{\beta}{2N}$, i.e.

$$b = \frac{c_1\ln\frac{4N}{\beta}}{3} + \sqrt{\frac{c_1^2\ln^2\frac{4N}{\beta}}{9} + 2F_1\ln\frac{4N}{\beta}} = c_1\ln\frac{4N}{\beta} = 128\gamma^2(CR_0)^2L^2\ln\frac{4N}{\beta}.$$

That is, with probability at least $1 - \frac{\beta}{2N}$

$$\underbrace{\text{either } \sum_{l=0}^{T-1}\hat{\sigma}_l^2 > F_1 \quad \text{or} \quad |\circled{3}| \leq 128\gamma^2(CR_0)^2L^2\ln\frac{4N}{\beta}}_{\text{probability event } E_{\circled{3}}}.$$

Next, we notice that probability event $E_{T-1}$ implies that

$$\sum_{l=0}^{T-1}\hat{\sigma}_l^2 \overset{(110)}{\leq} 8\gamma^2 c_1\sum_{l=0}^{T-1}\mathbb{E}_{\boldsymbol{\xi}^l}\left[\|\theta_l^u\|_2^2\right] \overset{(41)}{\leq} 144\gamma^2 c_1\sigma^2\frac{T}{m}$$
$$\overset{(85)}{\leq} \frac{32}{3}\gamma^2 c_1(CR_0)^2L^2\frac{T}{N}\ln\frac{4N}{\beta}$$
$$\overset{T\leq N}{\leq} \frac{c_1^2\ln\frac{4N}{\beta}}{6} \leq F_1.$$

**Upper bound for $\circled{4}$.** The probability event $E_{T-1}$ implies

$$\circled{4} = 4\gamma^2\sum_{l=0}^{T-1}\mathbb{E}_{\boldsymbol{\xi}^l}\left[\|\theta_l^u\|_2^2\right] \overset{(41)}{\leq} 72\gamma^2\sigma^2\sum_{l=0}^{T-1}\frac{1}{m} \overset{T\leq N}{\leq} \frac{72\gamma^2 N\sigma^2}{m}.$$

**Upper bound for ⑤.** Again, we use corollaries of probability event $E_{T-1}$:

$$⑤ \ = \ 4\gamma^2 \sum_{l=0}^{T-1} \|\theta_l^b\|_2^2 \ \overset{(39)}{\leq} \ 64\gamma^2\sigma^4 \frac{T}{m^2\lambda^2} \ \overset{(104)}{=} \ \frac{64\gamma^2\sigma^4}{4(CR_0)^2L^2} \cdot \frac{T}{m^2} \ \overset{T\leq N}{\leq} \ \frac{16\gamma^2N\sigma^4}{(CR_0)^2L^2m^2}.$$

Now we summarize all bound that we have: probability event $E_{T-1}$ implies

$$R_T^2 \ \overset{(101)}{\leq} \ R_0^2 + 2\gamma^2 \sum_{l=0}^{T-1} \|\theta_l\|_2^2 - 2\gamma \sum_{l=0}^{T-1} \langle x^l - x^*, \theta_l \rangle$$

$$\overset{(107)}{\leq} \ R_0^2 + ① + ② + ③ + ④ + ⑤,$$

$$② \ \leq \ \frac{4\gamma N\sigma^2}{mL}, \quad ④ \leq \frac{72\gamma^2 N\sigma^2}{m}, \quad ⑤ \leq \frac{16\gamma^2 N\sigma^4}{(CR_0)^2L^2m^2},$$

$$\sum_{l=0}^{T-1} \sigma_l^2 \ \leq \ F, \quad \sum_{l=0}^{T-1} \hat\sigma_l^2 \leq F_1$$

and

$$\mathbb{P}\{E_{T-1}\} \geq 1 - \frac{(T-1)\beta}{N}, \quad \mathbb{P}\{E_①\} \geq 1 - \frac{\beta}{2N}, \quad \mathbb{P}\{E_③\} \geq 1 - \frac{\beta}{2N},$$

where

$$E_① \ = \ \left\{ \text{either } \sum_{l=0}^{T-1} \sigma_l^2 > F \quad \text{or} \quad |①| \leq 8\gamma(CR_0)^2L\ln\frac{4N}{\beta} \right\},$$

$$E_③ \ = \ \left\{ \text{either } \sum_{l=0}^{T-1} \hat\sigma_l^2 > F_1 \quad \text{or} \quad |③| \leq 128\gamma^2(CR_0)^2L^2\ln\frac{4N}{\beta} \right\}.$$

Taking into account these inequalities and our assumptions on $m$ and $\gamma$ (see (85) and (86)) we get that probability event $E_{T-1} \cap E_① \cap E_③$ implies

$$R_T^2 \ \overset{(101)}{\leq} \ R_0^2 + 2\gamma^2 \sum_{l=0}^{T-1} \|\theta_l\|_2^2 - 2\gamma \sum_{l=0}^{T-1} \langle x^l - x^*, \theta_l \rangle$$

$$\leq \ R_0^2 + \left( \frac{1}{10} + \frac{1}{10} + \frac{1}{10} + \frac{1}{10} + \frac{1}{10} \right) C^2 R_0^2 \leq \left( 1 + \frac{1}{2}C^2 \right) R_0^2 \overset{(88)}{\leq} C^2 R_0^2. \quad (111)$$

Moreover, using union bound we derive

$$\mathbb{P}\left\{ E_{T-1} \cap E_① \cap E_③ \right\} = 1 - \mathbb{P}\left\{ \overline{E}_{T-1} \cup \overline{E}_① \cup \overline{E}_③ \right\} \geq 1 - \frac{T\beta}{N}. \quad (112)$$

That is, by definition of $E_T$ and $E_{T-1}$ we have proved that

$$\mathbb{P}\{E_T\} \ \overset{(111)}{\geq} \ \mathbb{P}\left\{ E_{T-1} \cap E_① \cap E_③ \right\} \overset{(112)}{\geq} 1 - \frac{T\beta}{N},$$

which implies that for all $k = 0, 1, \ldots, N$ we have $\mathbb{P}\{E_k\} \geq 1 - \frac{k\beta}{N}$. Then, for $k = N$ we have that with probability at least $1 - \beta$

$$ANf(\bar x^N) - f(x^*) \ \overset{(100)}{\leq} \ R_0^2 + 2\gamma^2 \sum_{k=0}^{N-1} \|\theta_k\|_2^2 - 2\gamma \sum_{k=0}^{N-1} \langle x^k - x^*, \theta_k \rangle \overset{(102)}{\leq} C^2 R_0^2.$$

Since $A = 2\gamma(1 - 2\gamma L)$ and $1 - \gamma L \geq \frac{1}{2}$ we get that with probability at least $1 - \beta$

$$f(\bar x^N) - f(x^*) \ \leq \ \frac{C^2 R_0^2}{AN} \leq \frac{C^2 R_0^2}{\gamma N} \overset{(86)}{\leq} \frac{80 C^2 R_0^2 L \ln\frac{4N}{\beta}}{N}.$$

In other words, `clipped-SGD` achieves $f(\bar{x}^N) - f(x^*) \le \varepsilon$ with probability at least $1 - \beta$ after $O\left(\frac{LR_0^2}{\varepsilon} \ln \frac{LR_0^2}{\varepsilon\beta}\right)$ iterations and requires

$$\sum_{k=0}^{N-1} m_k \overset{(85)}{=} \sum_{k=0}^{N-1} O\left(\max\left\{1, \frac{N\sigma^2}{C^2 R_0^2 L^2 \ln \frac{N}{\beta}}\right\}\right) = O\left(\max\left\{N, \frac{N^2\sigma^2}{C^2 R_0^2 L^2 \ln \frac{N}{\beta}}\right\}\right)$$
$$= O\left(\max\left\{\frac{LR_0^2}{\varepsilon}, \frac{\sigma^2 R_0^2}{\varepsilon^2}\right\} \ln \frac{LR_0^2}{\varepsilon\beta}\right)$$

oracle calls.

### G.3.2  Proof of Theorem G.2

First of all, consider behavior of `clipped-SGD` during the first run in `R-clipped-SGD`. We notice that the proof of Theorem G.1 will be valid if we substitute $R_0$ everywhere by its upper bound $R$. From $\mu$-strong convexity of $f$ we have

$$R_0^2 = \|x^0 - x^*\|_2^2 \overset{(10)}{\le} \frac{2}{\mu}\left(f(x^0) - f(x^*)\right),$$

therefore, one can choose $R = \sqrt{\frac{2}{\mu}\left(f(x^0) - f(x^*)\right)}$. It implies that after $N_0$ iterations of `clipped-SGD` we have

$$f(\bar{x}^{N_0}) - f(x^*) \le \frac{80LC^2 R^2 \ln \frac{4N_0\tau}{\beta}}{N_0} = \frac{160LC^2 R^2 \ln \frac{4N_0\tau}{\beta}}{N_0\mu}(f(x^0) - f(x^*)).$$

with probability at least $1 - \frac{\beta}{\tau}$, hence with the same probability $f(\bar{x}^{N_0}) - f(x^*) \le \frac{1}{2}(f(x^0) - f(x^*))$ since $\frac{N_0}{\ln \frac{4N_0\tau}{\beta}} \ge \frac{320C^2 L}{\mu}$. In other words, with probability at least $1 - \frac{\beta}{\tau}$

$$f(\hat{x}^1) - f(x^*) \le \frac{1}{2}\left(f(x^0) - f(x^*)\right) = \frac{1}{4}\mu R^2.$$

Then, by induction one can show that for arbitrary $k \in \{0, 1, \ldots, \tau - 1\}$ the inequality

$$f(\hat{x}^{k+1}) - f(x^*) \le \frac{1}{2}\left(f(\hat{x}^k) - f(x^*)\right)$$

holds with probability at least $1 - \frac{\beta}{\tau}$. Therefore, these inequalities hold simultaneously with probability at least $1 - \beta$. Using this we derive that inequality

$$f(\hat{x}^\tau) - f(x^*) \le \frac{1}{2}\left(f(\hat{x}^{\tau-1}) - f(x^*)\right) \le \frac{1}{2^2}\left(f(\hat{x}^{\tau-2}) - f(x^*)\right) \le \ldots \le \frac{1}{2^\tau}\left(f(x^0) - f(x^*)\right)$$
$$= \frac{\mu R^2}{2^{\tau+1}}$$

holds with probability $\ge 1 - \beta$. That is, after $\tau = \left\lceil \log_2 \frac{\mu R^2}{2\varepsilon} \right\rceil$ restarts `R-clipped-SGD` generates such point $\hat{x}^\tau$ that $f(\hat{x}^\tau) - f(x^*) \le \varepsilon$ with probability at least $1 - \beta$. Moreover, if $\frac{N_0}{\ln \frac{4N_0\tau}{\beta}} \le \frac{C_1 L}{\mu}$ with some numerical constant $C_1 \ge 320C^2$, then the total number of iterations of `clipped-SGD` equals

$$N_0\tau = O\left(\frac{L}{\mu} \ln\left(\frac{\mu R^2}{\varepsilon}\right) \ln\left(\frac{L}{\mu\beta} \ln \frac{\mu R^2}{\varepsilon}\right)\right)$$

and the overall number of stochastic first-order oracle calls is

$$\sum_{t=0}^{\tau-1} N_0 m^t = \sum_{t=0}^{\tau-1} O\left(\max\left\{N_0, \frac{2^t N_0^2 \sigma^2}{R^2 L^2 \ln \frac{4N_0\tau}{\beta}}\right\}\right)$$
$$= O\left(\max\left\{N_0\tau, \sum_{t=0}^{\tau-1} \frac{2^t N_0^2 \sigma^2}{R^2 L^2 \ln \frac{4N_0\tau}{\beta}}\right\}\right)$$
$$= O\left(\max\left\{\frac{L}{\mu} \ln\left(\frac{\mu R^2}{\varepsilon}\right), \frac{\sigma^2}{\mu\varepsilon}\right\} \ln\left(\frac{L}{\mu\beta} \ln \frac{\mu R^2}{\varepsilon}\right)\right).$$

### G.3.3 Proof of Theorem G.3

Since $f$ is $L$-smooth we have

$$
\begin{aligned}
f(x^{k+1}) &\leq f(x^k) - \gamma \left\langle \nabla f(x^k), \widetilde{\nabla} f(x^k, \boldsymbol{\xi}^k) \right\rangle + \frac{L\gamma^2}{2} \|\widetilde{\nabla} f(x^k, \boldsymbol{\xi}^k)\|_2^2 \\
&\overset{(12)}{\leq} f(x^k) - \gamma \|\nabla f(x^k)\|_2^2 - \gamma \left\langle \nabla f(x^k), \theta_k \right\rangle + L\gamma^2 \|\nabla f(x^k)\|_2^2 + L\gamma^2 \|\theta_k\|_2^2 \\
&= f(x^k) - \gamma(1 - L\gamma)\|\nabla f(x^k)\|_2^2 - \gamma \left\langle \nabla f(x^k), \theta_k \right\rangle + L\gamma^2 \|\theta_k\|_2^2 \\
&\leq f(x^k) - \frac{\gamma}{2} \|\nabla f(x^k)\|_2^2 - \gamma \left\langle \nabla f(x^k), \theta_k \right\rangle + L\gamma^2 \|\theta_k\|_2^2,
\end{aligned}
$$

$$
\theta_k \overset{\text{def}}{=} \widetilde{\nabla} f(x^k, \boldsymbol{\xi}^k) - \nabla f(x^k) \tag{113}
$$

where in the last inequality we use $1 - \gamma L \geq \frac{1}{2}$. Next, $\mu$-strong convexity of $f$ implies $\|\nabla f(x^k)\|_2^2 \geq 2\mu(f(x^k) - f(x^*))$ and

$$
\begin{aligned}
f(x^{k+1}) - f(x^*) &\leq f(x^k) - f(x^*) - \gamma\mu(f(x^k) - f(x^*)) - \gamma \left\langle \nabla f(x^k), \theta_k \right\rangle + L\gamma^2 \|\theta_k\|_2^2 \\
&= (1 - \gamma\mu)(f(x^k) - f(x^*)) - \gamma \left\langle \nabla f(x^k), \theta_k \right\rangle + L\gamma^2 \|\theta_k\|_2^2.
\end{aligned}
$$

Unrolling the recurrence we obtain

$$
\begin{aligned}
f(x^N) - f(x^*) &\leq (1 - \gamma\mu)^N (f(x^0) - f(x^*)) + \gamma \sum_{l=0}^{N-1} (1 - \gamma\mu)^{N-1-l} \left\langle -\nabla f(x^l), \theta_l \right\rangle \\
&\quad + L\gamma^2 \sum_{l=0}^{N-1} (1 - \gamma\mu)^{N-1-l} \|\theta_l\|_2^2,
\end{aligned} \tag{114}
$$

for all $N \geq 0$. Using notation $r_k \overset{\text{def}}{=} f(x^k) - f(x^*)$ we rewrite this inequality in the following form:

$$
r_k \leq (1 - \gamma\mu)^k r_0 + \gamma \sum_{l=0}^{k-1} (1 - \gamma\mu)^{k-1-l} \left\langle -\nabla f(x^l), \theta_l \right\rangle + L\gamma^2 \sum_{l=0}^{k-1} (1 - \gamma\mu)^{k-1-l} \|\theta_l\|_2^2. \tag{115}
$$

The rest of the proof is based on the refined analysis of inequality (115). In particular, via induction we prove that for all $k = 0, 1, \ldots, N$ with probability at least $1 - \frac{k\beta}{N}$ the following statement holds: inequalities

$$
\begin{aligned}
r_t &\overset{(115)}{\leq} (1 - \gamma\mu)^t r_0 + \gamma \sum_{l=0}^{t-1} (1 - \gamma\mu)^{t-1-l} \left\langle -\nabla f(x^l), \theta_l \right\rangle + L\gamma^2 \sum_{l=0}^{t-1} (1 - \gamma\mu)^{t-1-l} \|\theta_l\|_2^2 \\
&\leq 2(1 - \gamma\mu)^t r_0
\end{aligned} \tag{116}
$$

hold for $t = 0, 1, \ldots, k$ simultaneously. Let us define the probability event when this statement holds as $E_k$. Then, our goal is to show that $\mathbb{P}\{E_k\} \geq 1 - \frac{k\beta}{N}$ for all $k = 0, 1, \ldots, N$. For $t = 0$ inequality (116) holds with probability 1 since $2(1 - \gamma\mu)^0 \geq 1$, hence $\mathbb{P}\{E_0\} = 1$. Next, assume that for some $k = T - 1 \leq N - 1$ we have $\mathbb{P}\{E_k\} = \mathbb{P}\{E_{T-1}\} \geq 1 - \frac{(T-1)\beta}{N}$. Let us prove that $\mathbb{P}\{E_T\} \geq 1 - \frac{T\beta}{N}$. First of all, probability event $E_{T-1}$ implies that

$$
f(x^t) - f(x^*) \overset{(116)}{\leq} 2(1 - \gamma\mu)^t r_0 \tag{117}
$$

hold for $t = 0, 1, \ldots, T - 1$. Since $f$ is $L$-smooth, we have that probability event $E_{T-1}$ implies

$$
\left\| \nabla f(x^l) \right\|_2 \leq \sqrt{2L(f(x^l) - f(x^*))} \leq \sqrt{4L(1 - \gamma\mu)^l r_0} = \frac{\lambda_l}{2} \tag{118}
$$

for $t = 0, \ldots, T - 1$ and

$$
\lambda_l = 4\sqrt{L(1 - \gamma\mu)^l r_0}. \tag{119}
$$

Having inequalities (118) in hand we show in the rest of the proof that (116) holds for $t = T$ with big enough probability. First of all, we introduce new random variables:

$$\zeta_l = \begin{cases} -\nabla f(x^{l+1}), & \text{if } \|\nabla f(x^{l+1})\|_2 \le \frac{\lambda_l}{2}, \\ 0, & \text{otherwise,} \end{cases} \tag{120}$$

for $l = 0, 1, \ldots T - 1$. Note that these random variables are bounded with probability 1, i.e. with probability 1 we have

$$\|\zeta_l\|_2 \le \frac{\lambda_l}{2}. \tag{121}$$

Secondly, we use the introduced notation and get that $E_{T-1}$ implies

$$r_T \overset{(115),(116),(118),(120)}{\le} (1 - \gamma\mu)^T r_0 + \gamma \sum_{l=0}^{T-1} (1 - \gamma\mu)^{T-1-l} \langle \zeta_l, \theta_l \rangle$$

$$+ L\gamma^2 \sum_{l=0}^{T-1} (1 - \gamma\mu)^{T-1-l} \|\theta_l\|_2^2.$$

Finally, we do some preliminaries in order to apply Bernstein's inequality (see Lemma D.1) and obtain that $E_{T-1}$ implies

$$r_T \overset{(12)}{\le} (1 - \gamma\mu)^T r_0 + \underbrace{\gamma \sum_{l=0}^{T-1} (1 - \gamma\mu)^{T-1-l} \langle \theta_l^u, \zeta_l \rangle}_{①} + \underbrace{\gamma \sum_{l=0}^{T-1} (1 - \gamma\mu)^{T-1-l} \langle \theta_l^b, \zeta_l \rangle}_{②}$$

$$+ \underbrace{2L\gamma^2 \sum_{l=0}^{T-1} (1 - \gamma\mu)^{T-1-l} \left( \|\theta_l^u\|_2^2 - \mathbb{E}_{\boldsymbol{\xi}^l} \left[ \|\theta_l^u\|_2^2 \right] \right)}_{③}$$

$$+ \underbrace{2L\gamma^2 \sum_{l=0}^{T-1} (1 - \gamma\mu)^{T-1-l} \mathbb{E}_{\boldsymbol{\xi}^l} \left[ \|\theta_l^u\|_2^2 \right]}_{④} + \underbrace{2L\gamma^2 \sum_{l=0}^{T-1} (1 - \gamma\mu)^{T-1-l} \|\theta_l^b\|_2^2}_{⑤} \tag{122}$$

where we introduce new notations:

$$\theta_l^u \overset{\text{def}}{=} \widetilde{\nabla} f(x^l, \boldsymbol{\xi}^l) - \mathbb{E}_{\boldsymbol{\xi}^l} \left[ \widetilde{\nabla} f(x^l, \boldsymbol{\xi}^l) \right], \quad \theta_l^b \overset{\text{def}}{=} \mathbb{E}_{\boldsymbol{\xi}^l} \left[ \widetilde{\nabla} f(x^l, \boldsymbol{\xi}^l) \right] - \nabla f(x^l), \tag{123}$$

$$\theta_l = \theta_l^u + \theta_l^b.$$

It remains to provide tight upper bounds for ①, ②, ③, ④ and ⑤, i.e. in the remaining part of the proof we show that $① + ② + ③ + ④ + ⑤ \le (1 - \gamma\mu)^T r_0$.

**Upper bound for ①.** First of all, since $\mathbb{E}_{\boldsymbol{\xi}^l}[\theta_l^u] = 0$ summands in ① are conditionally unbiased:

$$\mathbb{E}_{\boldsymbol{\xi}^l} \left[ \gamma(1 - \gamma\mu)^{T-1-l} \langle \theta_l^u, \zeta_l \rangle \right] = 0.$$

Secondly, these summands are bounded with probability 1:

$$\left| \gamma(1 - \gamma\mu)^{T-1-l} \langle \theta_l^u, \zeta_l \rangle \right| \le \gamma(1 - \gamma\mu)^{T-1-l} \|\theta_l^u\|_2 \|\zeta_l\|_2$$

$$\overset{(38),(121)}{\le} \gamma(1 - \gamma\mu)^{T-1-l} \lambda_l^2 \overset{(119)}{=} 16\gamma L r_0 (1 - \gamma\mu)^{T-1}.$$

Finally, one can bound conditional variances $\sigma_l^2 \overset{\text{def}}{=} \mathbb{E}_{\boldsymbol{\xi}^l} \left[ \gamma^2 (1 - \gamma\mu)^{2(T-1-l)} \langle \theta_l^u, \zeta_l \rangle^2 \right]$ in the following way:

$$\sigma_l^2 \le \mathbb{E}_{\boldsymbol{\xi}^l} \left[ \gamma^2 (1 - \gamma\mu)^{2(T-1-l)} \|\theta_l^u\|_2^2, \|\zeta_l\|_2^2 \right] \overset{(121)}{\le} \gamma^2 (1 - \gamma\mu)^{2(T-1-l)} \frac{\lambda^2}{4} \mathbb{E}_{\boldsymbol{\xi}^l} \left[ \|\theta_l^u\|_2^2 \right]$$

$$\overset{(119)}{\le} 4\gamma^2 L r_0 (1 - \gamma\mu)^{2(T-1)-l} \mathbb{E}_{\boldsymbol{\xi}^l} \left[ \|\theta_l^u\|_2^2 \right]. \tag{124}$$

In other words, sequence $\left\{\gamma(1-\gamma\mu)^{T-1-l}\langle\theta_l^u,\zeta_l\rangle\right\}_{l\geq0}$ is a bounded martingale difference sequence with bounded conditional variances $\{\sigma_l^2\}_{l\geq0}$. Therefore, we can apply Bernstein's inequality, i.e. we apply Lemma D.1 with $X_l=\gamma(1-\gamma\mu)^{T-1-l}\langle\theta_l^u,\zeta_l\rangle$, $c=16\gamma Lr_0(1-\gamma\mu)^{T-1}$ and $F=\frac{c^2\ln\frac{4N}{\beta}}{6}$ and get that for all $b>0$

$$\mathbb{P}\left\{\left|\sum_{l=0}^{T-1}X_l\right|>b\text{ and }\sum_{l=0}^{T-1}\sigma_l^2\leq F\right\}\leq2\exp\left(-\frac{b^2}{2F+2cb/3}\right)$$

or, equivalently, with probability at least $1-2\exp\left(-\frac{b^2}{2F+2cb/3}\right)$

$$\text{either }\sum_{l=0}^{T-1}\sigma_l^2>F\quad\text{or}\quad\underbrace{\left|\sum_{l=0}^{T-1}X_l\right|}_{|\textcircled{1}|}\leq b.$$

The choice of $F$ will be clarified further, let us now choose $b$ in such a way that $2\exp\left(-\frac{b^2}{2F+2cb/3}\right)=\frac{\beta}{2N}$. This implies that $b$ is the positive root of the quadratic equation

$$b^2-\frac{2c\ln\frac{4N}{\beta}}{3}b-2F\ln\frac{4N}{\beta}=0,$$

hence

$$\begin{aligned}b&=\frac{c\ln\frac{4N}{\beta}}{3}+\sqrt{\frac{c^2\ln^2\frac{4N}{\beta}}{9}+2F\ln\frac{4N}{\beta}}=\frac{c\ln\frac{4N}{\beta}}{3}+\sqrt{\frac{4c^2\ln^2\frac{4N}{\beta}}{9}}\\&=c\ln\frac{4N}{\beta}=16\gamma Lr_0(1-\gamma\mu)^{T-1}\ln\frac{4N}{\beta}.\end{aligned}$$

That is, with probability at least $1-\frac{\beta}{2N}$

$$\underbrace{\text{either }\sum_{l=0}^{T-1}\sigma_l^2>F\quad\text{or}\quad|\textcircled{1}|\leq16\gamma Lr_0(1-\gamma\mu)^{T-1}\ln\frac{4N}{\beta}}_{\text{probability event }E_{\textcircled{1}}}.$$

Next, we notice that probability event $E_{T-1}$ implies that

$$\begin{aligned}\sum_{l=0}^{T-1}\sigma_l^2&\overset{(124)}{\leq}4\gamma^2Lr_0\sigma^2(1-\gamma\mu)^{2(T-1)}\sum_{l=0}^{T-1}\mathbb{E}_{\boldsymbol{\xi}^l}\left[\left\|\theta_l^u\right\|_2^2\right]\\&\overset{(41)}{\leq}72\gamma^2Lr_0\sigma^2(1-\gamma\mu)^{2(T-1)}\sum_{l=0}^{T-1}\frac{1}{m_l(1-\gamma\mu)^l}\\&\overset{(96)}{\leq}\frac{128}{3}\gamma^2L^2r_0^2(1-\gamma\mu)^{2(T-1)}\ln\frac{4N}{\beta}=\frac{c^2\ln\frac{4N}{\beta}}{6}=F,\end{aligned}$$

where the last inequality follows from $c=16\gamma Lr_0(1-\gamma\mu)^{T-1}$ and simple arithmetic.

**Upper bound for ②.** First of all, we notice that probability event $E_{T-1}$ implies

$$\begin{aligned}\gamma(1-\gamma\mu)^{T-1-l}\langle\theta_l^b,\zeta_l\rangle&\leq\gamma(1-\gamma\mu)^{T-1-l}\left\|\theta_l^b\right\|_2\left\|\zeta_l\right\|_2\\&\overset{(39),(121)}{\leq}\gamma(1-\gamma\mu)^{T-1-l}\frac{4\sigma^2}{m_l\lambda_l}\frac{\lambda_l}{2}\\&=\frac{2\sigma^2\gamma(1-\gamma\mu)^{T-1-l}\sigma^2}{m_l}\\&\overset{(96)}{=}\frac{64}{27}\frac{\gamma Lr_0(1-\gamma\mu)^{T-1}\ln\frac{4N}{\beta}}{N}.\end{aligned}$$

This implies that

$$② = \sum_{l=0}^{T-1} \gamma(1-\gamma\mu)^{T-1-l} \langle \theta_l^b, \zeta_l \rangle \overset{T \le N}{\le} \frac{64}{27}\gamma L r_0 (1-\gamma\mu)^{T-1} \ln \frac{4N}{\beta}.$$

**Upper bound for ③.** We derive the upper bound for ③ using the same technique as for ①. First of all, we notice that the summands in ③ are conditionally independent:

$$\mathbb{E}_{\boldsymbol{\xi}^l}\left[2L\gamma^2(1-\gamma\mu)^{T-1-l}\left(\|\theta_l^u\|_2^2 - \mathbb{E}_{\boldsymbol{\xi}^l}\left[\|\theta_l^u\|_2^2\right]\right)\right] = 0.$$

Secondly, the summands are bounded with probability 1:

$$
\begin{aligned}
\left|2L\gamma^2(1-\gamma\mu)^{T-1-l}\left(\|\theta_l^u\|_2^2 - \mathbb{E}_{\boldsymbol{\xi}^l}\left[\|\theta_l^u\|_2^2\right]\right)\right| &\le 2L\gamma^2(1-\gamma\mu)^{T-1-l}\left(\|\theta_l^u\|_2^2 + \mathbb{E}_{\boldsymbol{\xi}^l}\left[\|\theta_l^u\|_2^2\right]\right) \\
&\overset{(38)}{\le} 2L\gamma^2(1-\gamma\mu)^{T-1-l}\left(4\lambda_l^2 + 4\lambda_l^2\right) \\
&\overset{(119)}{=} 256\gamma^2 L^2 r_0 (1-\gamma\mu)^{T-1} \overset{\text{def}}{=} c_1. \quad (125)
\end{aligned}
$$

Finally, one can bound conditional variances $\hat{\sigma}_l^2 \overset{\text{def}}{=} \mathbb{E}_{\boldsymbol{\xi}^l}\left[\left|2L\gamma^2(1-\gamma\mu)^{T-1-l}\left(\|\theta_l^u\|_2^2 - \mathbb{E}_{\boldsymbol{\xi}^l}\left[\|\theta_l^u\|_2^2\right]\right)\right|^2\right]$ in the following way:

$$
\begin{aligned}
\hat{\sigma}_l^2 &\overset{(125)}{\le} c_1 \mathbb{E}_{\boldsymbol{\xi}^l}\left[\left|2L\gamma^2(1-\gamma\mu)^{T-1-l}\left(\|\theta_l^u\|_2^2 - \mathbb{E}_{\boldsymbol{\xi}^l}\left[\|\theta_l^u\|_2^2\right]\right)\right|\right] \\
&\le 2L\gamma^2(1-\gamma\mu)^{T-1-l}c_1 \mathbb{E}_{\boldsymbol{\xi}^l}\left[\|\theta_l^u\|_2^2 + \mathbb{E}_{\boldsymbol{\xi}^l}\left[\|\theta_l^u\|_2^2\right]\right] \\
&= 4L\gamma^2(1-\gamma\mu)^{T-1-l}c_1 \mathbb{E}_{\boldsymbol{\xi}^l}\left[\|\theta_l^u\|_2^2\right]. \quad (126)
\end{aligned}
$$

In other words, sequence $\left\{2L\gamma^2(1-\gamma\mu)^{T-1-l}\left(\|\theta_l^u\|_2^2 - \mathbb{E}_{\boldsymbol{\xi}^l}\left[\|\theta_l^u\|_2^2\right]\right)\right\}_{l \ge 0}$ is a bounded martingale difference sequence with bounded conditional variances $\{\hat{\sigma}_l^2\}_{l \ge 0}$. Therefore, we can apply Bernstein's inequality, i.e. we apply Lemma D.1 with $X_l = \hat{X}_l = 2L\gamma^2(1-\gamma\mu)^{T-1-l}\left(\|\theta_l^u\|_2^2 - \mathbb{E}_{\boldsymbol{\xi}^l}\left[\|\theta_l^u\|_2^2\right]\right)$, $c = c_1 = 256\gamma^2 L^2 r_0 (1-\gamma\mu)^{T-1}$ and $F = F_1 = \frac{c_1^2 \ln \frac{4N}{\beta}}{6}$ and get that for all $b > 0$

$$\mathbb{P}\left\{\left|\sum_{l=0}^{T-1} \hat{X}_l\right| > b \text{ and } \sum_{l=0}^{T-1} \hat{\sigma}_l^2 \le F_1\right\} \le 2\exp\left(-\frac{b^2}{2F_1 + 2c_1 b/3}\right)$$

or, equivalently, with probability at least $1 - 2\exp\left(-\frac{b^2}{2F_1 + 2c_1 b/3}\right)$

$$\text{either } \sum_{l=0}^{T-1} \hat{\sigma}_l^2 > F_1 \quad \text{or} \quad \underbrace{\left|\sum_{l=0}^{T-1} \hat{X}_l\right|}_{|③|} \le b.$$

As in our derivations of the upper bound for ① we choose such $b$ that $2\exp\left(-\frac{b^2}{2F_1 + 2c_1 b/3}\right) = \frac{\beta}{2N}$, i.e.

$$b = \frac{c_1 \ln \frac{4N}{\beta}}{3} + \sqrt{\frac{c_1^2 \ln^2 \frac{4N}{\beta}}{9} + 2F_1 \ln \frac{4N}{\beta}} = c_1 \ln \frac{4N}{\beta} = 256\gamma^2 L^2 r_0 (1-\gamma\mu)^{T-1} \ln \frac{4N}{\beta}.$$

That is, with probability at least $1 - \frac{\beta}{2N}$

$$\underbrace{\text{either } \sum_{l=0}^{T-1} \hat{\sigma}_l^2 > F_1 \quad \text{or} \quad |③| \le 256\gamma^2 L^2 r_0 (1-\gamma\mu)^{T-1} \ln \frac{4N}{\beta}.}_{\text{probability event } E_③}$$

Next, we notice that probability event $E_{T-1}$ implies that

$$
\begin{aligned}
\sum_{l=0}^{T-1} \hat{\sigma}_l^2 &\overset{(126)}{\le} 4L\gamma^2(1-\gamma\mu)^{T-1}c_1 \sum_{l=0}^{T-1} \frac{1}{(1-\gamma\mu)^l} \mathbb{E}_{\boldsymbol{\xi}^l}\left[\|\theta_l^u\|_2^2\right] \\
&\overset{(41)}{\le} 72L\gamma^2(1-\gamma\mu)^{T-1}c_1\sigma^2 \sum_{l=0}^{T-1} \frac{1}{(1-\gamma\mu)^l} \frac{1}{m_l} \overset{(96), T \le N}{\le} \frac{c_1^2 \ln \frac{4N}{\beta}}{6} = F_1.
\end{aligned}
$$

**Upper bound for ④.** The probability event $E_{T-1}$ implies

$$④ = 2L\gamma^2 \sum_{l=0}^{T-1}(1-\gamma\mu)^{T-1-l}\mathbb{E}_{\boldsymbol{\xi}^l}\left[\|\theta_l^u\|_2^2\right] \overset{(41)}{\leq} 2L\gamma^2(1-\gamma\mu)^{T-1}\sum_{l=0}^{T-1}\frac{1}{(1-\gamma\mu)^l}\frac{18\sigma^2}{m_l}$$

$$\overset{(96),T\leq N}{\leq} \frac{64}{3}\gamma^2 L^2 r_0(1-\gamma\mu)^{T-1}\ln\frac{4N}{\beta}.$$

**Upper bound for ⑤.** Again, we use corollaries of probability event $E_{T-1}$:

$$⑤ = 2L\gamma^2 \sum_{l=0}^{T-1}(1-\gamma\mu)^{T-1-l}\|\theta_l^b\|_2^2 \overset{(39)}{\leq} 2L\gamma^2(1-\gamma\mu)^{T-1}\sum_{l=0}^{T-1}\frac{1}{(1-\gamma\mu)^l}\frac{16\sigma^4}{m_l^2\lambda_l^2}$$

$$\overset{(119),(96)}{=} \frac{512}{729}\gamma^2 L^2 r_0(1-\gamma\mu)^{T-1}\ln^2\frac{4N}{\beta}\sum_{l=0}^{T-1}\frac{1}{N^2}$$

$$\overset{T\leq N}{\leq} \frac{512}{729}\frac{\gamma^2 L^2 r_0(1-\gamma\mu)^{T-1}\ln^2\frac{4N}{\beta}}{N}.$$

Now we summarize all bounds that we have: probability event $E_{T-1}$ implies

$$r_T \overset{(115)}{\leq} (1-\gamma\mu)^T r_0 + \gamma\sum_{l=0}^{T-1}(1-\gamma\mu)^{T-1-l}\left\langle -\nabla f(x^l),\theta_l\right\rangle + L\gamma^2\sum_{l=0}^{T-1}(1-\gamma\mu)^{T-1-l}\|\theta_l\|_2^2$$

$$\overset{(122)}{\leq} (1-\gamma\mu)^T r_0 + ① + ② + ③ + ④ + ⑤,$$

$$② \leq \frac{32}{27}\gamma L r_0(1-\gamma\mu)^{T-1}\ln\frac{4N}{\beta}, \quad ④ \leq \frac{64}{3}\gamma^2 L^2 r_0(1-\gamma\mu)^{T-1}\ln\frac{4N}{\beta},$$

$$⑤ \leq \frac{512}{729}\frac{\gamma^2 L^2 r_0(1-\gamma\mu)^{T-1}\ln^2\frac{4N}{\beta}}{N}, \quad \sum_{l=0}^{T-1}\sigma_l^2 \leq F, \quad \sum_{l=0}^{T-1}\hat\sigma_l^2 \leq F_1$$

and

$$\mathbb{P}\{E_{T-1}\} \geq 1 - \frac{(T-1)\beta}{N}, \quad \mathbb{P}\{E_①\} \geq 1 - \frac{\beta}{2N}, \quad \mathbb{P}\{E_③\} \geq 1 - \frac{\beta}{2N},$$

where

$$E_① = \left\{\text{either }\sum_{l=0}^{T-1}\sigma_l^2 > F \quad\text{or}\quad |①| \leq 16\gamma L r_0(1-\gamma\mu)^{T-1}\ln\frac{4N}{\beta}\right\},$$

$$E_③ = \left\{\text{either }\sum_{l=0}^{T-1}\hat\sigma_l^2 > F_1 \quad\text{or}\quad |③| \leq 256\gamma^2 L^2 r_0(1-\gamma\mu)^{T-1}\ln\frac{4N}{\beta}\right\}.$$

Taking into account these inequalities and our assumptions on $m_k$ and $\gamma$ (see (96) and (97)) we get that probability event $E_{T-1}\cap E_①\cap E_③$ implies

$$r_T \overset{(115)}{\leq} (1-\gamma\mu)^T r_0 + \gamma\sum_{l=0}^{T-1}(1-\gamma\mu)^{T-1-l}\left\langle -\nabla f(x^l),\theta_l\right\rangle + L\gamma^2\sum_{l=0}^{T-1}(1-\gamma\mu)^{T-1-l}\|\theta_l\|_2^2$$

$$\leq (1-\gamma\mu)^T r_0 + \left(\frac{1}{5}+\frac{1}{5}+\frac{1}{5}+\frac{1}{5}+\frac{1}{5}\right)(1-\gamma\mu)^T r_0 = 2(1-\gamma\mu)^T r_0. \qquad (127)$$

Moreover, using union bound we derive

$$\mathbb{P}\left\{E_{T-1}\cap E_①\cap E_③\right\} = 1 - \mathbb{P}\left\{\overline{E}_{T-1}\cup\overline{E}_①\cup\overline{E}_③\right\} \geq 1 - \frac{T\beta}{N}. \qquad (128)$$

That is, by definition of $E_T$ and $E_{T-1}$ we have proved that

$$\mathbb{P}\{E_T\} \overset{(127)}{\geq} \mathbb{P}\left\{E_{T-1}\cap E_①\cap E_③\right\} \overset{(128)}{\geq} 1 - \frac{T\beta}{N},$$

which implies that for all $k = 0, 1, \ldots, N$ we have $\mathbb{P}\{E_k\} \geq 1 - \frac{k\beta}{N}$. Then, for $k = N$ we have that with probability at least $1 - \beta$

$$f(x^N) - f(x^*)) \overset{(114)}{\leq} (1 - \gamma\mu)^N (f(x^0) - f(x^*)) + \gamma \sum_{l=0}^{N-1} (1 - \gamma\mu)^{N-1-l} \left\langle -\nabla f(x^l), \theta_l \right\rangle$$

$$+ L\gamma^2 \sum_{l=0}^{N-1} (1 - \gamma\mu)^{N-1-l} \|\theta_l\|_2^2 \overset{(116)}{\leq} 2(1 - \gamma\mu)^N (f(x^0) - f(x^*)) \quad (129)$$

As a result, we get that with probability at least $1 - \beta$

$$
\begin{aligned}
f(x^N) - f(x^*) &\leq 2(1 - \gamma\mu)^N (f(x^0) - f(x^*)) \leq 2\exp\left(-\gamma\mu N\right)(f(x^0) - f(x^*)) \\
&\overset{(97)}{\leq} 2\exp\left(-\frac{\mu N}{80L \ln\frac{4N}{\beta}}\right)(f(x^0) - f(x^*)).
\end{aligned}
$$

In other words, clipped-SGD achieves $f(x^N) - f(x^*) \leq \varepsilon$ with probability at least $1 - \beta$ after

$$O\left(\frac{L}{\mu}\ln\left(\frac{r_0}{\varepsilon}\right)\ln\left(\frac{L}{\mu\beta}\ln\left(\frac{r_0}{\varepsilon}\right)\right)\right)$$

iterations, where $r_0 = f(x^0) - f(x^*)$ and requires

$$
\begin{aligned}
\sum_{k=0}^{N-1} m_k &\overset{(96)}{=} \sum_{k=0}^{N-1} O\left(\max\left\{1, \frac{N\sigma^2}{Lr_0(1 - \gamma\mu)^k \ln\frac{4N}{\beta}}\right\}\right) \\
&\overset{(97)}{=} O\left(\max\left\{N, \frac{N\sigma^2}{\mu r_0 (1 - \gamma\mu)^{N-1}}\right\}\right) = O\left(\max\left\{N, \frac{N\sigma^2}{\mu\varepsilon}\right\}\right) \\
&= O\left(\max\left\{\frac{L}{\mu}, \frac{\sigma^2}{\mu\varepsilon} \cdot \frac{L}{\mu}\right\}\ln\left(\frac{r_0}{\varepsilon}\right)\ln\left(\frac{L}{\mu\beta}\ln\left(\frac{r_0}{\varepsilon}\right)\right)\right).
\end{aligned}
$$

oracle calls.

## H  Extra Experiments

### H.1  Detailed Description of Experiments from Section 1.2

In this section we provide a detailed description of experiments from Section 1.2 together with additional experiments. In these experiments we consider the following problem:

$$\min_{x\in\mathbb{R}^n} f(x), \quad f(x) = \|x\|_2^2/2 = \mathbb{E}_\xi\left[f(x,\xi)\right], \quad f(x,\xi) = \|x\|_2^2/2 + \langle \xi, x\rangle \tag{130}$$

where $\xi$ is a random vector with zero mean and bounded variance. Clearly, $f(x)$ is $\mu$-strongly convex and $L$-smooth with $\mu = L = 1$. We assume that $\mathbb{E}\left[\|\xi\|_2^2\right] \leq \sigma^2$ for some non-negative number $\sigma$. Then, the stochastic gradient $\nabla f(x,\xi) = x + \xi$ satisfies conditions (2) and the state-of-the-art theory (e.g. [24, 25]) says that after $k$ iterations of SGD with constant stepsize $\gamma \leq 1/L = 1$ we have $\mathbb{E}\left[\|x^k - x^*\|_2^2\right] \leq (1 - \gamma\mu)^k \|x^0 - x^*\|_2^2 + \gamma\sigma^2/\mu$. Taking into account that for our problem $x^* = 0$, $f(x) = \frac{1}{2}\|x\|_2^2$, $f(x^*) = 0$ and $\mu = 1$ we derive

$$\mathbb{E}\left[f(x^k) - f(x^*)\right] \leq (1-\gamma)^k\left(f(x^0) - f(x^*)\right) + \gamma\sigma^2/2. \tag{131}$$

That is, for given $k$ the r.h.s. of the formula above depends only on the stepsize $\gamma$, initial suboptimality $f(x^0) - f(x^*)$ and the variance $\sigma$.

We emphasize that the obtained bound and the convergence in expectation itself does not imply non-trivial upper bound for $f(x^k) - f(x^*)$ with high-probability without additional assumptions on the distribution of random vector $\xi$. In fact, the trajectory of SGD significantly depends on the distribution of $\xi$. To illustrate this we consider 3 different distributions of $\xi$ with the same $\sigma$.

1. In the first case we consider $\xi$ from standard normal distribution, i.e. $\xi$ is a Gaussian random vector with zero mean and covariance matrix $I$. Clearly, in this situation $\sigma^2 = n$.

2. Next, we consider a random vector $\xi$ with i.i.d. components having Weibull distribution [69]. The cumulative distribution function (CDF) for Weibull distribution with parameters $c > 0$ and $\alpha > 0$ is

$$\text{CDF}_W(x) = \begin{cases} 1 - \exp\left(-\left(\frac{x}{\alpha}\right)^c\right), & \text{if } x \geq 0, \\ 0, & \text{if } x < 0. \end{cases} \tag{132}$$

   There are explicit formulas for mean and variance for Weibull distribution:

$$\text{mean} = \alpha\Gamma\left(1 + \frac{1}{c}\right), \quad \text{variance} = \alpha^2\left(\Gamma\left(1 + \frac{2}{c}\right) - \left(\Gamma\left(1 + \frac{1}{c}\right)\right)^2\right),$$

   where $\Gamma$ denotes the gamma function. Having these formulas one can easily shift and scale the distribution in order to get a random variable with zero mean and the variance equal 1. In our experiments, we take $c = 0.2$,

$$\alpha = \frac{1}{\sqrt{\Gamma\left(1 + \frac{2}{c}\right) - \left(\Gamma\left(1 + \frac{1}{c}\right)\right)^2}},$$

   shift the distribution by $-\alpha\Gamma\left(1 + \frac{1}{c}\right)$ and sample from the obtained distribution $n$ i.i.d. random variables to form $\xi$. Such a choice of parameters implies that $\mathbb{E}[\xi] = 0$ and $\mathbb{E}[\|\xi\|_2^2] = n$.

3. Finally, we consider a random vector $\xi$ with i.i.d. components having Burr Type XII distribution [3] having the following cumulative distribution function

$$\text{CDF}_B(x) = \begin{cases} 1 - (1 + x^c)^{-d}, & \text{if } x > 0, \\ 0, & \text{if } x \leq 0, \end{cases} \tag{133}$$

   where $c > 0$ and $d > 0$ are the positive parameters. There are explicit formulas for mean and variance for Burr distribution:

$$\text{mean} = \mu_1, \quad \text{variance} = -\mu_1^2 + \mu_2,$$

where the $r$-th moment (if exists) is defined as follows [42]:

$$\mu_r = d\mathrm{B}\left(\frac{cd-r}{c}, \frac{c+r}{c}\right),$$

where B denotes the beta function.

In our experiments, we take $c = 1$ and $d = 2.3$ and then apply shifts and scales similarly to the case with Weibull distribution. Again, such a choice of parameters implies that $\mathbb{E}[\xi] = 0$ and $\mathbb{E}[\|\xi\|_2^2] = n$.

For all experiments we considered the dimension $n = 100$, the stepsize $\gamma = 0.001$ and for `clipped-SGD` we set $\lambda = 100$. The result of 10 independent runs of `SGD` and `clipped-SGD` are presented in Figures 6-10. These numerical tests show that for Weibull and Burr Type XII distributions `SGD` have significantly larger oscillations than for Gaussian distribution in all 10 tests. In contrast, `clipped-SGD` behaves much more robust in all 3 cases during all 10 runs without significant oscillations.

Figure 6: 2 independent runs of `SGD` (blue) and `clipped-SGD` (red) applied to solve (130) with $\xi$ having Gaussian (left column), Weibull (central column) and Burr Type XII (right column) tails.

## H.2    Additional Details and Experiments with Logistic Regression

In this section, we provide additional details of the experiments presented in Section 4 together with extra numerical results. In particular, we consider the logistic regression problem:

$$\min_{x \in \mathbb{R}^n} f(x) = \frac{1}{r} \sum_{i=1}^{r} \underbrace{\log\left(1 + \exp\left(-y_i \cdot (Ax)_i\right)\right)}_{f_i(x)} \tag{134}$$

Figure 7: 2 independent runs of `SGD` (blue) and `clipped-SGD` (red) applied to solve (130) with $\xi$ having Gaussian (left column), Weibull (central column) and Burr Type XII (right column) tails.

where $A \in \mathbb{R}^{r \times n}$ is matrix of instances and $y \in \{0,1\}^r$ is vector of labels. It is well-known that $f(x)$ from (134) is convex and $L$-smooth with $L = \lambda_{\max}(A^\top A)/4r$ where $\lambda_{\max}(A^\top A)$ denotes the maximal eigenvalue of $A^\top A$. One can consider problem (134) as a special case of (1) where $\xi$ is a random index uniformly distributed on $\{1, \ldots, r\}$ and $f(x,\xi) = f_\xi(x)$. We take the datasets from LIBSVM library [4]: see Table 3 with the summary of the datasets we used.

Table 3: Summary of used datasets.

|  | heart | diabetes | australian | a9a | w8a |
|---|---|---|---|---|---|
| Size | 270 | 768 | 690 | 32561 | 49749 |
| Dimension | 13 | 8 | 13 | 123 | 300 |

We notice that in all experiments that we did with logistic regression the initial suboptimality $f(x^0) - f(x^*)$ was of order 10. Moreover, as it was mentioned in the main part of the paper the parameters for the methods were tuned. One can find parameters that we used in the experiments from Section 4 in Table 4.

Next, we provide our numerical study of the distribution of $\|\nabla f_i(x^k) - \nabla f(x^k)\|_2$, where $x^k$ is the last iterate produced by `SGD` in experiments presented in Section 4, see Figure 11. As we mentioned in the main part of the paper these histograms are very similar to ones presented in Figure 2, so, the insights that we got from Figure 2 are right. However, in our experiments with `australian` dataset `SGD` with the stepsize $\gamma = 1/L$ did not reach needed suboptimality in order to oscillate.

Therefore, we run `SGD` along with its clipped variants with the same batchsize $m = 50$ for bigger number of epochs and also tuned their parameters. One can find the results of these runs in Figure 12.

Figure 8: 2 independent runs of SGD (blue) and clipped-SGD (red) applied to solve (130) with $\xi$ having Gaussian (left column), Weibull (central column) and Burr Type XII (right column) tails.

Table 4: Parameters that are used to produce plots presented in Figures 3-5. In the first contains the name of the dataset and the batchsize $m$ that was used for all methods tested on the dataset. For d-clipped-SGD $\lambda_0$ is an initial clipping level, $l$ is a period (in terms of epochs) of decreasing the clipping level and $\alpha$ is a coefficient of decrease, i.e. every $l$ epochs the clipping level is multiplied by $\alpha$. For SSTM parameter $a$ was picked the same as for clipped-SSTM in order to emphasize the effect of clipping.

| | SGD | clipped-SGD | d-clipped-SGD | SSTM | clipped-SSTM |
|---|---|---|---|---|---|
| heart<br>$m = 20$ | $\gamma = \frac{1}{2L}$ | $\gamma = \frac{1}{2L}, \lambda = 2.72$ | $\gamma = \frac{1}{2L}, \lambda_0 = 2.72,$<br>$l = 10^3, \alpha = 0.9$ | $a = 10^4$ | $a = 10^4,$<br>$B = 2 \cdot 10^{-4}$ |
| diabetes<br>$m = 100$ | $\gamma = \frac{1}{10L}$ | $\gamma = \frac{1}{10L}, \lambda = 68.86$ | $\gamma = \frac{1}{10L}, \lambda_0 = 68.86,$<br>$l = 10^3, \alpha = 0.7$ | $a = 5 \cdot 10^3$ | $a = 5 \cdot 10^3,$<br>$B = 7 \cdot 10^{-4}$ |
| australian<br>$m = 50$ | $\gamma = \frac{1}{L}$ | $\gamma = \frac{1}{L}, \lambda = 74.47$ | $\gamma = \frac{1}{L}, \lambda_0 = 74.47,$<br>$l = 1000, \alpha = 0.9$ | $a = 10^3$ | $a = 5 \cdot 10^3,$<br>$B = 2 \cdot 10^{-4}$ |
| a9a<br>$m = 100$ | $\gamma = \frac{1}{2L}$ | $\gamma = \frac{1}{2L}, \lambda = 0.025$ | $\gamma = \frac{1}{L}, \lambda_0 = 4.9,$<br>$l = 5, \alpha = 0.5$ | $a = 1$ | $a = 1,$<br>$B = 3 \cdot 10^{-2}$ |
| w8a<br>$m = 1000$ | $\gamma = \frac{1}{L}$ | $\gamma = \frac{1}{L}, \lambda = 1.3$ | $\gamma = \frac{1}{L}, \lambda_0 = 64.78,$<br>$l = 50, \alpha = 0.9$ | $a = 1$ | $a = 1,$<br>$B = 19 \cdot 10^{-2}$ |

We see that SGD with this stepsize achieves better suboptimality but it also oscillates significantly more. In contrast, clipped-SGD and d-clipped-SGD do not have significant oscillations and converge with the same rate as SGD. Moreover, clipped-SSTM shows slightly better performance in this case. Finally, we numerically studied the distribution of $\|\nabla f_i(x^k) - \nabla f(x^k)\|_2$, where $x^k$ is the last iterate produced by SGD, see Figure 13. These histograms imply that the noise in stochastic gradients is heavy-tailed and explain an unstable behavior of SGD in this case.

Figure 9: 2 independent runs of SGD (blue) and clipped-SGD (red) applied to solve (130) with $\xi$ having Gaussian (left column), Weibull (central column) and Burr Type XII (right column) tails.

Finally, we conducted experiments on larger datasets: a9a and w8a. The results of our numerical test are reported on Figures 14 and 15. We notice that SSTM with given stepsize and batchsize suffers from noise accumulation, while clipped-SSTM does not have this drawback and shows comparable performance with SGD on a9a and much better performance on w8a.

Figure 15 shows the gradient's noise distributions for both datasets. While the distribution of stochastic gradients at the optimum for a9a have sub-Gaussian-like distribution, for w8a they have heavy-tailed distribution.

Figure 10: 2 independent runs of SGD (blue) and clipped-SGD (red) applied to solve (130) with $\xi$ having Gaussian (left column), Weibull (central column) and Burr Type XII (right column) tails.

Figure 11: Histograms of $\|\nabla f_i(x^k) - \nabla f(x^k)\|_2$ for different datasets (the first row) and synthetic Gaussian samples with mean and variance estimated via empirical mean and variance of real samples $\|\nabla f_1(x^k) - \nabla f(x^k)\|_2, \ldots, \|\nabla f_r(x^k) - \nabla f(x^k)\|_2$ (the second row) where $x^k$ is the last point produced by SGD. Red lines correspond to probability density functions of normal distributions with empirically estimated means and variances.

Figure 12: Trajectories of SGD, clipped-SGD, d-clipped-SGD and clipped-SSTM applied to solve logistic regression problem on australian dataset. For SGD and its clipped variants stepsize $\gamma = \frac{20}{L}$ was used. For clipped-SGD we used $\lambda = 18.62$ and for d-clipped-SGD the parameters are as follows: $\lambda_0 = 74.47$, $l = 1500$, $\alpha = 0.9$. Parameters for clipped-SSTM are the same as in the corresponding cell in Table 4.

Figure 13: Histograms of $\|\nabla f_i(x^k) - \nabla f(x^k)\|_2$ for australian dataset and synthetic Gaussian samples with mean and variance estimated via empirical mean and variance of real samples $\|\nabla f_1(x^k) - \nabla f(x^k)\|_2, \ldots, \|\nabla f_r(x^k) - \nabla f(x^k)\|_2$ where $x^k$ is the last point produced by SGD with $\gamma = \frac{20}{L}$. Red lines correspond to probability density functions of normal distributions with empirically estimated means and variances.

Figure 14: Trajectories of SGD, clipped-SGD, d-clipped-SGD and clipped-SSTM applied to solve logistic regression problem on a9a and w8a datasets. Parameters of the methods used in experiments are presneted in Table 4.

Figure 15: Histograms of $\|\nabla f_i(x^*)\|_2$ for a9a and w8a dataset and synthetic Gaussian samples with mean and variance estimated via empirical mean and variance of real samples $\|\nabla f_1(x^*)\|_2, \ldots, \|\nabla f_r(x^*)\|_2$. Red lines correspond to probability density functions of normal distributions with empirically estimated means and variances.