[Reviews · NeurIPS 2020]

Review 1

Summary and Contributions: This paper studies and establishes complexity bounds for clipped SGD for (strongly) convex problems, specifically in the heavy tailed noise regime. The main contributions are - present an algorithm similar to the method of similar triangles which uses clipping to deal with heavy tails in the gradient distribution - analyse this algorithm in the convex and strongly convex settings and claim new SotA bounds on complexity without requiring light tails assumptions - using a similar framework, study the effect of clipping and restarts on SGD and derive high probability complexity bounds for (strongly) convex settings - study the effects on 3 datasets from the libSVM library with logistic regression tasks (2 of which exhibit heavy tailed gradients)

Strengths: - The paper reaches a (to me) impressive result by recovering the complexity bound of AC-SA up to a logarithmic factor, while removing a lot of restrictions from the analysis - the method gives impressive results on the studied datasets - the authors give a detailed discussion of the different implications and notions

Weaknesses: - the bounds add an unfavorable batchsize dependency on the desired epsilon, which makes the batches huge to the point that it can be argued that the analysed method is almost not stochastic anymore ( m_k~ k^2N). However, the authors present numerical evidence that the algorithm can perform well with reasonable batch sizes and also present an analysis of medium and constant batch sizes which somewhat alleviates this Edit: commented on by authors - - the experimental section is limited, I'd at least test it with SVM on top of logistic regression, and maybe also on another dataset with more stochasticity (e.g. a time series dataset to fit and ARIMA, or a portfolio optimisation problem) Edit: authors fully adressed this part, adding more experiments and pointing out SVM leads to non-smooth losses (which was a "doh" mistake on my part)

Correctness: I have not been able to go through the proofs in depth, but the general arguments seem to make sense to me.

Clarity: The paper is well written and clear to me, despite not being an expert I felt I could follow the general shape of the proofs. Some qualifications: - On the triple-plots in figure 3,4 and 5 I'd propose to the authors to ensure the y axes are matched to ease comparison across algorithms - I think some parts of the batch size discussion from the supplementary to the main paper would be warranted, given that it is the main drawback of the new method.

Relation to Prior Work: I'm not an expert in this field, however I could not find any obviously missing related work

Reproducibility: Yes

Additional Feedback: If the additional experiments are added and show the large batchsize continues not to be a dealbreaker on other tasks, I feel comfortable upgrading this to an accept # After rebuttal and discussion I will upgrade my score after the authors reply (see edited weakness section) and discussion with the other reviewers. I agree with the other reviewers that a more detailed comparison with [71] is warranted in the final text, to make clear the differences in setup, not only in terms of the results (probability vs expectation)


Review 2

Summary and Contributions: This work introduces a new stochastic accelerated optimisation algorithm. More specifically, the approach targets problems where the noise in the stochastic gradients has a heavy-tailed distribution and the objective function is smooth and convex. The method is justified with novel convergence results and a set of small experiments in the convex setting.

Strengths: There are strong theoretical results to justify the proposed algorithms. In particular, the high-probability convergence bounds help characterise the behaviour of the method when the problem is very noisy, in a way that is more representative of practical performance than the standard results in expectation.

Weaknesses: The main weakness of this work is that the conditions of applicability are restrictive (in particular convexity), and thus far off most machine learning applications. Relatedly, the experiments are only performed on small and relatively simple problems by modern machine learning standards. This is of course balanced by the strong theoretical results, but it would have been great to see practical use cases of the algorithm.

Correctness: The claims and method seem correct at a high level (I have not checked the proofs). The empirical methodology seems correct and fair as well.

Clarity: The paper is well organised and easy to follow.

Relation to Prior Work: There is a detailed comparison with existing convergence results in tables 1 and 2.

Reproducibility: Yes

Additional Feedback: 1. It seems that clipped-SSTM suffers from oscillations in the experiments, could this be alleviated by the use of an iterate averaging scheme (as used by clipped-SGD)? And / or would that make the convergence proof more difficult? 2. Some applications of transfer learning can be formulated as convex problems (e.g. using a pre-trained network for feature extraction and learning only a linear classifier on-top of it). I am not sure whether they also have heavy-tail distribution, but this could be a way to show practical impact of the method on more "modern" problems. *** Post-rebuttal comment *** After reading the rebuttal, I am keeping my score and recommendation for acceptance.


Review 3

Summary and Contributions: This paper proposes a novel stochastic optimization algorithm that is able to handle “heavy-tailed” gradient noise with provable guarantees. The proposed method is then evaluated on a logistic regression problem on different datasets and the benefits of the algorithm is illustrated. In recent years, many studies have reported heavy-tailed behaviors in stochastic optimization. While the presence of such heavy-tailed behavior seems ubiquitous in machine learning, current theories often do not cover such heavy-tailed systems. In this line of research, this paper can be considered as another building brick towards building a more comprehensive theory for stochastic optimization.

Strengths: The problem statement of the paper is quite clear and so are the contributions: the authors develop a method that achieves provable guarantees under heavy-tailed gradients, where the guarantees are stronger than the state-of-the-art, as the current paper does not require a bounded domain assumption. The main contribution of the paper is mainly theoretical and I believe that it answers an important question. While I don’t believe that the paper would have a great practical impact in the general ML community, nevertheless the authors tested their algorithm in different settings and illustrated their theory in practice.

Weaknesses: * In [71] there are several theoretical guarantees both for convex and non-convex cases. I am wondering why they are not mentioned in Table 2. On the other hand, their analysis also covers the case where the domain doesn’t need to be compact. Doesn’t this reduce the novelty of this paper? I am willing to increase my grade if this concern is addressed. * A related result was provided in “On the Heavy-Tailed Theory of Stochastic Gradient Descent for Deep Neural Networks” by Simsekli et al (Sec 4). It would be interesting to see a comparison between the results in this paper and theirs. * Throughout the paper, the authors frame heavy-tails as a negative outcome; however, recently there have been several studies indicating that the heavy-tailed gradient noise can be beneficial in non-convex optimization, for instance see [65]. I’m not sure if this also applies for convex optimization, but the authors should have a discussion on this issue to cover the bigger picture and avoid being misleading. * I find the characterization of heavy-tailedness a bit weak. The authors only have a bounded second moment assumption on the gradient noise, which can allow for many different distributions with heavy-tails; however, the “heaviness” is never quantified. The theory would be much more meaningful if the authors could quantify the heaviness and link their theory to that quantification. For instance, the maximum order of the moments can be an option. (Yet I checked [47] and the assumption is similar hence the same criticism applies to prior work). * I find the uniformity of the condition in Eq2 to be rather restrictive. Can sigma depend on x in some way? * This is a minor point: very recently it has been shown that the heavy-tails can emerge in very simple settings (like linear regression with Gaussian data). See for instance: "The Heavy-Tail Phenomenon in SGD” or “Multiplicative noise and heavy tails in stochastic optimization”. In this paper, the source of the heavy tailed noise seems to be the data, which is different from that setup. Do the authors think their method can be applied to those settings as well?

Correctness: I have gone over the proofs but didn’t check them line by line. I appreciate the fact that the authors have provided a proof sketch that outlines the overall proof. Overall, it’s not clear to me what is the (novel?) mathematical technique that improves over [47]. It would be nice if the authors could comment of why [47] has to consider bounded domains and this paper does not. Also as mentioned by the authors, the proof technique seems to be adapted from [22] and [23], it would be nice if the authors could comment what technical novelty they brought on top of 22 and 23.

Clarity: The paper is well-written overall. One minor comment: y^0 and z^0 are not defined in Algorithm 1.

Relation to Prior Work: See my comments in the weakness part.

Reproducibility: Yes

Additional Feedback: Post rebuttal: Thank you for the clarifications. I am increasing my score to 7. While I am convinced by the explanations of the authors, I still do believe that the authors should provide a more explicit comparison with [71] in the paper.


Review 4

Summary and Contributions: This paper proposes a new stochastic gradient method, called clipped-SSTM because the current SGD convergence expectations does not hold in heavy-tailed noise conditions. Authors present clipped-SGD and its variations and prove its convergence bound. They also conducted experiments in small datasets (from LibSVM).

Strengths: The paper addresses a theoretically and practically important problem in machine learning, optimization under heavy-tailed stochastic noise, which are also addressed from multiple papers in optimization (listed in the paper). The paper suggests few solutions on this important problem based on clipping and shows updated convergence bounds of the solutions. While it is not the first result for clipped SGD (unliked claimed in the paper), the bounds are interesting and could be a good comparison to other proposals (e.g. cited as [71] in this paper). --------------------------------------------------------------------------------------------------------------------- Post rebuttal comment: Reviewers had active discussion on this paper, particularly what's the main contribution of this paper (particularly in conjunction with [71]). The consensus from the discussion is that the two papers (this paper and [71]) complements each other since they cover different scenarios in the long-tail. While [71] require a bounded domain condition (alpha-stable condition to be specific -- see [65] also) and this paper does not. Hence, I agree [71] does not degrades novelty and theoretical contribution of this paper too much. Please see also weakness section for some more comments.

Weaknesses: One of the weaknesses is the paper misses discussions and comparisons with significantly related research, particularly [71]. [71] also suggests gradient clipping in order to have a convergence rate on SGD under heavy-tailed noise and proposes adaptive clipping method and convergence analysis. Even the authors were aware of this work (since they cited the paper), they simply mentioned and did not perform in-depth discussion with the paper. I think the authors should have spent significant effort comparing this work with theirs. The experiment section is too weak. For example in Figure 2, I am not really sure the right two histograms are really showing they are following Gaussian unlike the leftmost one. There are too small data points. Aren’t they just outliers? Moreover, authors did not conduct any experiments on large scale experiments which also weakens their claims. Lastly, the paper is not reading smoothly. Each section was quite separated and they suddenly suggested some methods without suitable explanation (e.g. what does “Restart” mean?) Many of the essential discussions are not in the main paper making the paper hard to follow. --------------------------------------------------------------------------------------------------------------------- Post rebuttal comment: Beside of the consensus on the paper's contribution, all reviewers agreed to ask the authors to include detailed discussions with [71] should be in the main text. By believing the authors follow the suggestion, I am increasing my score. Although I am increasing my score, my concerns on the weak experiments are still valid. I hope authors can redraw the Figure 2 to be more descriptive (e.g. change the bucket size of the histogram / run the experiment on a larger data set can clearly show the noise is heavy-tailed). I believe this will make the paper stronger.

Correctness: Some of their claims need to be updated while comparing with [71]

Clarity: No. Please refer to the weaknesses section.

Relation to Prior Work: Missed many important papers such as [71]. See Weakness section for details.

Reproducibility: No

Additional Feedback: Similar to other reviewers, please add more explicit comparison with [71] in the main paper

[Author Response · NeurIPS 2020]

We thank the reviewers for their feedback and time! We are encouraged they found our theoretical results "impressive" (R1; score 6), "strong" (R2; score 7), and "interesting" (R4; score 3), noticed that our work "answers an important question" (R3; score 6), and emphasized that the paper is clearly written and well-organized (R1, R2, R3).

R1: *1) the bounds add...* Large batchsizes help us to obtain complexity guarantees beating the state-of-the-art ones. Moreover, as R1 pointed out, we also provide analysis with smaller batchsizes in Section F and empirically show that `clipped-SSTM` can work well in practice even with moderate batchsizes. *2) the exp-al section is limited...* We will add additional experiments with logistic regression on other datasets. We did not try SVM since it leads to non-smooth optimization while our paper focuses on smooth problems. *3) On the triple-plots...* We will modify the plots for the final version. *4) I think some parts...* We can add these details to the main body using an additional 9th page.

R2: *1) The main weakness of this work is...* We agree with this criticism. However, there are still a lot of important open problems in stochastic *convex* optimization that should be resolved as well. *2) It seems that clipped-SSTM suffers from oscillations...* Averaging is an interesting idea, but clipped-SSTM already suffers from oscillation significantly less than SSTM and, we guess than other accelerated stochastic methods. *3) Some applications of transfer learning...* We will try to test our methods on this task and investigate the heavy-tailedness of stochastic gradients for this problem.

R3: *1) In [71] there are several theoretical...* First of all, [71] contains the analysis of several versions of clipped-SGD establishing the rates of convergence *in expectation* while we focus on the *high-probability* complexity guarantees. Secondly, we consider convex and strongly convex cases while [71] provides an analysis in non-convex and strongly convex cases. Finally, [71] relies on the following assumption: there exist such $G > 0$ and $\alpha \in (1, 2]$ that the stochastic gradient $g(x)$ satisfies $\mathbb{E}\|g(x)\|_2^\alpha \leq G^\alpha$ for all $x \in \mathbb{R}^n$. This assumption implies the boundedness of the gradient of the objective function $f(x)$ on the whole space which is quite restrictive and contradicts to the strong convexity: there is no functions that are strongly convex and have bounded gradients on $\mathbb{R}^n$. One can argue that boundedness of the gradient is needed only on some compact, but this claim requires more refined analysis than one presented in [71]. In contrast, we assume the boundedness of the variance. Moreover, we consider *smooth* problems that allows us to accelerate clipped-SGD and obtain clipped-SSTM, while in [71] there is no analysis showing an acceleration of any algorithm. Taking all of this into account, we conclude that [71] is far from the setup we consider to be mentioned in Table 2, and it doesn't reduce the novelty of our results. *2) A related result was provided in...* Indeed, it is highly relevant, but Simsekli et al. focus on *non-convex* problems and rates of convergence *in expectation*. Anyway, we will mention this work in the final version. *3) Throughout the paper, the authors frame...* We agree that for non-convex optimization the heavy-tailed gradient noise can be beneficial for escaping bad local minima. We will add a remark about that in the final version. The reason why we treat heavy-tails as a negative outcome is that for a long time, it was not clear whether it is possible to obtain the convergence rates of AC-SA without light-tails assumption. *4) I find the characterization...* It is a good direction for further research. However, even in the setup, we consider there were important open questions addressed by our paper. *5) I find the uniformity of the condition in Eq2...* Indeed, this assumption is quite strong, but it is used in many papers on stochastic optimization. There are several extensions allowing $\sigma$ to depend on $x$ for the convergence in expectation, and it would be interesting to develop similar non-trivial extensions for the convergence with high-probability. *6) This is a minor point: very recently...* Thank you for the references. However, the examples of the problems they consider do not directly fit to the setup we focus on, but it is an interesting direction for further research. *7) Overall, it's not clear to me what is...* We use a classical clipping operator while in [47] authors apply different truncation operator ignoring the direction of the stochastic gradient if it is too big which makes the estimator less accurate in some sense. This is the main reason why in [47] boundedness of the domain is needed. Also, it seems that the direct acceleration of the method from [47] is not possible. *8) Also as mentioned by the authors, the proof technique...* In [23], authors consider the convergence in expectation and in [22] authors focus on the convergence with high-probability under "light-tails assumption" which offers them to apply Azuma–Hoeffding inequality. In our paper, we consider high-probability convergence without the "light-tails assumption" that forces us to apply Bernstein's inequality and makes the analysis more complicated. So, our analysis is novel while it reminds the approach used in [22] and [23]. *9) $y^0$ and $z^0$ are not defined...* Thanks for spotting this. The definition is $y^0 = z^0 = x^0$, we will add it.

R4: *1) While it is not the first result for clipped SGD...* We have never claimed this: we gave the first *high-probability* convergence results for clipped-SGD *without "light-tails assumption"* and also managed to *accelerate* clipped-SGD. *2) One of the weaknesses is the paper misses discussions and comparisons with significantly related research, particularly [71]...* We politely disagree. First of all, see R3 *(1)*. Secondly, we do compare our results with all relevant research that we aware of. R4 mentioned only [71] as an example of "missing comparison" which is far from the setup that we consider to be discussed in detail. *3) The experiment section is too weak...* The main contribution is theoretical, so, our experimental part is not too weak: it justifies our theoretical findings. Figure 2: the presence of the outliers can be considered as a source of heavy-tailedness. See also R1 *(2)*. *3) Lastly, the paper is not reading smoothly...* We politely disagree. Moreover, R1, R2, R3 found our work clearly written. Next, restarts technique is a classical tool in optimization (e.g., see "Gradient methods for minimizing composite objective function"). Finally, R4 claimed *"Many of the essential discussions are not in the main paper"* but did not provide any examples of such discussions. *4) Some of their claims need... Missed many important papers such as [71].* This is not true, see R4 *(1)*.

**Conclusion:** we politely disagree with the main criticism of R4 and respectfully ask R4 to increase the score.

[Meta-Review · NeurIPS 2020]

After discussion, all reviewers agree that this paper makes a good contribution to the study of clipped sgd. It substantially improves existing analysis